# HELIX: EVOLUTIONARY REINFORCEMENT LEARNING FOR OPEN-ENDED SCIENTIFIC PROBLEM SOLVING

**Chang Su**[1,2,4]**, Zhongkai Hao**[2,3]**, Zhizhou Zhang**[1]**, Zeyu Xia**[2,4]**, Youjia Wu**[1]**, Hang Su**[4]**,
Jun Zhu**[2,4]*

[1]Bosch (China) Investment Co., Ltd.,
[2]Tsinghua-Bosch Joint Center for ML, Tsinghua University,
[3]Dept. of EE, Tsinghua University,
[4]Dept. of Comp. Sci. & Techn., Tsinghua University
{su-c24, hzj21, xia-zy25}@mails.tsinghua.edu.cn,
{dcszj, suhangss}@tsinghua.edu.cn,
{Zhizhou.zhang, youjia.wu}@cn.bosch.com

## ABSTRACT

Large language models (LLMs) with reasoning abilities have demonstrated growing promise for tackling complex scientific problems. Yet such tasks are inherently domain-specific, unbounded and open-ended, demanding exploration across vast and flexible solution spaces. Existing approaches, whether purely learning-based or reliant on carefully designed workflows, often suffer from limited exploration efficiency and poor generalization. To overcome these challenges, we present **HE-LIX**—a **H**ierarchical **E**volutionary reinforcement **L**earning framework with **I**n-context e**X**periences. HELIX introduces two key novelties: (i) a diverse yet high-quality pool of candidate solutions that broadens exploration through in-context learning, and (ii) reinforcement learning for iterative policy refinement that progressively elevates solution quality. This synergy enables the discovery of more advanced solutions. On the circle packing task, HELIX achieves state-of-the-art result with a sum of radii of 2.63598308 using only a 14B model. Across standard machine learning benchmarks, HELIX further surpasses GPT-4o with a carefully engineered pipeline, delivering an average F1 improvement of 5.95 points on the Adult and Bank Marketing datasets.

## 1  INTRODUCTION

Solving complex scientific problems with large language models (LLMs) is an important and increasingly active research direction (Forootani, 2025). By leveraging and enhancing their reasoning capabilities, LLMs have demonstrated promising results in tackling challenging scientific tasks, such as symbolic regression (Shojaee et al., 2024), molecular generation (Liu et al., 2024), and difficult mathematical optimization problems (Ahmed & Choudhury, 2024). Addressing such problems holds the potential to advance the boundaries of human knowledge and reshape scientific discovery.

While LLMs have shown promising applications, complex scientific problems remain particularly challenging due to three intrinsic characteristics. First, they are *domain-specific*, with unique environments and problem-specific constraints that differ across various tasks. Second, they are *open-ended*, requiring exploration of vast and flexible solution spaces. Third, they are *unbounded*, often with no known or guaranteed global optimum.

To address these challenges, we argue that a powerful LLM for solving complex scientific problems must possess three corresponding key abilities: **(1) learning from experience**, i.e., it should enable task-specific policy adaptation by incorporating feedback from previous trials, addressing the *domain-specific* nature of each problem. **(2) Balancing quality and diversity**, i.e., it should maintain a diverse population to thoroughly explore the vast and flexible solution spaces inherent in

---

*Corresponding author.

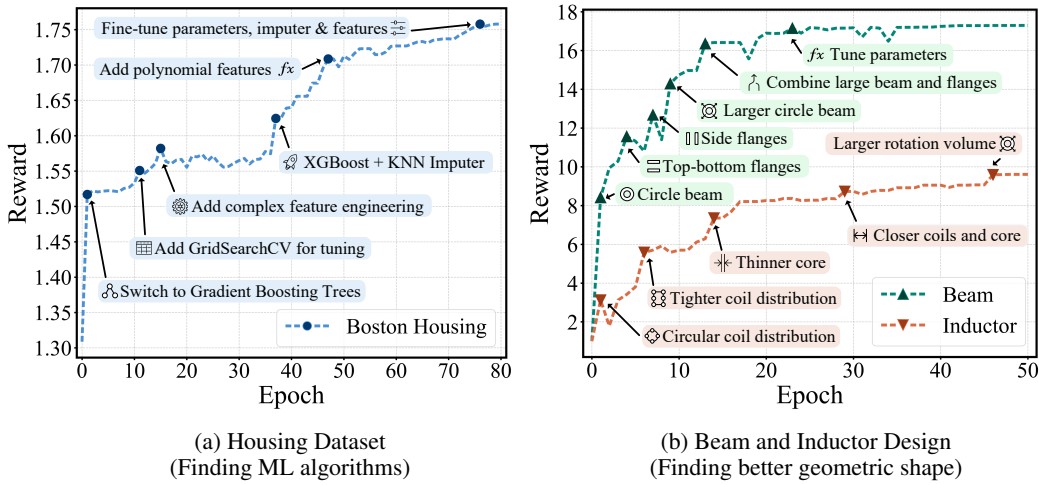

(a) Housing Dataset
(Finding ML algorithms)

(b) Beam and Inductor Design
(Finding better geometric shape)

Figure 1: The figure demonstrates how our framework progressively discovers new insights and refines solutions over iterations. **(a):** Reward curve for the housing dataset optimization, where improvements are achieved through iterative adoption of better models, parameter tuning, and feature engineering, with the final reward of 1.758 corresponding to an RMSE of 1.747. **(b):** Reward curves for the beam and inductor design tasks, where the algorithm explores novel geometries and combines favorable structural features to enhance performance.

*open-ended* tasks. **(3) Exploration based on the shoulder of giants**, i.e., it should iteratively build upon existing high-quality solutions to extend the known limits of *unbounded* problems.

However, recent works largely lack the capabilities outlined above, limiting their effectiveness on complex scientific problems. Existing approaches fall into two categories. *Post-training methods* (e.g., SFT, RLVR) fine-tune LLMs on domain-specific datasets, as in AlphaCode (Li et al., 2022) and Deepseek-R1 (Ren et al., 2025), achieving strong results in code generation and mathematical reasoning. Yet such methods often suffer from entropy collapse (Cui et al., 2025) and, as shown in Yue et al. (2025), rarely move beyond the base model's capabilities. This makes it difficult to discover fundamentally new solutions, especially when sparse rewards further limit exploration. *Workflow-driven approaches* embed LLMs in predefined pipelines to improve task-specific performance. Examples include integrating genetic algorithms with LLMs for enzyme discovery (Nana Teukam et al., 2025), establishing LLM-driven evolutionary loops such as LLaMEA (van Stein & Bäck, 2024), or applying evolutionary strategies to prompts (Agrawal et al., 2025). While effective on narrow tasks, these systems are highly sensitive to workflow design and rely on static pretrained knowledge, making it hard to reuse past discoveries to guide iterative search. Both categories thus struggle to generalize in open-ended scientific domains where efficient exploration and continual refinement are essential.

To this end, we propose **HELIX**—a **H**ierarchical **E**volutionary reinforcement **L**earning framework with **I**n-context e**X**periences. First, to learn from experience, HELIX updates the LLM policy using reward signals by reinforcement learning to progressively improve solution quality. Meanwhile candidate solutions explored by the model forms a population for evolving algorithms. Secondly, to balance the quality and diversity, we propose to rank and select samples using both diversity and reward, inspired by a classic multi-objective evolutionary algorithm named NSGA-II(Deb et al., 2002). Specifically, to better measure the novelty of a solution, we compute diversity using a pretrained language embedding model and estimate the diversity by K-Nearest Neighbors. Finally, we enable the model to stand on the shoulder of giants by adding a prompt constructed by the best solutions in the population to guide the model to generate new solutions. By using the in-context learning paradigm, we seamlessly unify and integrate evolutionary learning with reinforcement learning to explore the vast solution space in complex scientific problems.

In experiments, we evaluated HELIX on 20 tasks across five diverse categories. Compared with strong task-specific baselines and advanced proprietary models such as GPT-4o, HELIX achieves superior performance on 17 tasks, demonstrating its ability to iteratively refine solutions and update

its policy towards better results. Further analysis via ablation studies confirms that each component of HELIX contributes critically to performance. Notably, success on these unbounded and open-ended tasks suggests that iterative, diversity-aware exploration can provide useful insights for other scientific and engineering problems.

## 2 RELATED WORK

**Reinforcement learning of LLMs.** Training LLMs or LLM-based agents with reinforcement learning (RL) has recently attracted significant attention. This includes reinforcement learning from human feedback (RLHF) to align models with human preferences, as well as RL with verifiable rewards (RLVR) to enhance reasoning, mathematical problem-solving, and coding capabilities. Beyond improving reasoning, RLVR-style training can also elicit new capabilities such as tool use (Feng et al., 2025) and information retrieval (Jin et al., 2025). A representative method is GRPO (Shao et al., 2024), which normalizes rewards within groups of samples. Variants such as DAPO (Yu et al., 2025) and Dr.GRPO (Liu et al., 2025) further improve GRPO through refined data sampling strategies and advantage estimation techniques. While RL can improve generalization in specific domains, the training process often suffers from decreasing entropy and diversity over time, hindering effective exploration. Some approaches, such as KL-Cov (Cui et al., 2025), attempt to address this limitation by applying KL penalty solely to tokens with high covariance to preserve entropy. However, for complex scientific problems, these memory-less RL methods—where the sampling context for the same problem remains fixed—struggle to leverage solutions that have already been discovered, making it difficult to build upon prior explorations.

**Evolutionary algorithms.** Evolutionary algorithms are a classic approach for tackling complex optimization problems. They use "gene" to represent a solution for the problem and use random mutation to explore the whole solution space. Recently, AlphaEvolve (Novikov et al., 2025) treats code as the "gene" and applies LLM-driven mutations, successfully integrating LLM agents with evolutionary algorithms—opening the door to solving complex scientific problems. Since then, many works have adopted similar agent-based workflows to address scientific tasks such as CUDA code optimization (Lange et al., 2025), drug discovery (Gao et al., 2025), and complex scientific software usage (Fan et al., 2025; Pham et al., 2025). However, such methods typically require highly problem-specific workflow logic and prompt design, which greatly limit their effectiveness in solving more general and complex problems.

## 3 PROPOSED METHOD

### 3.1 OVERVIEW

To tackle the challenges of applying large language models (LLMs) to complex scientific discovery tasks, we propose HELIX, a hybrid framework that integrates reinforcement learning with evolutionary search. The goal is to enable LLMs to efficiently explore large and flexible solution spaces while maintaining diversity and exploiting previously discovered high-quality solutions. The framework is composed of three complementary modules: (1) **A reinforcement learning framework** that updates the policy parameters based on verifiable reward, allowing the model to *learn from experience* and progressively improve its reasoning capability. (2) **A multi-objective evolutionary mechanism** that *balancing solution quality and diversity*, ensuring that the population retains both high-performing and diverse candidates for further expansion. (3) **An in-context learning mechanism** that incorporates multiple past trials into the prompt, enabling the model to build upon previously discovered solutions and *expand its exploration on the shoulder of giants*.

We consider the task as an optimization problem that has a solution space of code. Let $s \in \mathcal{S}$ denote a candidate solution, represented as code written in a domain-specific language (e.g., Python, YAML, or other DSLs). We define an objective reward function $R(\cdot)$ which only depends on the current solution (state). The optimization objective is to find a valid $s \in \mathcal{S}$ to maximize the reward:

$$\max_{s \in \mathcal{S}} R(s). \tag{1}$$

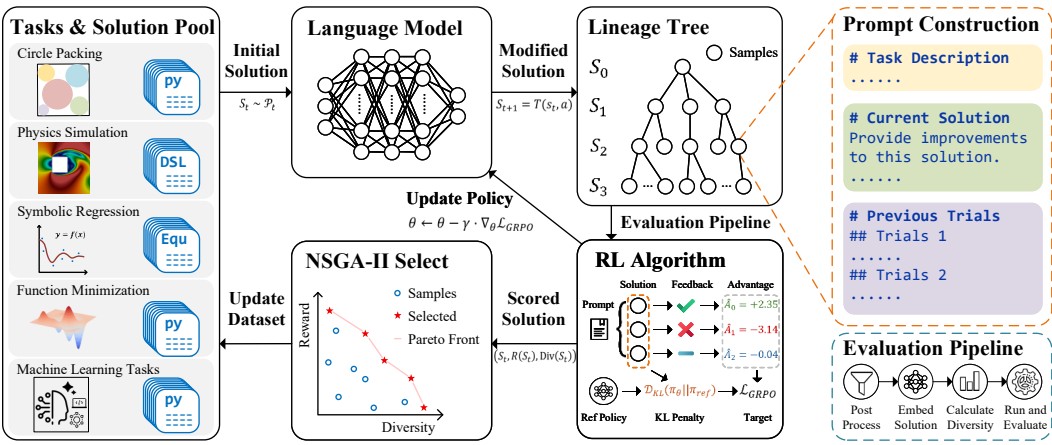

Figure 2: Illustration of HELIX framework. The workflow begins with a dataset containing task descriptions and a pool of initial solutions, which are taken by LLM as inputs. The LLM will modify and update the original solution and generate a new one, represented as descendants in lineage tree. After the evaluation pipeline, the resulting reward-labeled solutions will be used to update policy parameters via reinforcement learning. Those samples will also be selected by NSGA-II algorithm to construct promising yet diverse candidate solutions for population evolution.

To explore and search for new solutions, we use an LLM policy $\pi_\theta$ that iteratively mutates(improves) current solutions. Given timestep $t$, we sample a solution $s_t$ from $\mathcal{P}_t$, the set of candidate solutions at $t$-th step. The LLM will output an action $a_t \in \mathcal{A}$, which is an edit or modification applied to $s_t$, to obtain a new solution $s_{t+1} = T(s_t, a)$, where $T$ is the transition function. Our goal is to improve the policy's ability to find better solutions. The objective is defined as follows,

$$\max_\theta \ \mathbb{E}_{s_t \sim \mathcal{P}_t, \ a_t \sim \pi_\theta(\cdot|q,s_t)} \left[ R(s_t, a_t) \right], \tag{2}$$

where $q$ is the prompt constructed in equation 5 and $R(s_t, a_t) = R(s_{t+1})$ is the reward of the new solution with a slight abuse of notations. We leverage GRPO (Shao et al., 2024), a reinforcement learning algorithm, to update LLM policy $\pi_\theta$. By maximize the reward in equation 2, the LLM will learn to enhance current solution $s_t$ towards higher reward, which will finally leads to improvement in equation 1.

To address the exploration–exploitation trade-off and prevent entropy collapse in RL, we incorporate evolutionary algorithm in selection of candidate solutions. Suppose $\mathcal{D}_t = \{s_t\}$ is the set of all solutions generated in the $t$-th iteration and $\mathcal{D}_0 = \{s_0\}$ is the set of initial solution, the candidate solution for $t$-th step can be constructed as

$$\mathcal{P}_t = \text{SelectTop}_{\text{NSGA-II}}(\bigcup_{s=0}^{t} \mathcal{D}_s), \tag{3}$$

where NSGA-II (Deb et al., 2002) is a sample selection strategy widely adopt in evolutionary algorithms, which ensures retention of high-reward and diverse candidates. This formulation allows the model to iteratively improve its policy while exploiting previously found high-quality solutions as starting points for further exploration. Figure 2 provides a brief summary of our method and the formalized algorithm can be found in Appendix G.

## 3.2 Policy Optimization Aligned with Evolutionary Search

As the evolutionary process unfolds, updating the model parameters becomes crucial: it enables the policy to learn from both successful and failed trials, generate higher-quality solutions, and dynamically adapt to the shifting input distribution induced by the evolutionary search. Reinforcement learning is particularly suitable in this scientific setting, since open-ended scientific tasks lack standard answers and typically provide only sparse reward feedback. Motivated by the design of GRPO (Shao et al., 2024), we develop a reinforcement learning–based policy update mechanism

tailored to our framework. GRPO has proven effective in enhancing LLM reasoning on mathematical and programming tasks (Guo et al., 2025), and its multi-sample generation naturally provides diverse reasoning-driven outputs that enrich the evolutionary dataset, making it a natural inspiration for our method.

Formally, given a prompt $q$, the model will generate $G$ rollout sequences $\{a_j\}_{1 \leq j \leq G}$ with policy $\pi_{\theta_{\text{old}}}$. The GRPO objective is then defined as:

$$
\mathcal{J}_{\text{GRPO}}(\theta) = \mathbb{E}_{s_t \sim \mathcal{P}_t, \{a_j\}_{j=1}^G \sim \pi_{\theta_{\text{old}}}(\cdot|q,s_t)}
$$
$$
\left[ \frac{1}{G} \sum_{j=1}^G \frac{1}{|a_j|} \sum_{k=1}^{|a_j|} \Big( \min \big(r_{j,k}(\theta)\hat{A}_{j,k}, \text{clip}(r_{j,k}(\theta), 1-\epsilon, 1+\epsilon)\hat{A}_{j,k}\big) - \beta D_{\text{KL}}(\pi_\theta \| \pi_{\text{ref}}) \Big) \right],
$$
$$(4)$$

where $r_{j,k}(\theta) = \frac{\pi_\theta(a_{j,k}|q,a_{j,<k})}{\pi_{\theta_{\text{old}}}(a_{j,k}|q,a_{j,<k})}$ is the token-level policy ratio, $\hat{A}_{j,k} = \frac{R(s_t,a_j) - \text{mean}_j\{R(s_t,a_j)\}}{\text{std}_j\{R(s_t,a_j)\}}$ is the token-level advantage, $\epsilon$ is the clipping parameter, and $\beta$ controls the KL divergence penalty against a reference policy $\pi_{\text{ref}}$.

In order to fully leverage the in-context learning ability of LLMs, enabling the model to learn from feedback of previous trials and propose advanced solutions, we construct the prompt $q$ in the following manner:

$$
q = \text{ConstructPrompt}(\{p\} \cup \{s_t, R(s_t), F(s_t)\} \cup \{f^{(k)}(s_t), R(f^{(k)}(s_t)), F(f^{(k)}(s_t))\}_{1 \leq k < n}), \quad (5)
$$

where $p$ is the problem description, $f^{(k)}(s_t)$ is the $k$-th ancestor of $s_t$ in lineage tree (a historical trace of the solution $s_t$'s iterative refinement), $R(\cdot)$ represents the reward function and $F(\cdot)$ denotes the auxiliary feedback (e.g., textual or structured evaluations) provided by the evaluator to guide future refinements. By constructing prompts using memory of previous feedback and rewards along a lineage tree, it ensures the model effectively explores across challenging solution spaces.

## 3.3 EVOLUTIONARY MECHANISM FOR BALANCING QUALITY AND DIVERSITY

In unbounded scientific research tasks, it is crucial to explore multiple promising ideas or directions. Thus, the optimization process must balance *quality*, i.e., high-reward solutions that serve as strong starting points for refinement, with *diversity*, which sustains broad exploration across the solution space. We design the evolutionary search algorithm to be a multi-objective optimization that naturally achieves a trade-off by maintaining a population that simultaneously improves in reward and preserves diverse candidates. Specifically, we innovatively adopt NSGA-II Deb et al. (2002), which is a powerful multi-objective optimization algorithm, to filter high quality and diverse samples on the Pareto front of reward and diversity for subsequent expansion. To further encourage more diverse exploration and enable more accurate diversity computation, we propose to compute the diversity score based on its semantic embedding similarity using a pretrained language embedding model.

**Diversity measurement.** To quantify the diversity of candidate solutions, we first normalize each solution into a canonical code format and encode it into an embedding vector using a pretrained embedding model. Let $\mathcal{D} = \bigcup_{0 \leq s \leq t} \mathcal{D}_s$ represents the union of all solutions, $E(s) \in \mathbb{R}^d$ denote the embedding of solution $s \in \mathcal{D}$. For any solution $s_i$, its diversity score is computed by measuring the average similarity to its $k$ nearest neighbors in the embedding space:

$$
\text{Div}(s_i) = 1 - \frac{1}{k} \sum_{j \in \mathcal{N}_k(i)} \frac{E(s_i) \cdot E(s_j)}{\|E(s_i)\| \|E(s_j)\|}, \quad (6)
$$

where $\mathcal{N}_k(i)$ denotes the indices of the $k$ nearest neighbors of $s_i$ in $\mathcal{D}$, measured by cosine similarity. A higher $\text{Div}(s_i)$ indicates that $s_i$ is more distinct from other solutions, thereby contributing to population diversity.

**NSGA-II based selection.** Given both reward score $R(s)$ and diversity score $\text{Div}(s)$, each candidate solution can be mapped to a two-dimensional objective space. We then adopt the NSGA-II (Deb et al., 2002) algorithm to select high-quality and diverse samples. NSGA-II first applies a

nondominated sorting procedure to partition solutions into multiple fronts based on Pareto dominance, where a solution $s_a$ dominates $s_b$ if $R(s_a) \geq R(s_b)$ and $\text{Div}(s_a) \geq \text{Div}(s_b)$ with at least one strict inequality. To further ensure diversity preservation within each front, NSGA-II computes a crowding-distance measure and selects representative samples that are well spread in the objective space.

By combining nondominated sorting with diversity preservation, the resulting population $\mathcal{P}$ retains candidates that are both high-reward and diverse. This mechanism allows the model to continuously exploit promising solutions while sustaining exploration across multiple distinct solution trajectories.

## 4 EXPERIMENT

In this section, we first introduce the experimental setup, including the tasks we selected for benchmarking the model's ability to solve open-ended scientific problems. Then, we present extensive experiments demonstrating that HELIX effectively enhances model capability, integrates historical experience, and balances reward with diversity, leading to significant improvements over existing baselines in solving unbounded and open-ended scientific challenges. Finally, the ablation studies reveal how different components of the framework work together in a complementary manner.

### 4.1 EXPERIMENT SETTING

**Tasks.** To comprehensively evaluate the model's capacity for complex scientific reasoning, we design experiments on five representative categories of tasks. These tasks are particularly suited for our study because they are *unbounded*, lacking a guaranteed global optimum, *open-ended*, requiring exploration over vast and flexible solution spaces and *domain-specific*, containing unique constraints and complex background. Success in these tasks not only demonstrates the model's ability to search beyond local optima, but also provides insights that can inspire solutions in broader scientific and engineering domains.

1. **Machine Learning Tasks.** We selected three representative datasets: Adult income (Becker & Kohavi, 1996), Bank marketing (Moro et al., 2014) and Boston housing (Harrison Jr & Rubinfeld, 1978) dataset to evaluate the model's ability to solve machine learning tasks. These tasks reflects the open-ended challenge of combining ML algorithms for novel applications, with potential implications for autonomous scientific workflows.

2. **Physics Simulation Tasks.** These tasks combine geometric structures design and optimization in multi-physics environments in distinct fields. The design space of these problems has a very high degree of freedom with few global optimal solution.

3. **Circle Packing Problems.** The objective of these tasks is to maximize the sum of radii of circles packed within given shapes. It allows multiple feasible arrangements and there is no proved global optimum solution currently.

4. **Function Minimization.** It requires LLM to write a code to find the global minimum point of given functions. Agents can search freely for new mathematical optimization methods in code space.

5. **Symbolic Regression.** A benchmark (Shojaee et al., 2025) evaluates the ability of LLMs to hypothesize underlying expressions for noisy data. The model needs to search among a vast possible expression set and utilize domain specific knowledge to find solution.

**Models.** We selected the DeepSeek-R1-Distill-Qwen model family for our experiment due to its strong reasoning capabilities and manageable size, which is critical for performing complex scientific tasks under computational constraints. Among the model family, the 14B version offers an optimal balance between efficiency and performance, and was selected as the model in the main results. For physics simulation tasks that require strong geometric reasoning ability and physical prior knowledge, we utilize the 32B version of the model.

**Baselines.** We compare our approach against three key baselines:

Table 1: Results of main experiments. All values correspond to the best outcome obtained across all attempts. We use ↑ to indicate that larger values correspond to better performance, and ↓ represents the opposite. We highlighted the best results in each task in **bold**. "NA" denotes non-convergence or unsuitability for given case.

| | | Task Specific Methods | | Direct Prompt | | Open Evolve | | Ours |
|---|---|---|---|---|---|---|---|---|
| | **Tasks** | LightGBM | RRL | Qwen | GPT-4o | Qwen | GPT-4o | - |
| Machine Learning | Adult Income ↑ | 80.36 | 80.72 | 73.72 | 76.91 | 76.90 | 72.27 | **82.07** |
| | Bank Marketing ↑ | 75.28 | 76.32 | 0.00 | 76.91 | 75.66 | 78.54 | **80.65** |
| | Boston Housing ↓ | 3.258 | 3.966 | 3.149 | 3.031 | 2.937 | 2.937 | **1.747** |
| | Transparent Conductors ↓ | 0.060 | NA | 0.060 | 0.059 | 0.059 | 0.056 | **0.049** |
| | **Tasks** | Parameter Scan | Topology Opt | Qwen | GPT-4o | Qwen | GPT-4o | - |
| Physics Simulation | Inductor ↑ | 6.111 | 6.248 | 2.584 | 0.001 | 1.637 | 1.652 | **9.609** |
| | Beam Bending ↑ | 4.771 | NA | 5.407 | 4.005 | 10.793 | 6.352 | **17.298** |
| | Magnetic Torque ↑ | 10.273 | NA | 0.323 | 1.201 | 3.488 | 1.607 | **11.045** |
| | Periodic Heat ↑ | 1.206 | NA | 1.258 | 1.255 | 1.233 | 1.266 | **1.278** |
| | Demultiplexer ↑ | 18.322 | **23.555** | 3.364 | 4.532 | 12.341 | 8.645 | 14.260 |
| | **Tasks** | SLSQP | Genetic Algo | Qwen | GPT-4o | Qwen | GPT-4o | - |
| Circle Packing | Packing in Unit Square ↑ | 2.519 | 2.345 | 1.673 | 1.900 | 1.586 | 2.611 | **2.636** |
| | Packing in Unit Disk ↑ | 4.522 | 3.896 | 4.608 | 3.290 | 4.604 | 3.984 | **4.664** |
| | **Tasks** | SLSQP | Trust-constr | Qwen | GPT-4o | Qwen | GPT-4o | - |
| Function Minimization | Eggholder ↑ | 0.705 | 0.688 | **1.000** | 0.959 | **1.000** | **1.000** | **1.000** |
| | Mishras Bird ↑ | 0.814 | 0.764 | **1.000** | 0.996 | **1.000** | **1.000** | **1.000** |
| | Keanes Bump 10d ↑ | 0.714 | 0.692 | 0.886 | 0.987 | **1.000** | 0.997 | **1.000** |
| | Keanes Bump 20d ↑ | 0.603 | NA | 0.794 | 0.657 | 0.596 | 0.983 | **1.000** |
| | Keanes Bump 30d ↑ | 0.594 | NA | 0.923 | 0.625 | 0.677 | 0.668 | **0.994** |
| | **Tasks** | LLM-SR | LaSR | Qwen | GPT-4o | Qwen | GPT-4o | - |
| Symbolic Regression | Chemistry ↓ | 4.12e-6 | 9.11e-5 | 2.66e-5 | **2.44e-6** | 1.59e-5 | 9.52e-6 | 7.32e-6 |
| | Biology ↓ | 3.06e-6 | 1.53e-4 | 1.26e-4 | 7.52e-5 | 1.64e-4 | 5.31e-5 | **2.98e-8** |
| | Physics ↓ | 7.62e-5 | 9.94e-4 | 2.71e-4 | 1.13e-4 | 2.76e-5 | 1.22e-4 | **2.76e-5** |
| | Material Science ↓ | **3.21e-9** | 9.23e-6 | 7.14e-6 | 1.85e-6 | 6.99e-7 | 1.94e-6 | 4.46e-6 |

1. **Direct Prompt (Test-Time Scaling)**: Queries the model directly and selects the best outcome from multiple samples to establish a performance upper bound of base model.

2. **Open Evolve** (Sharma, 2025): An open-source implementation of the AlphaEvolve (Novikov et al., 2025) framework, which uses an evolutionary algorithm with multiple LLM roles (e.g., proposing code mutations, evaluating fitness) to iteratively generate, test, and evolve code or solutions across generations.

3. **Task-Specific Methods**: Represents results from established algorithms designed for each specific problem. Details of these methods can be found in Appendix C.

## 4.2 MAIN RESULTS

Table 1 presents the results of our methods compared to various baselines. The best results in each task are highlighted in **bold**. Since we selected multiple heterogeneous tasks, their evaluation metrics are not the same. The detailed definitions and specific evaluation criteria are deferred to Appendix B.

Across the 20 benchmark tasks, our method achieves the best performance on **17** tasks, surpassing all competing baselines. Compared under the same model settings, our framework consistently outperforms Direct Prompting across **all** benchmarks. Against OpenEvolve—the open-source version of AlphaEvolve—it achieves superior results on **19** tasks. These results clearly highlight the strength of our framework in solving open-ended scientific problems among various domains compared to other approaches.

Notably, we observe that the base Qwen models perform relatively poorly on certain tasks such as Bank Marketing and Magnetic Torque, exhibiting low rewards even in the best of 64 direct trials. However, our framework significantly improves performance in these cases by leveraging parameter updates and in-context learning to effectively incorporate feedback from the exploration process.

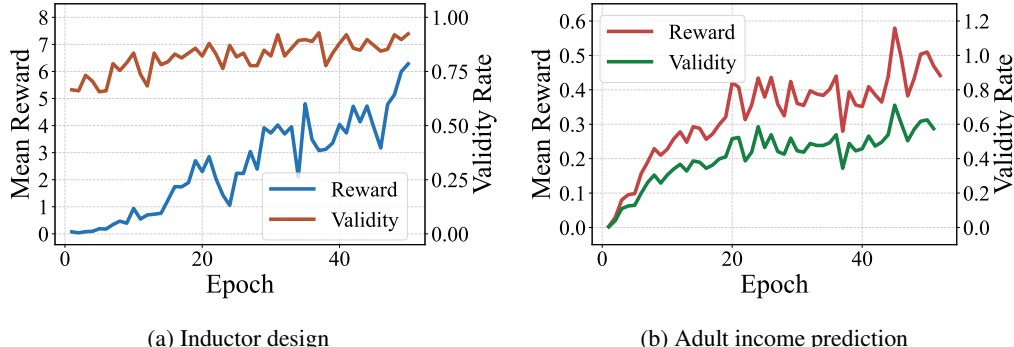

(a) Inductor design          (b) Adult income prediction

Figure 3: Convergence analysis on the Inductor and Adult tasks. The curves show the progressive improvement of average reward and validity during training, demonstrating that our framework effectively leverages reinforcement learning feedback and evolutionary dynamics to produce increasingly valid and high-quality solutions.

This demonstrates that our approach can partially overcome the limitations of weaker base models by iteratively evolving toward superior solutions.

To further assess the competitiveness of our approach against state-of-the-art scientific discovery systems, we compared it with GPT-4o, one of the most advanced closed-source models. Remarkably, our method outperforms GPT-4o on **18** tasks, regardless of whether GPT-4o is equipped with multi-role collaborative reasoning frameworks. These results highlight that our framework can fully exploit the prior knowledge of smaller models through reinforcement learning, enabling cost-efficient and effective solutions to complex scientific problems.

In comparison with task-specific methods, which are typically crafted by human experts for particular domains, our framework still achieves superior performance on **17** tasks. Specifically, in the circle packing task, we establish a new world record 2.63598308 using only a 14B model. For the Transparent Conductors dataset, derived from a human-participation competition (Ziletti et al., 2017), our framework attains the second-highest score on the participants' leaderboard. This highlights its ability to iteratively evolve within open-ended solution spaces and to autonomously uncover novel solutions that go beyond manually designed approaches.

To provide further evidence that our framework effectively integrates reinforcement learning and evolutionary algorithms, we analyze its convergence behavior on two representative cases: inductor design and adult income prediction. Figure 3 plots the average reward and validity of model outputs during training. Both metrics exhibit a clear upward trend: the validity rate rises steadily, showing that the model increasingly generates outputs that satisfy task constraints, while the average reward improves, reflecting higher-quality solutions. This dual improvement demonstrates that reinforcement learning progressively strengthens the model's intrinsic reasoning ability. It also indicates that the quality of the evolving population keeps improving, enabling the model to leverage in-context feedback as well as intuitions from high-reward solutions to generate better outputs.

### 4.3 ABLATION STUDY

#### 4.3.1 EFFECTIVENESS OF FRAMEWORK COMPONENTS

To better understand the contribution of each component in our framework, we conduct ablation studies on the Boston Housing and Circle Packing tasks. We design several controlled variants by selectively disabling or simplifying parts of the algorithm: **TopScore**, where only the highest-reward candidate in the dataset is selected for further evolution; **TopDiv**, where selection relies solely on diversity without considering reward; **Random**, where candidates are sampled randomly from the population; **EvoOnly**, where the model parameters are kept fixed and only the evolutionary pipeline is applied; and **TrainOnly**, which removes the evolutionary mechanism and in-context prompting, reducing the framework to pure GRPO reinforcement learning. These variants allow us to

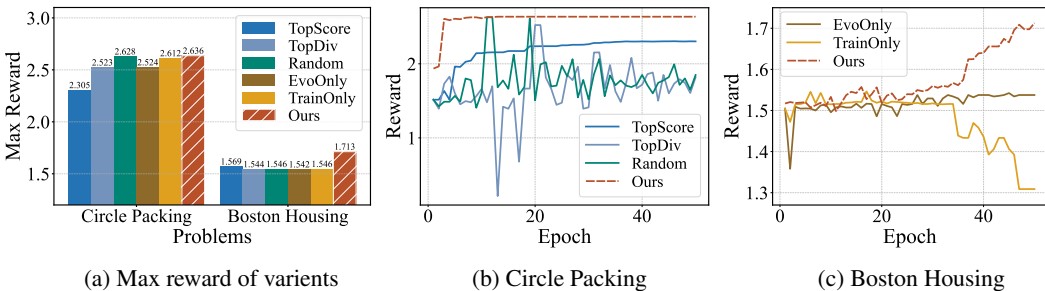

(a) Max reward of varients     (b) Circle Packing     (c) Boston Housing

Figure 4: Ablation analysis of framework components. **(a):** Maximum reward achieved by different ablation variants. **(b):** Curve of epoch-wise maximum reward on the Circle Packing task, highlighting the critical role of balancing diversity and quality for stable optimization. **(c):** Curve of epoch-wise maximum reward on the Boston Housing task, showing the necessity of combining reinforcement learning with evolutionary guidance.

disentangle the relative importance of reward-driven selection, diversity maintenance, evolutionary population updates, and reinforcement learning in driving overall performance.

Figure 4a reports the maximum reward achieved under different ablation settings. Across both tasks, all variants perform worse than our full framework, confirming the necessity of each component. We next analyze the results task by task.

For the **Circle Packing** problem, high-quality solutions rely on diverse initial starting points for optimization algorithms. As shown in Figure 4b, eliminating diversity (TopScore) significantly reduces reward, since the search quickly collapses into narrow solution modes. In contrast, Random and TopDiv maintain higher diversity, enabling the model to extend from a richer set of initial states. However, focusing solely on diversity also leads to instability—visible in the large variance of TopDiv and Random—whereas TopScore and our full method (Ours) remain relatively stable. This instability disrupts training and prevents the model from finding strong solutions in later epochs. Moreover, we conducted a detailed analysis on the performance gap between OpenEvolve and HELIX in Appendix E, which demonstrate the effectiveness of explicitly combining diversity with embedding model in our framework. These results highlight that balancing diversity and solution quality is critical for solving such problems.

For the **Boston Housing** task, strong performance requires careful parameter tuning and complex feature engineering, which typically emerge from iteratively learning from past experience. As shown in Figure 4c, disabling either reinforcement learning or evolution severely limits performance. With EvoOnly, the model remains bounded by its initial capacity and fails to break through training bottlenecks. Conversely, with TrainOnly, the model cannot effectively accumulate knowledge in context and collapses during training. These results demonstrate that both parameter updates and in-context evolutionary guidance are indispensable for helping the model accumulate expertise and progressively refine its solutions.

### 4.3.2 SCALING EXPERIMENTS

Here, we discuss the impact of base model size on task performance. We evaluate our framework on two representative tasks, Magnetic Torque Maximization and Inductor Design, using the DeepSeek-R1-Distill-Qwen model family with 1.5B, 7B, 14B, and 32B parameters. As shown in Figure 5, for the magnetic torque task, the reward steadily increases with model size, indicating stronger reasoning ability and more effective exploration. For the inductor design task, we observe a reward plateau around 9.6. However, the mean reward continues to grow as model size increases, suggesting that larger models generate more valid and higher-quality candidates. These results demonstrate that our framework exhibits scaling property: as the underlying LLM grows, the system can push the boundaries of scientific discovery by enabling more efficient and higher-quality exploration.

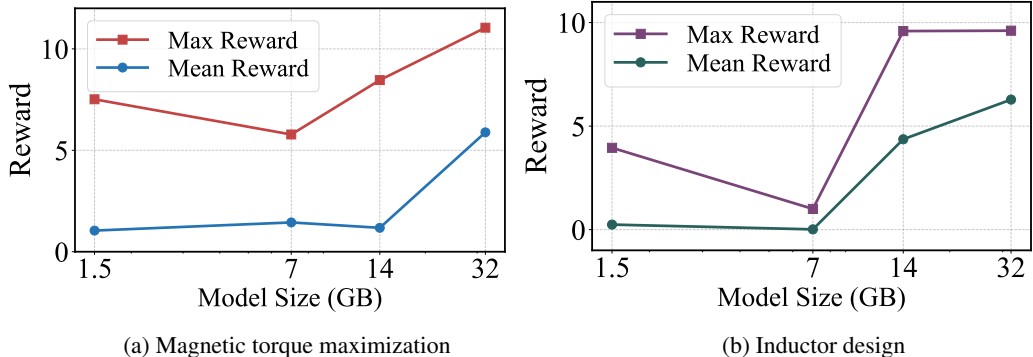

(a) Magnetic torque maximization                    (b) Inductor design

Figure 5: Scaling analysis of model parameter scale on **(a):** magnetic torque maximization and **(b):** inductor design tasks.

## 5  CONCLUSION

In this work, we proposed HELIX, a hierarchical evolutionary reinforcement learning framework with in-context experiences. By integrating reinforcement learning, evolutionary selection, and in-context trial incorporation, HELIX effectively balances exploration and exploitation, enables task-specific adaptation, and iteratively refines solutions. Extensive experiments across 20 tasks in five diverse categories demonstrate that HELIX consistently outperforms strong task-specific baselines and advanced proprietary models. Overall, HELIX shows strong potential for advancing open-ended scientific discovery by enabling iterative, diversity-aware exploration. Looking ahead, it could provide a foundation for broader applications in engineering, optimization, and autonomous research systems.

## ETHICS STATEMENT

This work focuses on developing a hybrid reinforcement learning and evolutionary framework for solving complex scientific problems. It does not involve human subjects, sensitive personal data, or proprietary datasets, and thus raises no direct ethical or privacy concerns. All datasets used are publicly available and widely adopted in prior research. All authors have reviewed and agree to abide by the ICLR Code of Ethics as linked above, and affirm that this submission complies with the principles of honesty, transparency, fairness, and responsible conduct.

## REPRODUCIBILITY STATEMENT

Detailed descriptions of the experimental setup, task definitions, and evaluation metrics are provided in Appendix A and Appendix B.

Source code will be available at `https://github.com/EdwardIX/HELIX`.

## ACKNOWLEDGEMENT

This work was supported by Fundamental and Interdisciplinary Disciplines Breakthrough Plan of the Ministry of Education of China (No. JYB2025XDXM101), NSF of China Projects (Nos. 62550004U25B6003, 92370124, 92248303); Beijing Natural Science Foundation L247011; the High Performance Computing Center, Tsinghua University. J.Z was also supported by the XPlorer Prize.

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

## A  TRAINING AND EVALUATION DETAILS

**Training.**  We primarily use DeepSeek-Distill-Qwen-14B and 32B as the backbone models in our experiments. The models are fine-tuned with the VERL framework (Sheng et al., 2024) under the GRPO algorithm. Each model is trained for 80 epochs with a fixed learning rate of $1 \times 10^{-6}$, updating all parameters. We set the KL coefficient in GRPO to $1 \times 10^{-3}$ and the number of rollouts to 16. The rollouts are generated via VLLM (Kwon et al., 2023) backend with temperature equals to $1.0$ and top_p equals to $0.95$. Training was conducted using eight A100 GPUs for 14B models and sixteen H100 GPUs for 32B models. For training efficiency, we use Pytorch FSDP (Zhao et al., 2023) with parameter offload and optimizer offload. Gradient checkpoint and Flash-Attention (Dao, 2024) are used by default.

**Evaluation.**  The evaluation is performed on a Slurm Workload Manager system. For each job, we allocate 4 Intel(R) Xeon(R) Platinum 8168 CPUs for execution and impose time limits for each task: five minutes for physics simulation, two minutes for machine learning and function minimization, and one minute for circle packing and symbolic regression. The execution time includes the time for task-dependent evaluators to calculate reward. For the detailed evaluate metric and reward calculation, please refer to Appendix B.

## B  DEFINITION AND EVALUATION OF PROBLEMS

In this section we explain the detailed problem definition and evaluation metrics of all the tasks used in the experiment.

### B.1  MACHINE LEARNING

We selected 3 classic machine learning datasets, and the model has to write Python code to maximize the F1 score for classification tasks and minimize the rooted mean square error (RMSE) for regression tasks. The details are described below.

#### B.1.1  ADULT INCOME

The Adult income dataset (Becker & Kohavi, 1996) is a well-known binary classification task. The goal is to predict whether a person's income exceeds $50,000 per year based on various demographic features such as age, education, marital status, and occupation. The dataset is sourced from the 1994 U.S. Census and contains both categorical and numerical features, with some missing values.

The dataset itself contains a separate train and test split. We then load the train set for model's training and evaluate its result on the test set. The reward is the Macro F1 score, defined as:

$$R = \frac{1}{C} \sum_{c=1}^{C} \frac{2 \cdot P_c \cdot R_c}{P_c + R_c} \tag{7}$$

where $P_c, R_c$ are the precision and recall for class $c$, and $C = 2$ is the total number of classes.

#### B.1.2  BANK MARKETING

The Bank marketing dataset (Moro et al., 2014) is another binary classification problem. It includes data from a Portuguese bank's direct marketing campaigns, where the objective is to predict whether a client will subscribe to a term deposit. This dataset is characterized by a high number of categorical features and a significant class imbalance, making it a good benchmark for evaluating model performance under challenging real-world conditions.

To ensure a robust evaluation, we use a 5-fold cross-validation strategy with StratifiedKFold in sklearn to handle the class imbalance. The data is randomly split into five folds, maintaining the same class distribution in each fold as in the original dataset. The model is trained and evaluated five times, with each fold serving as the test set once. The final reward is the average of the Macro F1 scores obtained from all five folds. If a task fails to produce a result in any fold, its reward is considered to be 0 for that fold. The final result is the Macro F1 score, as defined in equation 7.

### B.1.3 BOSTON HOUSING

The Boston housing dataset (Harrison Jr & Rubinfeld, 1978) is a classic regression problem. The task is to predict the median value of owner-occupied homes in Boston suburbs, based on 13 features. These features include per capita crime rate, a number of rooms per dwelling, and the proportion of non-retail business acres. While the original dataset is no longer widely used for research due to ethical concerns, it remains a common benchmark for teaching and evaluating regression models.

To evaluate model performance, we use a 5-fold cross-validation strategy with KFold, splitting the data into five folds. The model is trained and evaluated five times, with each fold serving as the test set once. The final reward for this task is the average of the scores from all five folds. The reward is calculated using the following formula:

$$R = 2 - \log_{10}(\text{RMSE} + 10^{-10}), \tag{8}$$

where:

$$\text{RMSE} = \sqrt{\frac{1}{N} \sum_{i=1}^{N} (y_i - \hat{y}_i)^2} \tag{9}$$

This reward metric is designed to penalize larger RMSE values while rewarding smaller ones. If a task fails in any fold, its reward is considered to be 0 for that fold.

### B.1.4 TRANSPARENT CONDUCTORS

The Transparent Conductors dataset (Ziletti et al., 2017) is motivated by the need for accelerated discovery of materials that simultaneously exhibit optical transparency and electrical conductivity—two properties that are typically at odds. Such materials are central to modern technologies including photovoltaic cells, LEDs, sensors, touch screens, and display panels. Despite their importance, only a limited number of compounds are currently known to meet the desired transparency–conductivity trade-off, making data-driven exploration an appealing alternative to costly experimental or quantum-mechanical searches.

The dataset contains computationally derived information for 3,000 candidate materials belonging to the sesquioxide alloy family $(Al_x Ga_y In_z)_{2N} O_{3N}$, where the compositional ratios satisfy $x + y + z = 1$ and the total number of atoms in the unit cell ranges from 5 to 100. These materials are of particular interest due to their large bandgaps, chemical stability, and relatively low production cost. Each entry includes crystallographic descriptors (e.g., space group, lattice parameters), compositional ratios, and structural characteristics, offering a rich feature space for modeling.

The task is to predict two key target properties for each material: (1) formation energy, which reflects thermodynamic stability, and (2) bandgap energy, which determines visible-range transparency. Accurate prediction of these quantities enables efficient screening of new transparent conductor candidates without the need for expensive density-functional theory (DFT) calculations.

Model performance is evaluated using the root mean squared logarithmic error (RMSLE), computed column-wise for the two target properties. For a single target, the RMSLE is defined as:

$$\text{RMSLE} = \sqrt{\frac{1}{n} \sum_{i=1}^{n} (\log(p_i + 1) - \log(a_i + 1))^2}, \tag{10}$$

where $n$ is the number of samples, $p_i$ denotes the predicted value, and $a_i$ the ground-truth value. The final reward for model training is:

$$R = 1 - \text{RMSLE} \tag{11}$$

### B.2 PHYSICS SIMULATION

To test the model's capacity for geometric reasoning and ability to utilize physics prior knowledge to discover better designs, we proposed the following physics simulation tasks. These tasks mainly require the model to generate a yaml representation of a complex geometry under certain constraints to maximize the reward. We utilize COMSOL Multiphysics® (COMSOL AB, 2024), a commercial FEA software for industrial multiphysics simulations, for the evaluation backend.

### B.2.1 ACOUSTIC DEMULTIPLEXER

This task aims to design an acoustic demultiplexer. The demultiplexer is a data distributing device which takes acoustic energy from the input port and distributes different frequency bands to the specific output port. The model is asked to propose the cavity geometry within a circular domain as seen in Fig. 6 to maximize the acoustic pressure at output port 2 while minimizing the pressure at output port 3. The input acoustic pressure level is set to 1 Pa at port 1, and the frequency level is set to 7500 Hz.

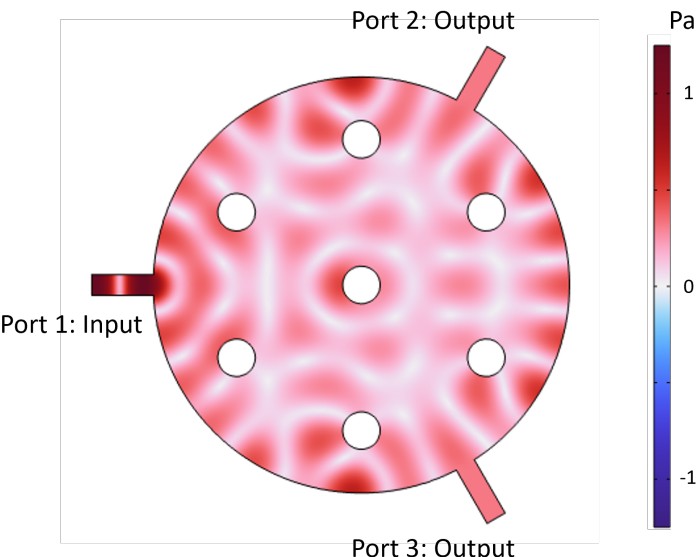

Figure 6: The RMS pressure field of an acoustic demultiplexer at frequency level 7500 Hz. The RMS pressure field in log scale is proportional to the acoustic power.

The model is guided by the following reward $R$ where $P_i$ is the power output at port $i$, $p_{rms}$ is the Root Mean Square (RMS) pressure field, $\rho$ is fluid density, and $c$ is sound speed.

$$P = \int_{port} \frac{p_{rms}^2}{\rho c} dl$$
$$R = \frac{log_{10}(P_2) - log_{10}(P_3)}{0.292} \tag{12}$$

We use a value of 0.292 on the denominator of Eq. 12 to normalize the reward. And Fig. 6 shows a symmetric design with 7 circular cavities in the computation domain, producing equal acoustic pressure at the two output ports and thus $R = 0$. Notice that LLM is not limited by the circular cavity pattern, and is prompted to freely explore any viable cavity geometries within the computation domain.

### B.2.2 MAGNETIC TORQUE

This task aims to design the geometry of an iron core that generates large torque when subjected to a uniform magnetic field. Fig. 7 shows the problem setting, an example iron core geometry and the corresponding magnetic flux density norm field. A uniform magnetic field intensity of $\mathbf{H} = [0, 1e5]$ A/m is applied to the circular boundary. The iron core possesses a large permeability $\mu \gg \mu_0$ distorts the magnetic flux density field $\mathbf{B}$ within the circular air domain. The distorted $\mathbf{B}$ thus applies a torque on the iron core, which can be obtained from Comsol by solving the static Maxwell's equations.

To guide the model reinforcement learning and evolutionary search, the following reward $R$ is computed as below where $\mathbf{T}$ is Maxwell stress tensor, $\mathbf{r}$ is position vector, and $\boldsymbol{\tau}$ represents magnetic torque which is simplified to $\tau_z$ in 2D simulations:

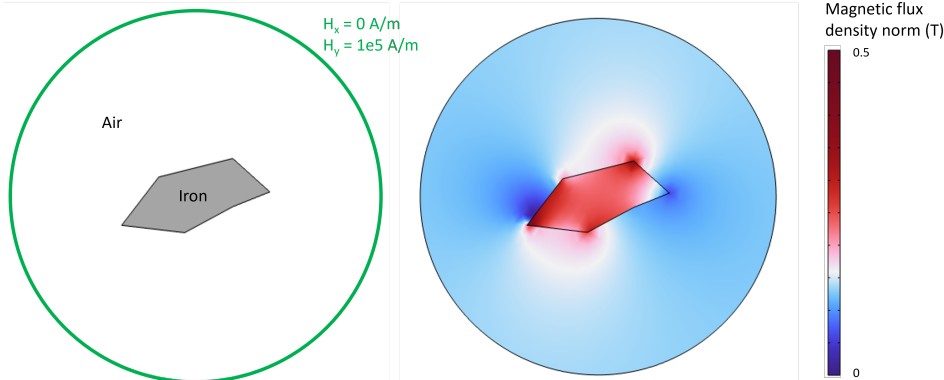

Figure 7: The magnetic flux density field generated by an iron core subject to a uniform magnetic field boundary condition. The distorted magnetic flux density field then applies a torque on the iron core.

$$\mathbf{T} = \frac{1}{\mu_0}(\mathbf{B}\mathbf{B} - \frac{1}{2}B^2\mathbf{I})$$

$$R = \frac{||\boldsymbol{\tau}||}{9241.99 \cdot A} = \frac{1}{9241.99 \cdot A}||\int_S \mathbf{r} \times (\mathbf{T} \cdot \hat{\mathbf{n}})dA||$$

(13)

We use a value of $9241.99$ on the denominator of Eq. 13 to normalize the reward. Notice that a perfectly symmetric iron core (for instance a circle) would have $\tau_z = 0$. Therefore, we expect to train and evolve the LLM to produce a highly irregular iron core geometry to generate large magnetic torque values. We set a minimum area of $2e^{-4}$ m$^2$ to avoid naive designs.

### B.2.3 BEAM BENDING

This task aims to design the cross section geometry of a cantilever beam subject to a superposed loading of bending moments $M_x$ and $M_y$, shear forces $T_x$ and $T_y$ along the two in-plane directions, and twisting moment $T_z$ along the out-of-plane direction. The cantilever beam is assumed to be linear elastic with Young's modulus 1 GPa and Poisson's ratio 0.3. Fig. 8 shows an example beam cross section design and the von Mises stress distribution as calculated from Eq. 14, solved using the Beam Cross Section module in Comsol. As the cross section stays in the x-y plane, $\sigma_{xx}$, $\sigma_{yy}$, and $\tau_{xy}$ take 0 values.

$$\sigma_{vm} = \sqrt{\frac{1}{2}[(\sigma_{xx} - \sigma_{yy})^2 + (\sigma_{xx} - \sigma_{zz})^2 + (\sigma_{yy} - \sigma_{zz})^2] + 3 \cdot (\tau_{xy}^2 + \tau_{xz}^2 + \tau_{yz}^2)}$$

(14)

The reward is set to be $R = \frac{I_1^{0.8} \cdot I_2^{0.2}}{1.32e^{-3} \cdot A}$ where $A$ is the cross section area, $I_1$ is the largest second moment of inertia, $I_2$ is the smallest second moment of inertia. We use a value of $1.32e^{-3}$ on the denominator to normalize the reward. $I_1$ and $I_2$ represent the beam's largest and smallest resistance over different bending loading directions, and can be calculated from the stress field following the classical beam bending theory (Bauchau & Craig, 2009). We set a minimum area of $2e^{-3}$ m$^2$ to avoid naive designs.

### B.2.4 PERIODIC HEAT

This task aims to design the unit cell geometry of a periodic meta-material for best effective thermal conductivity. The base material is assumed to be aluminum with density 2700 kg/m$^3$ and thermal conductivity 238 W/mK. Fig. 9 shows an example 2D unit cell geometry which will be extruded in the z direction to form the 3D unit cell. The resultant temperature distribution and effective

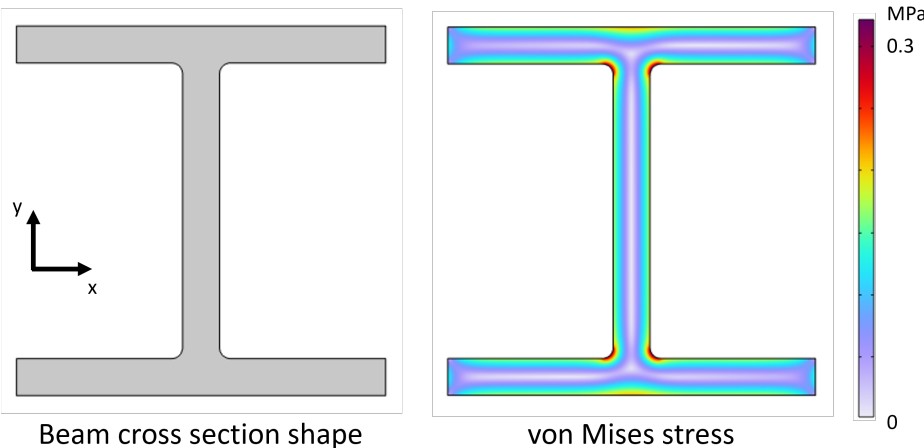

Figure 8: The von Mises stress field generated by applying bending moment, shear force, and twisting moment on a cantilever beam cross section design.

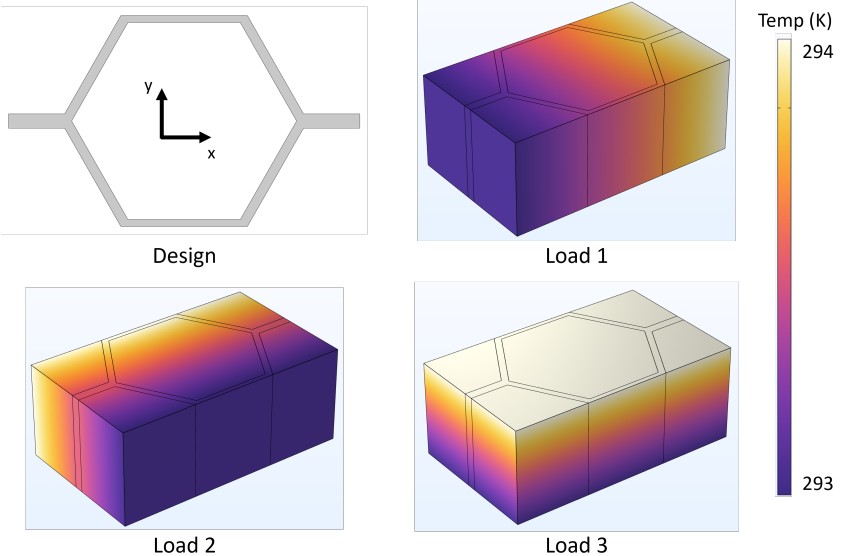

Figure 9: Temperature distribution of the meta-material under three loading conditions. The effective properties are calculated based on temperature distributions according to the homogenization theory.

properties are solved using Comsol based on the homogenization theory. The results are calculated from a 1 K temperature difference boundary conditions along x, y, and z directions.

$$R = \frac{trace(\mathbf{k}_{eff})}{0.178 \cdot \rho_{eff}} \tag{15}$$

where $\mathbf{k}_{eff}$ is the homogenized effective thermal conductivity matrix, and $\rho_{eff}$ is the effective density, which simply equals to the percentage of volume filled by aluminum. We use a value of 0.178 on the denominator to normalize the reward. This objective function targets to maximize the thermal conductivity along x, y, and z directions under limited material usage. We set a maximum effective density $\rho_{eff} \leq 2000$ kg/m$^3$ to avoid naive designs.

### B.2.5 INDUCTOR

This task aims to design an inductor which is a critical component in power electronics. Fig. 10 shows an example inductor consisting of an iron core and coil windings in a cylindrical coordinate. A sinusoidal current excitation is supplied to the coils at a frequency of 1000 Hz and magnitude 500 A. The iron core possesses a nonlinear magnetization curve with an initial permeability of 663 H/m and saturates at 5 T. The resultant magnetic field is calculated using Comsol by solving the Maxwell's equations in frequency domain. The model is asked to propose the optimal iron core geometry as well as the placement of the coil windings (coil shapes are fixed) to produce the maximum inductance with limited material usage.

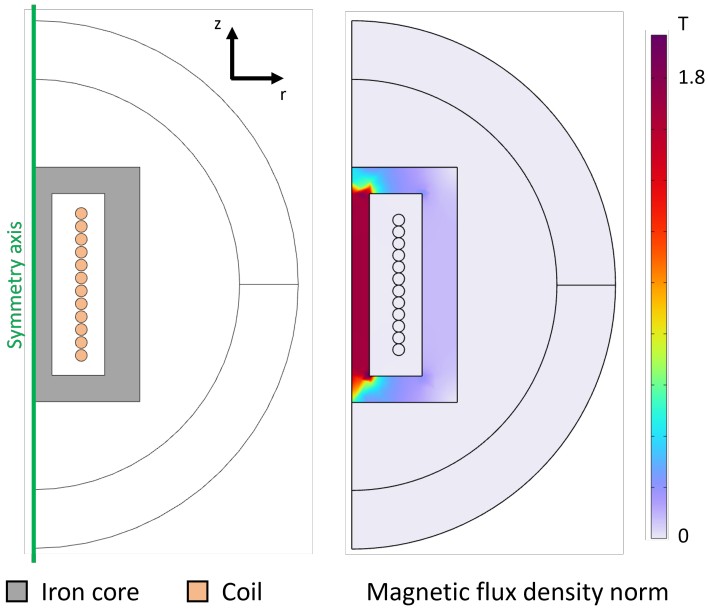

Figure 10: The magnetic flux density norm field generated by an inductor. The copper coils are excited by a 500 A, 1000 Hz sinusoidal current.

$$R = \frac{L}{43.11 \cdot V} = \frac{0.5 \cdot \int_{\Omega}(B_r \cdot \overline{H_r} + B_\phi \cdot \overline{H_\phi} + B_z \cdot \overline{H_z})dV}{43.11 \cdot V} \tag{16}$$

The reward calculation is shown in Eq. 16 where $B_r$, $B_\phi$, and $B_z$ are cylindrical components of magnetic flux density field, and $H_r$, $H_\phi$, $H_z$ are components of magnetic intensity field. Both fields take complex values for frequency domain response. We use a value of 43.11 on the denominator to normalize the reward. The numerator stands for the inductance which is a volume integral of magnetic energy. We set a minimum iron core volume of $1e^{-3}$ m$^3$ to avoid naive designs.

### B.3 CIRCLE PACKING

The objective of these tasks is to pack a fixed number of circles in a specific domain and maximize the sum of the radii of these circles. The circles cannot overlap with each other or exceed the domain boundary. All the centers and radii can change as long as the constraints are satisfied.

Formally, let $n = 26$ be the number of circles, $\{x_i\}_{i \leq n}$, $\{y_i\}_{i \leq n}$ be the coordinates of centers and $\{r_i\}_{i \leq n}$ be the radii. The objective can be written as:

$$R = \sum_{i=1}^{n} r_i, \tag{17}$$

while the constraint is

$$
\begin{aligned}
\sqrt{(x_i - x_j)^2 + (y_i - y_j)^2} &\geq r_i + r_j, && \forall 1 \leq i < j \leq n \\
x_i - r_i &\geq 0, && \forall 1 \leq i \leq n \\
x_i + r_i &\leq 1, && \forall 1 \leq i \leq n \\
y_i - r_i &\geq 0, && \forall 1 \leq i \leq n \\
y_i + r_i &\leq 1, && \forall 1 \leq i \leq n,
\end{aligned}
\tag{18}
$$

for the packing in a unit square, and

$$
\begin{aligned}
\sqrt{(x_i - x_j)^2 + (y_i - y_j)^2} &\geq r_i + r_j, && \forall 1 \leq i < j \leq n \\
\sqrt{x_i^2 + y_i^2} + r_i &\leq 1, && \forall 1 \leq i \leq n,
\end{aligned}
\tag{19}
$$

for the packing in a unit disk.

### B.4 FUNCTION MINIMIZATION

These tasks require the model to find an effective algorithm to locate the global minimum of a complex function with various local minima. For a given function $f(\mathbf{x}^*)$ and the model's prediction $\hat{\mathbf{x}}^*$, The evaluation metric is defined as:

$$
R = \frac{|f(\mathbf{x}^*)|}{|f(\mathbf{x}^*)| + |f(\hat{\mathbf{x}}^*) - f(\mathbf{x}^*)|}.
\tag{20}
$$

This metric is suitable for distinct functions with varying scales of $|f(\mathbf{x}^*)|$. It satisfies $0 \leq R \leq 1$ and if the model successfully finds the global minimum, the reward will be $R = 1.0$.

#### B.4.1 EGGHOLDER FUNCTION

The Eggholder function is a classical task for evaluating evolutionary optimization algorithms with various local minima. It can be defined as:

$$
f(\mathbf{x}) = -(x_2 + 47)\sin\left(\sqrt{\left|(x_2 + 47) + \frac{x_1}{2}\right|}\right) - x_1 \sin\left(\sqrt{|x_1 - (x_2 + 47)|}\right),
\tag{21}
$$

with constraint $-512 \leq x_1, x_2 \leq 512$ and a global minimum $f((512, 404.2319)) \approx -959.6407$ under such constraint.

Figure 11 illustrates the landscape and the global minimum point of the Eggholder function.

#### B.4.2 MISHRA'S BIRD FUNCTION

The Mishra's Bird function is a classic test function used in optimization to evaluate the performance of algorithms. It is known for having a unique "bird-shaped" landscape with multiple local minima and a single global minimum. It's often used to test an algorithm's ability to avoid getting stuck in suboptimal solutions.

The function is defined as:

$$
f(\mathbf{x}) = \sin(x_2)e^{(1 - \cos(x_1))^2} + \cos(x_1)e^{(1 - \sin(x_2))^2} + (x_1 - x_2)^2
\tag{22}
$$

with the constraints:

$$
\begin{aligned}
-10 &\leq x_1 \leq 0 \\
-6.5 &\leq x_2 \leq 0 \\
x_1^2 + x_2^2 &\geq 25.
\end{aligned}
\tag{23}
$$

The global minimum is $f((-3.1302, -1.5822)) \approx -106.7645$.

Figure 12 shows the landscape of the Mishra's Bird function and its global minimum point.

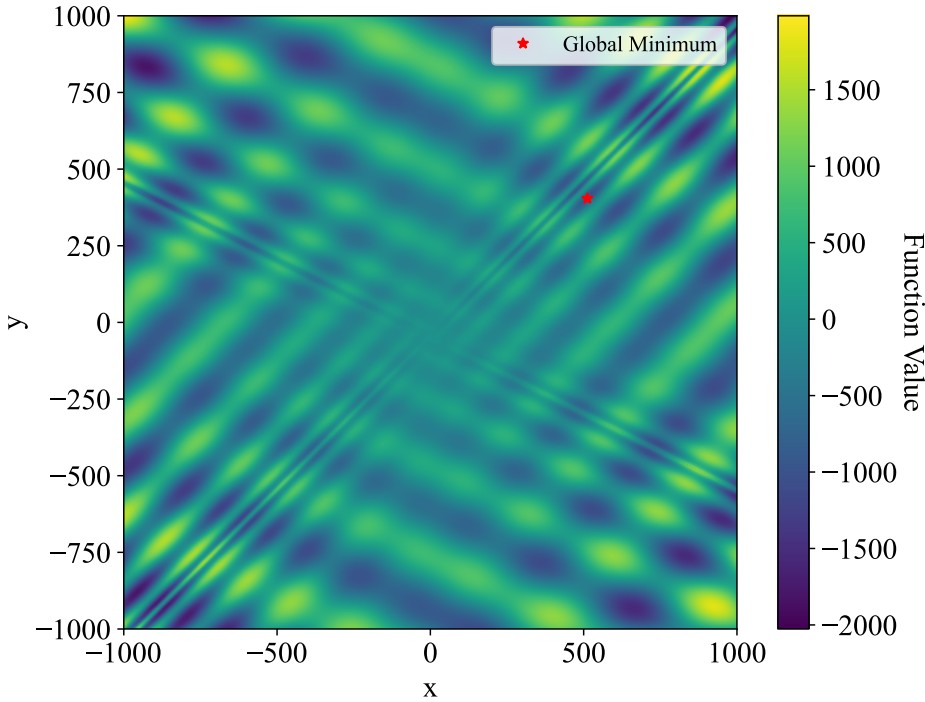

Figure 11: The landscape and global minimum point of Eggholder function with constraints $-512 \leq x, y \leq 512$

### B.4.3 KEANES BUMP FUNCTION

The Keanes Bump function is a challenging, non-convex test function commonly used to evaluate the performance of optimization algorithms in handling high-dimensional problems with complex constraints. The function's landscape is highly irregular, containing numerous local minima, and its feasible region is a small, irregular subset of the search space.

Let $d$ be the dimension of variables and $f : \mathbb{R}^d \rightarrow \mathbb{R}$, the function is defined as:

$$f(\mathbf{x}) = \frac{-|\sum_{i=1}^{d} \cos^4(x_i) - 2\prod_{i=1}^{d} \cos^2(x_i)|}{\sqrt{\sum_{i=1}^{d} ix_i^2}} \tag{24}$$

with the following constraints:

$$0 < x_i \leq 10, \qquad \forall 1 \leq i \leq d$$

$$\sum_{i=1}^{d} x_i \leq 7.5d \tag{25}$$

$$\prod_{i=1}^{d} x_i \geq 0.75.$$

The global minimum is located within the feasible region, which is a small, bounded area defined by these constraints. The image in this document, Figure 13, shows a two-dimensional visualization of the function's landscape. However, for our experiments, we tested the function in its 10-D, 20-D, and 30-D versions, where the complexity increases significantly. The global minima and their corresponding function values for these dimensions are listed below.

- **10-D Version**: The global minimum value is approximately $-0.747310362$.
- **20-D Version**: The global minimum value is approximately $-0.803619104$.
- **30-D Version**: The global minimum value is approximately $-0.818056222$.

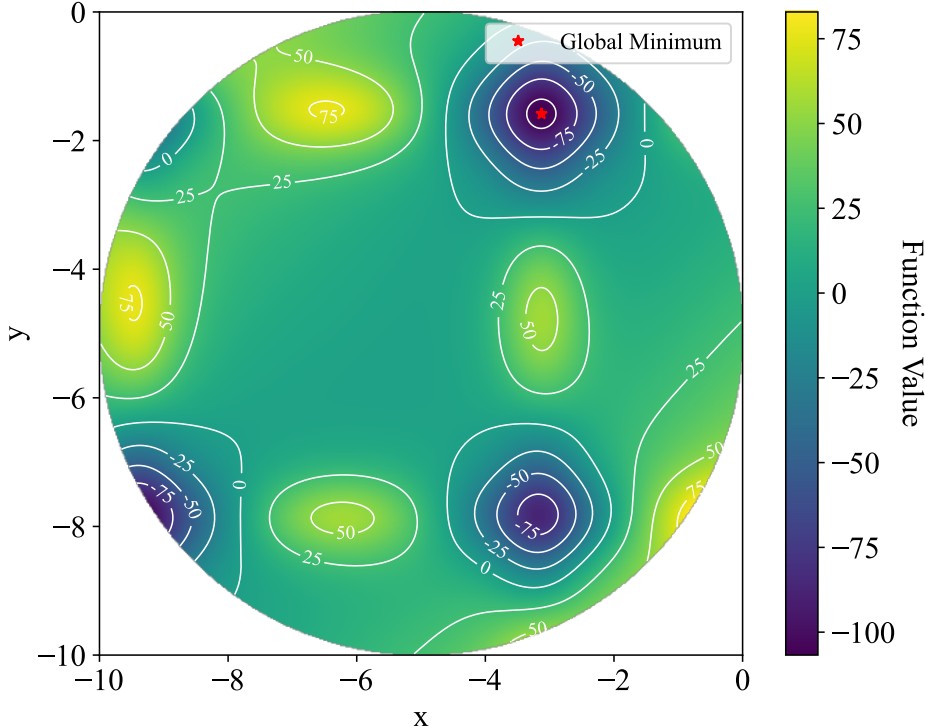

Figure 12: The landscape and global minimum point of Mishra's Bird function with constraints $x_1 \in [-10, 0]$, $x_2 \in [-6.5, 0]$ and $x_1^2 + x_2^2 \geq 25$.

### B.5 SYMBOLIC REGRESSION

In this task, the model has to uncover symbolic mathematical expressions from observational data. The benchmark and baselines are provided by Shojaee et al. (2025), which includes equations and data in chemistry, biology, physics and material science domains. In each category, several cases are created, each containing its own train and test sets generated by the same underlying equation. The model trained on the train set has to propose an expression to minimize the normalized mean square error (NMSE) on the test set, which is defined as:

$$\text{NMSE} = \frac{\sum_{i=1}^{N} (\hat{y}_i - y_i)^2}{\sum_{i=1}^{N} (y_i - \bar{y})^2}, \quad (26)$$

where $N$ is the number of observations in the test set.

To ensure a fair and robust comparison with the benchmark paper's results, we use the median of the NMSE calculated across all tasks within the same category $c$:

$$\text{NMSE}_c = \text{median}(\text{NMSE}_{c,1}, \text{NMSE}_{c,2}, \ldots, \text{NMSE}_{c,n}). \quad (27)$$

The reward we used for reinforcement learning for category $c$ is then set to:

$$R_c = -\log_{10}(\text{NMSE}_c). \quad (28)$$

In the benchmark, all the methods have a limit of 1000 trials for each single case, and we obey the same rule in our experiments, adjusting the number of training steps accordingly.

## C    DESCRIPTION OF TASK SPECIFIC BASELINES

In this section, we introduce the task-specific baseline methods and describe their implementation details.

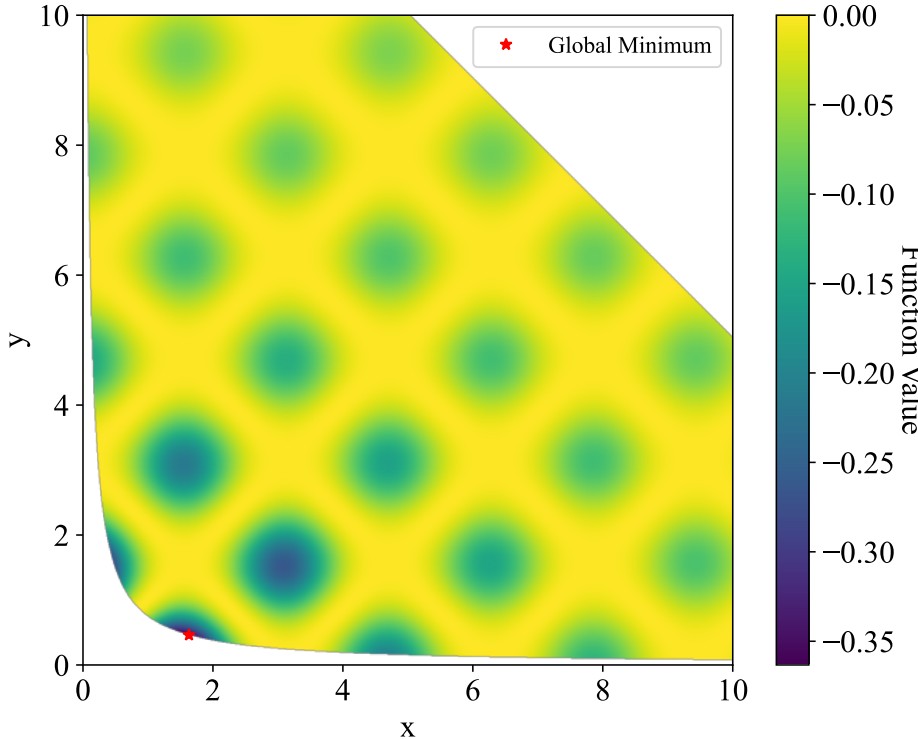

Figure 13: The landscape and global minimum point of the 2-D Keanes Bump function. The feasible region, a small part of the search space, is the only area with finite function values.

**Machine Learning.** For machine learning benchmarks, we evaluate two interpretable yet competitive models: LightGBM (Ke et al., 2017), a gradient boosting framework widely adopted in practice, and Rule-based Representation Learner (RRL) (Wang et al., 2021), which learns discrete non-fuzzy rules via gradient grafting to achieve both scalability and interpretability.

**Physics Simulation.** For physics-related optimization problems, we use two widely adopted modules in COMSOL Multiphysics: parameter search and topology optimization. For parameter search, we first parameterize the geometry based on initial solutions provided by human experts, and then optimize within the search space defined by these parameters. For topology optimization, human experts specify deformable geometric regions, while COMSOL applies its built-in topology optimization solvers to iteratively refine the structure.

**Circle Packing.** We consider two strong baselines: Sequential Least Squares Programming (SLSQP) (Lawson & Hanson, 1995) and a Genetic Algorithm (GA). SLSQP formulates circle packing as a constrained optimization problem, maximizing the sum of radii subject to boundary and non-overlap constraints. The GA baseline encodes circle positions and radii, evolves a feasible population with selection, crossover, and mutation, and evaluates fitness by the total radii.

**Function Minimization.** We adopt two standard constrained optimization solvers from `scipy.optimize`: Sequential Least Squares Programming (SLSQP) and the trust-constr method (Conn et al., 2000). Both are widely used gradient-based methods that provide strong task-specific baselines for function minimization.

**Symbolic Regression.** For symbolic regression tasks, we directly use results reported in LLM-SRBench (Shojaee et al., 2025), obtained by GPT-4o-mini running two recent methods: LaSR (Grayeli et al., 2024), which enhances evolutionary search with LLM-guided concept discovery, and

LLM-SR (Shojaee et al., 2024), which combines LLM scientific priors with evolutionary equation search. These represent competitive state-of-the-art baselines for symbolic regression.

# D  EXAMPLE OF PROMPTS USED IN EACH EXPERIMENT

In this section, we will demonstrate the prompts we used in our experiments.

## D.1  MACHINE LEARNING

---

**Prompt for task Adult Income**

```
You are an expert software developer tasked with iteratively improving
↪  a codebase.
Your job is to analyze the current program and suggest improvements
↪  based on feedback from previous attempts.
Focus on making targeted changes that will increase the program's
↪  performance metrics.
Respond in the following format: <think>
...
</think>
<answer>
...
</answer>.
# Problem Description

You are an expert in traditional machine learning.
Your task is to build a predictive model using the **Adult Income
↪  Dataset** (also known as the "Census Income" dataset).
This dataset contains demographic and employment-related attributes
↪  collected from the 1994 U.S. Census database.
The goal is to **predict whether a client will subscribe to a term
↪  deposit** (`y` column: yes/no) based on demographic and
↪  marketing-related features.
The goal is to **predict whether a person's income exceeds \$50K per
↪  year** (income column: >50K / <=50K) based on individual and
↪  employment features.

## Dataset Features

### Target variable:
- **income**: `>50K`, `<=50K`. (Parsed to 1/0 in Program)

### Input variables:

1. **age** *(numeric)*
   Age of the individual.

2. **workclass** *(categorical)*
   Type of employment:
   `Private`, `Self-emp-not-inc`, `Self-emp-inc`, `Federal-gov`,
   ↪  `Local-gov`,
   `State-gov`, `Without-pay`, `Never-worked`.

3. **fnlwgt** *(numeric)*
   Final sampling weight | indicates the number of people represented
   ↪  by this record.

4. **education** *(categorical)*
   Education level:
   `Bachelors`, `Some-college`, `11th`, `HS-grad`, `Prof-school`,
   `Assoc-acdm`, `Assoc-voc`, `9th`, `7th-8th`, `12th`, `Masters`,
   `1st-4th`, `10th`, `Doctorate`, `5th-6th`, `Preschool`.
```

---

5. **education-num** *(numeric)*
   Encodes years of education.

6. **marital-status** *(categorical)*
   `Married-civ-spouse`, `Divorced`, `Never-married`, `Separated`,
   `Widowed`, `Married-spouse-absent`, `Married-AF-spouse`.

7. **occupation** *(categorical)*
   `Tech-support`, `Craft-repair`, `Other-service`, `Sales`,
   ↪ `Exec-managerial`,
   `Prof-specialty`, `Handlers-cleaners`, `Machine-op-inspct`,
   ↪ `Adm-clerical`,
   `Farming-fishing`, `Transport-moving`, `Priv-house-serv`,
   ↪ `Protective-serv`,
   `Armed-Forces`.

8. **relationship** *(categorical)*
   `Wife`, `Own-child`, `Husband`, `Not-in-family`, `Other-relative`,
   ↪ `Unmarried`.

9. **race** *(categorical)*
   `White`, `Asian-Pac-Islander`, `Amer-Indian-Eskimo`, `Other`,
   ↪ `Black`.

10. **sex** *(categorical)*
    `Female`, `Male`.

11. **capital-gain** *(numeric)*
    Income from capital gains.

12. **capital-loss** *(numeric)*
    Losses from capital assets.

13. **hours-per-week** *(numeric)*
    Number of working hours per week.

14. **native-country** *(categorical)*
    `United-States`, `Cambodia`, `England`, `Puerto-Rico`, `Canada`,
    ↪ `Germany`,
    `Outlying-US(Guam-USVI-etc)`, `India`, `Japan`, `Greece`, `South`,
    ↪ `China`,
    `Cuba`, `Iran`, `Honduras`, `Philippines`, `Italy`, `Poland`,
    ↪ `Jamaica`,
    `Vietnam`, `Mexico`, `Portugal`, `Ireland`, `France`,
    ↪ `Dominican-Republic`,
    `Laos`, `Ecuador`, `Taiwan`, `Haiti`, `Columbia`, `Hungary`,
    ↪ `Guatemala`,
    `Nicaragua`, `Scotland`, `Thailand`, `Yugoslavia`, `El-Salvador`,
    `Trinadad&Tobago`, `Peru`, `Hong`, `Holand-Netherlands`.

## Additional Notes
- You may add, delete, or modify functions arbitrarily, but the
  ↪ program must still contain the `run_model()` function.
- If you want to use new packages, please import them explicitly.
- Try different **data preprocessing**, **feature engineering**, and
  ↪ **modeling techniques** to improve performance.
- Pay attention to Missing values: represented as "?". Handle them
  ↪ properly.
- Pay attention to Categorical encoding: many features are
  ↪ categorical; choose an effective encoding strategy.

## Task

```
Write a machine learning model to **predict** whether a person's
↪   income exceeds \$50K.

You will be given a **starter program** in Python.
Your goal is to **improve this program** to maximize the **F1-score**
↪   on the test set.

Your code execution time should **not exceed 60 seconds**.
You MUST use the exact SEARCH/REPLACE diff format shown below when
↪   modifying code:

<<<<<<< SEARCH
# Original code to find and replace (must match exactly)
=======
# New replacement code
>>>>>>> REPLACE

## Current Program
Status: {current_status}
```python
{current_program}
```
```

**Prompt for task Bank Marketing**

```
You are an expert software developer tasked with iteratively improving
↪   a codebase.
Your job is to analyze the current program and suggest improvements
↪   based on feedback from previous attempts.
Focus on making targeted changes that will increase the program's
↪   performance metrics.
Respond in the following format: <think>
...
</think>
<answer>
...
</answer>.
# Problem Description

You are an expert in traditional machine learning.
Your task is to build a predictive model using the **Bank Marketing
↪   Dataset**.
This dataset contains information collected from direct marketing
↪   campaigns conducted by a Portuguese banking institution.
The goal is to **predict whether a client will subscribe to a term
↪   deposit** (`y` column: yes/no) based on demographic and
↪   marketing-related features.

## Dataset Features
    Input variables:
### bank client data:
    1 - age (numeric)
    2 - job : type of job (categorical: "admin.", "blue-collar",
    ↪   "entrepreneur", "housemaid", "management", "retired",
    ↪   "self-employed", "services", "student", "technician",
    ↪   "unemployed", "unknown")
    3 - marital : marital status (categorical: "divorced", "married",
    ↪   "single", "unknown"; note: "divorced" means divorced or
    ↪   widowed)
```

```
    4 - education (categorical: "basic.4y", "basic.6y", "basic.9y",
    ↪  "high.school", "illiterate", "professional.course",
    ↪  "university.degree", "unknown")
    5 - default: has credit in default? (categorical:
    ↪  "no","yes","unknown")
    6 - housing: has housing loan? (categorical: "no","yes","unknown")
    7 - loan: has personal loan? (categorical: "no","yes","unknown")
### related with the last contact of the current campaign:
    8 - contact: contact communication type (categorical:
    ↪  "cellular","telephone")
    9 - month: last contact month of year (categorical: "jan", "feb",
    ↪  "mar", ..., "nov", "dec")
   10 - day_of_week: last contact day of the week (categorical:
    ↪  "mon","tue","wed","thu","fri")
   11 - duration: last contact duration, in seconds (numeric).
### other attributes:
   12 - campaign: number of contacts performed during this campaign and
    ↪  for this client (numeric, includes last contact)
   13 - pdays: number of days that passed by after the client was last
    ↪  contacted from a previous campaign (numeric; 999 means client
    ↪  was not previously contacted)
   14 - previous: number of contacts performed before this campaign and
    ↪  for this client (numeric)
   15 - poutcome: outcome of the previous marketing campaign
    ↪  (categorical: "failure","nonexistent","success")
### social and economic context attributes
   16 - emp.var.rate: employment variation rate - quarterly indicator
    ↪  (numeric)
   17 - cons.price.idx: consumer price index - monthly indicator
    ↪  (numeric)
   18 - cons.conf.idx: consumer confidence index - monthly indicator
    ↪  (numeric)
   19 - euribor3m: euribor 3 month rate - daily indicator (numeric)
   20 - nr.employed: number of employees - quarterly indicator
    ↪  (numeric)

## Task

Write a machine learning model to **predict** whether a client
↪  subscribes to a term deposit.

You will be given a **starter program** in Python.
Your goal is to **improve this program** to maximize the **F1-score**
↪  on the test set.

Your code execution time should **not exceed 60 seconds**.
You MUST use the exact SEARCH/REPLACE diff format shown below when
↪  modifying code:

<<<<<<< SEARCH
# Original code to find and replace (must match exactly)
=======
# New replacement code
>>>>>>> REPLACE

## Additional Notes
- You may add, delete, or modify functions arbitrarily, but the
↪  program must still contain the `run_model()` function.
- If you want to use new packages, please import them explicitly.
- Try different **data preprocessing**, **feature engineering**, and
↪  **modeling techniques** to improve performance.

## Current Program
```

```
Status: {current_status}
```python
{current_program}
```
```

---

**Prompt for task Boston Housing**

```
You are an expert software developer tasked with iteratively improving
↪   a codebase.
Your job is to analyze the current program and suggest improvements
↪   based on feedback from previous attempts.
Focus on making targeted changes that will increase the program's
↪   performance metrics.
Respond in the following format: <think>
...
</think>
<answer>
...
</answer>.
```

# Problem Description

You are an **expert in traditional machine learning**.
Your task is to build a **predictive regression model** using the
↪   **Boston Housing Dataset**.

The **Boston Housing Dataset** contains information collected by the
↪   U.S. Census Service concerning housing in the Boston,
↪   Massachusetts area.
The goal is to **predict the median value of owner-occupied homes**
↪   (`MEDV`, measured in \$1000s) based on various demographic,
↪   economic, and geographic factors.

## Dataset Features

The dataset contains **13 numerical and categorical features**. Some
↪   of them may have missing values (nan in dataframe)

1. **CRIM** { Per capita crime rate by town
2. **ZN** { Proportion of residential land zoned for lots over 25,000
↪   sq.ft.
3. **INDUS** { Proportion of non-retail business acres per town
4. **CHAS** { Charles River dummy variable (1 if tract bounds river; 0
↪   otherwise)
5. **NOX** { Nitric oxides concentration (parts per 10 million)
6. **RM** { Average number of rooms per dwelling
7. **AGE** { Proportion of owner-occupied units built prior to 1940
8. **DIS** { Weighted distances to five Boston employment centres
9. **RAD** { Index of accessibility to radial highways
10. **TAX** { Full-value property tax rate per \$10,000
11. **PTRATIO** { Pupil-teacher ratio by town
12. **LSTAT** { Percentage of lower status population
13. **MEDV** { **Target variable**: Median value of owner-occupied
↪   homes in \$1000s

## Task

You will be provided with a **starter Python program**.
Your objective is to **improve the program** to build a more accurate
↪   **regression model** for predicting `MEDV`.
Your improvements should focus on **maximizing the RMSE score** on the
↪   **test set** (RMSE score = 2 - log10(RMSE)).

## Requirements

* Your code execution time **must not exceed 60 seconds**.
* You MUST use the **SEARCH/REPLACE diff format** exactly as shown
  ↪  below when modifying the code:

```
<<<<<<< SEARCH
# Original code to find and replace (must match exactly)
=======
# New replacement code
>>>>>>> REPLACE
```

## Additional Notes

* You **may add, delete, or modify functions** as needed, but the
  ↪  program **must still contain** the `run_model()` function.
* If you want to use new packages, please import them explicitly.
  ↪  Usable packages: pandas, numpy, sklearn, scipy, statsmodels,
  ↪  xgboost, lightgbm, catboost, category_encoders, imbalanced-learn
* Try different **data preprocessing**, **feature engineering**, and
  ↪  **modeling techniques** to improve performance.

## Current Program
Status: {current_status}
```python
{current_program}
```

---

**prompt for task Predict Transparent Conductors**

```
You are an expert software developer tasked with iteratively improving
↪  a codebase.
Your job is to analyze the current program and suggest improvements
↪  based on feedback from previous attempts.
Focus on making targeted changes that will increase the program's
↪  performance metrics.
Respond in the following format: <think>
...
</think>
<answer>
...
</answer>.
```
# Overview

## Description

```
Innovative materials design is needed to tackle some of the most
↪   important health, environmental, energy, social, and economic
↪   challenges of this century. In particular, improving the
↪   properties of materials that are intrinsically connected to the
↪   generation and utilization of energy is crucial if we are to
↪   mitigate environmental damage due to a growing global demand.
↪   Transparent conductors are an important class of compounds that
↪   are both electrically conductive and have a low absorption in the
↪   visible range, which are typically competing properties. A
↪   combination of both of these characteristics is key for the
↪   operation of a variety of technological devices such as
↪   photovoltaic cells, light-emitting diodes for flat-panel displays,
↪   transistors, sensors, touch screens, and lasers. However, only a
↪   small number of compounds are currently known to display both
↪   transparency and conductivity suitable enough to be used as
↪   transparent conducting materials.

Aluminum (Al), gallium (Ga), indium (In) sesquioxides are some of the
↪   most promising transparent conductors because of a combination of
↪   both large bandgap energies, which leads to optical transparency
↪   over the visible range, and high conductivities. These materials
↪   are also chemically stable and relatively inexpensive to produce.
↪   Alloying of these binary compounds in ternary or quaternary
↪   mixtures could enable the design of a new material at a specific
↪   composition with improved properties over what is current
↪   possible. These alloys are described by the formula $(Al_x Ga_y
↪   In_z)_{{2N}}O_{{3N}}$ ; where x, y, and z can vary but are limited
↪   by the constraint x+y+z = 1. The total number of atoms in the unit
↪   cell, $\N_{{total}}=2N+3N$ (where N is an integer), is typically
↪   between 5 and 100. However, the main limitation in the design of
↪   compounds is that identification and discovery of novel materials
↪   for targeted applications requires an examination of enormous
↪   compositional and configurational degrees of freedom (i.e., many
↪   combinations of x, y, and z). To avoid costly and inefficient
↪   trial-and-error of synthetic routes, computational data-driven
↪   methods can be used to guide the discovery of potentially more
↪   efficient materials to aid in the development of advanced (or
↪   totally new) technologies. In computational material science, the
↪   standard tool for computing these properties is the
↪   quantum-mechanical method known as density-functional theory
↪   (DFT). However, DFT calculations are expensive, requiring hundreds
↪   or thousands of CPU hours on supercomputers for large systems,
↪   which prohibits the modeling of a sizable number of possible
↪   compositions and configurations. As a result, potential $(Al_x
↪   Ga_y In_z)_{{2N}}O_{{3N}}$ materials remain relatively unexplored.
↪   Data-driven models offer an alternative approach to efficiently
↪   search for new possible compounds in targeted applications but at
↪   a significantly reduced computational cost.

This competition aims to accomplish this goal by asking participants
↪   to develop or apply data analytics/data mining/machine-learning
↪   models for the prediction of two target properties: the formation
↪   energy (which is an indication of the stability of a new material)
↪   and the bandgap energy (which is an indication of the potential
↪   for transparency over the visible range) to facilitate the
↪   discovery of new transparent conductors and allow for advancements
↪   in the above-mentioned technologies.

## Evaluation

Submissions are evaluated on the column-wise root mean squared
↪   logarithmic error.
```

```
The RMSLE for a single column calculated as

$\sqrt{{\frac{{1}}{{n}}\sum_{{i=1}}^n(\log{{(p_i + 1)}} -
↪  \log{{(a_i+1)}})^2}}$

where:

$n$ is the total number of observations
$p_i$ is your prediction
$a_i$ is the actual value
$\log(x)$ is the natural logarithm of $x$

The final score is the mean of the RMSLE over all columns (in this
↪  case, 2).
```

# Dataset Description

```
High-quality data are provided for 3,000 materials that show promise
↪  as transparent conductors. The following information has been
↪  included:

- spacegroup: Crystallographic space group number describing the
↪  symmetry of the material.
- number_of_total_atoms: Total number of atoms in the unit cell.
- percent_atom_al, percent_atom_ga, percent_atom_in: Relative
↪  composition of Al, Ga, and In in the material (remaining fraction
↪  is O).
- lattice_vector_1_ang, lattice_vector_2_ang, lattice_vector_3_ang:
↪  Lengths of the three lattice vectors (in angstroms), describing
↪  the unit cell dimensions.
- lattice_angle_alpha_degree, lattice_angle_beta_degree,
↪  lattice_angle_gamma_degree: Angles between the lattice vectors (in
↪  degrees), defining the unit cell geometry.

A domain expert will understand the physical meaning of the above
↪  information but those with a data mining background may simply use
↪  the data as input for their models.

The task for this competition is to predict two target properties:

- Formation energy (an important indicator of the stability of a
↪  material)
- Bandgap energy (an important property for optoelectronic
↪  applications)
```

# Task

## Additional Notes

```
* You **may add, delete, or modify functions** as needed, but the
↪  program **must still contain** the `run_model()` function.
* If you want to use new packages, please import them explicitly.
↪  Usable packages: pandas, numpy, sklearn, scipy, statsmodels,
↪  xgboost, lightgbm, catboost, category_encoders, imbalanced-learn
* Try different **data preprocessing**, **feature engineering**, and
↪  **modeling techniques** to improve performance.
```

## Requirements

```
* Your code execution time **must not exceed 60 seconds**.
* You MUST use the **SEARCH/REPLACE diff format** exactly as shown
↪  below when modifying the code:
```

```
```
<<<<<<< SEARCH
# Original code to find and replace (must match exactly)
=======
# New replacement code
>>>>>>> REPLACE
```

## Current Program
Status: {current_status}
```python
{current_program}
```
```

## D.2  Physics Simulation

> **Prompt for task Inductor**
>
> ```
> You are a helpful AI Assistant that provides well-reasoned and
> ↪   detailed responses.
> You first think about the reasoning process as an internal monologue
> ↪   and then provide the user with the answer.
> Respond in the following format: <think>
> ...
> </think>
> <answer>
> ...
> </answer>.
> ## Task Description
>
> You are a helpful AI Assistant and scientist with strong physical
> ↪   background and wonderful geometric designing ideas.
> You are asked to generate the geometry design of a component using
> ↪   yaml files under certain constraints. You will first create
> ↪   geometries of your design, and then assign functions to the
> ↪   geometries according to the specific requirements.
>
> Your final answer should contain a yaml file enclosed in
> ↪   ```yaml\n(your code)```. The yaml file should have at least two
> ↪   parts: geometry and selection. The specific requirements are as
> ↪   follow:
>
> 1. geometry: A list of objects with type and type-specific parameters.
> ↪   The types and parameters are as follows:
>     Polygon: (2D) You can use it to create rectangles, triangles, etc.
>         table:  Ordered list of n vertices as [x, y] points. The
>         ↪   polygon is formed by **connecting consecutive points**
>         ↪   (p_i->p_{i+1}) and **automatically closing** the shape
>         ↪   (p_n->p_1).
>         fillet: (Optional) A list of [i, r] tuples, where i is the
>         ↪   index (starting from 1) of a polygon vertex defined in the
>         ↪   above table, and r is the fillet radius for that
>         ↪   corresponding vertex.
>     Ellipse: (2D) You can use it to create circles.
>         semiaxes: [horizontal, vertical] axis lengths
>         pos: [center_x, center_y] center position
>         rot: (Optional) Rotation angle (degree) counterclockwise
>         angle: (Optional) Angular span (degree) counterclockwise. e.g.
>         ↪   by setting angle=180 you can draw a upward semicircle.
>     LineSegment: (1D)
>         coord1: [start_x, start_y]
>         coord2: [end_x, end_y]
> ```

```
    CircularArc: (1D)
        r: Radius
        angle1: Start angle (degree) counterclockwise, 0 degree
        ↪   represent positive direction of X-axis.
        angle2: End angle (degree) counterclockwise
    CubicBezier: (1D)
        p: Control points as [[x0,x1,x2,x3], [y0,y1,y2,y3]]
        w: Weight values as [w0,w1,w2,w3]
    InterpolationCurve: (1D)
        table: Ordered list of [x,y] points to interpolate through.
        ↪   The curve will pass every points smoothly (polynomial
        ↪   interpolation for x and y).
    ParametricCurve: (1D)
        parname: Name of parameter
        parmin: Minimum value of parameter
        parmax: Maximum value of parameter
        coord: Expressions about the parameter like ["expression_x",
        ↪   "expression_y"]. Trigonometric functions here use radians
    ConvertToSolid: (2D) Geometry formed by end-to-end connected 1D
    ↪   curves.
        geometries: A dictionary of 1D geometries (using the same
        ↪   structure as the top-level geometry section, recursive).
        ↪   **They Must connect end-to-end and form a simply connected
        ↪   space**.
    Union: (2D) Union of 2D geometries.
        geometries: A dictionary of geometries (recursive).
    Intersection: (2D) Intersection of 2D geometries.
        geometries: A dictionary of geometries (recursive).
    Difference: (2D) Difference of the 2D geometries.
        geometries_add: A dictionary of geometries to keep
        ↪   (recursive).
        geometries_subtract: A dictionary of geometries to subtract
        ↪   (recursive).

After **geometry** was created, the shapes will be splitted into
↪   **non-overlapping connected regions**.
    - Overlapping 2D shapes create new regions (e.g., two intersecting
    ↪   circles → 3 regions)
    - Enclosed 2D shapes split regions (e.g., circle inside polygon →
    ↪   2 regions: circle interior + polygon-ring)
    - 1D curves through 2D shapes create sub-regions (e.g., line
    ↪   segment through rectangle → alternating regions)
The **regions** can be represented by the following ways:
    - point: You can select an interior point of the region to
    ↪   represent it. The point should never on boundaries/corners.
    ↪   One point per region suffices.
    - geometry: The 2d shapes you created might be splitted into
    ↪   several regions. You can select the geometry to represent all
    ↪   the regions in it.

2. selection: After regions are created, you will assign different
↪   functions to regions using selections.
    UnionSelection: Union of all the regions selected below.
        points: (Optional) List of [x,y] points representing distinct
        ↪   regions.
        geometries: (Optional) List of 2d geometry names you created
        ↪   above. By listing geometries here, you can select all the
        ↪   region this geometry contains.
        selections: (Optional) List of other selection names you
        ↪   created.
    IntersectionSelection: Intersection of all the regions selected
    ↪   below.
        same parameters as UnionSelection
```

```
    DifferenceSelection: Select the regions in Add but not in
    ↪   Subtract.
        add: same parameters as UnionSelection.
        subtract: same parameters as UnionSelection.

Finally a yaml file will be like the following sample:
```

```yaml
geometry:
    uni1: # Name of this geometry
        type: Union
        geometries: # create geometries recursively below
            uni_el1: # Name of the first ellipse to union
                type: Ellipse # Type of this geometry
                semiaxes: [2.0, 1.0]  # Specific parameters
                pos: [1.0, 1.0]
            uni_pol1: # Name of the second polygon
                type: Polygon
                table:
                - [-1.0, -0.3]
                - [2.0, -1.0]
                - [1.0, 1.0]
    line1: # This line splits the ellipse into 2 regions.
        type: LineSegment
        coord1: [1.0, 2.0]
        coord2: [3.0, 1.0]

selection:
    sel1: # Name of this selection
        type: DifferenceSelection
        add:
            geometries:
            - uni1 # Select all the regions in uni1
        subtract:
            points:
            - [2.5, 1.5] # Remove the region where (2.5, 1.5) in. This
            ↪   region is part of ellipse but splitted by the line
            ↪   segment.
```

## Geometric Design of Inductor2d

You are asked to design an inductor. The objective is to maximize the
↪   inductance value of the inductor which is calculated as
↪   $0.5*real(Br*conj(Hr)+conj(Hphi)*Bphi+conj(Hz)*Bz)/V$ with B, H to
↪   be the magnetic induction intensity and magnetic field intensity
↪   and V to be the volume of the core.

The geometry should be designed inside a semicircle of radius 0.35m
↪   centered at (0,0), opening in the positive x-direction. Then we
↪   will generate a 3D geometry by rotating the semicircle around the
↪   axis x=0.

You are required to give the geometry of the core, and the location of
↪   the coils. After you create the geometry, you should select the
↪   regions of the core. **The Name of the selection must be `core`**.
↪   Finally, you need to give the center of the coils, which are
↪   circles with radius 0.01m. You don't need to give the geometry of
↪   the coils.

The constraints are as follow:
**1.** There's no overlapping of different coils. There must be 12 coils
↪   in total.

**2.** The core and the coils should not overlap or adjacent.
**3.** The geometry and coils should be placed inside the semicircle of
↪  radius 0.35m centered at (0,0), opening in the positive
↪  x-direction.
**4.** The volume of the core should be more than 0.001 m^3. This means
↪  cores that are **extremely thin or extremely fine** are not
↪  allowed.

The reward is calculated as follow:
**1.** 0 if constraints are violated.
**2.** $0.5*real(Br*conj(Hr)+conj(Hphi)*Bphi+conj(Hz)*Bz)/V$, the
↪  inductance value of the inductor, if constraints are satisfied.

## Example
An example solution is shown below. You should not copy the example
↪  solution, but you can refer to it to understand the task and
↪  create better ones.

```yaml
geometry:
  main:
    type: Difference
    geometries_add:
      outer:
        type: Polygon
        table:
        - [0, 0.2]
        - [0.18, 0.2]
        - [0.18, -0.2]
        - [0, -0.2]
    geometries_subtract:
      inner:
        type: Polygon
        table:
        - [0.03, -0.155]
        - [0.03, 0.155]
        - [0.12, 0.155]
        - [0.12, -0.155]
selection:
  core:
    type: UnionSelection
    geometries:
    - main
coils:
  - [0.08, 0.11]
  - [0.08, 0.09]
  - [0.08, 0.07]
  - [0.08, 0.05]
  - [0.08, 0.03]
  - [0.08, 0.01]
  - [0.08, -0.01]
  - [0.08, -0.03]
  - [0.08, -0.05]
  - [0.08, -0.07]
  - [0.08, -0.09]
  - [0.08, -0.11]
```

---

**Prompt for task Beam Bending**

```
You are a helpful AI Assistant that provides well-reasoned and
↪  detailed responses.
You first think about the reasoning process as an internal monologue
↪  and then provide the user with the answer.
Respond in the following format: <think>
...
</think>
<answer>
...
</answer>.
## Task Description

You are a helpful AI Assistant and scientist with strong physical
↪  background and wonderful geometric designing ideas.
You are asked to generate the geometry design of a component using
↪  yaml files under certain constraints. You will first create
↪  geometries of your design, and then assign functions to the
↪  geometries according to the specific requirements.

Your final answer should contain a yaml file enclosed in
↪  ```yaml\n(your code)```. The yaml file should have at least two
↪  parts: geometry and selection. The specific requirements are as
↪  follow:

1. geometry: A list of objects with type and type-specific parameters.
↪  The types and parameters are as follows:
    Polygon: (2D) You can use it to create rectangles, triangles, etc.
        table:  Ordered list of n vertices as [x, y] points. The
        ↪  polygon is formed by **connecting consecutive points**
        ↪  (p_i->p_{i+1}) and **automatically closing** the shape
        ↪  (p_n->p_1). **NO Intersections between edges/nodes are
        ↪  allowed**.
        fillet: (Optional) A list of [i, r] tuples, where i is the
        ↪  index (starting from 1) of a polygon vertex defined in the
        ↪  above table, and r is the fillet radius for that
        ↪  corresponding vertex.
    Ellipse: (2D) You can use it to create circles.
        semiaxes: [horizontal, vertical] axis lengths
        pos: [center_x, center_y] center position
        rot: (Optional) Rotation angle (degree) counterclockwise
        angle: (Optional) Angular span (degree) counterclockwise. e.g.
        ↪  by setting angle=180 you can draw a upward semicircle.
    LineSegment: (1D)
        coord1: [start_x, start_y]
        coord2: [end_x, end_y]
    CircularArc: (1D)
        r: Radius
        angle1: Start angle (degree) counterclockwise, 0 degree
        ↪  represent positive direction of X-axis.
        angle2: End angle (degree) counterclockwise
    CubicBezier: (1D)
        p: Control points as [[x0,x1,x2,x3], [y0,y1,y2,y3]]
        w: Weight values as [w0,w1,w2,w3]
    InterpolationCurve: (1D)
        table: Ordered list of [x,y] points to interpolate through.
        ↪  The curve will pass every points smoothly (polynomial
        ↪  interpolation for x and y).
    ParametricCurve: (1D)
        parname: Name of parameter
        parmin: Minimum value of parameter
        parmax: Maximum value of parameter
```

```
            coord: Expressions about the parameter like ["expression_x",
            ↪   "expression_y"]. Trigonometric functions here use radians
        ConvertToSolid: (2D) Geometry formed by end-to-end connected 1D
        ↪   curves.
            geometries: A dictionary of 1D geometries (using the same
            ↪   structure as the top-level geometry section, recursive).
            ↪   **They Must connect end-to-end and form a simply connected
            ↪   space**.
        Union: (2D) Union of 2D geometries.
            geometries: A dictionary of geometries (recursive).
        Intersection: (2D) Intersection of 2D geometries.
            geometries: A dictionary of geometries (recursive).
        Difference: (2D) Difference of the 2D geometries.
            geometries_add: A dictionary of geometries to keep
            ↪   (recursive).
            geometries_subtract: A dictionary of geometries to subtract
            ↪   (recursive).

After **geometry** was created, the shapes will be splitted into
↪   **non-overlapping connected regions**.
    - Overlapping 2D shapes create new regions (e.g., two intersecting
    ↪   circles → 3 regions)
    - Enclosed 2D shapes split regions (e.g., circle inside polygon →
    ↪   2 regions: circle interior + polygon-ring)
    - 1D curves through 2D shapes create sub-regions (e.g., line
    ↪   segment through rectangle → alternating regions)
The **regions** can be represented by the following ways:
    - point: You can select an interior point of the region to
    ↪   represent it. The point should never on boundaries/corners.
    ↪   One point per region suffices.
    - geometry: The 2d shapes you created might be splitted into
    ↪   several regions. You can select the geometry to represent all
    ↪   the regions in it.

2. selection: After regions are created, you will assign different
↪   functions to regions using selections.
    UnionSelection: Union of all the regions selected below.
        points: (Optional) List of [x,y] points representing distinct
        ↪   regions.
        geometries: (Optional) List of 2d geometry names you created
        ↪   above. By listing geometries here, you can select all the
        ↪   region this geometry contains.
        selections: (Optional) List of other selection names you
        ↪   created.
    IntersectionSelection: Intersection of all the regions selected
    ↪   below.
        same parameters as UnionSelection
    DifferenceSelection: Select the regions in Add but not in
    ↪   Subtract.
        add: same parameters as UnionSelection.
        subtract: same parameters as UnionSelection.

Finally a yaml file will be like the following sample:

```yaml
geometry:
    uni1: # Name of this geometry
        type: Union
        geometries: # create geometries recursively below
            uni_el1: # Name of the first ellipse to union
                type: Ellipse # Type of this geometry
                semiaxes: [2.0, 1.0]  # Specific parameters
                pos: [1.0, 1.0]
```

```yaml
            uni_pol1: # Name of the second polygon
                type: Polygon
                table:
                - [-1.0, -0.3]
                - [2.0, -1.0]
                - [1.0, 1.0]
    line1: # This line splits the ellipse into 2 regions.
        type: LineSegment
        coord1: [1.0, 2.0]
        coord2: [3.0, 1.0]

selection:
    sel1: # Name of this selection
        type: DifferenceSelection
        add:
            geometries:
            - uni1 # Select all the regions in uni1
        subtract:
            points:
            - [2.5, 1.5] # Remove the region where (2.5, 1.5) in. This
            ↪   region is part of ellipse but splitted by the line
            ↪   segment.
```

## Beam Cross Section Geometry Design

You are asked to design the cross section of a beam. The objective is
↪   to maximize both the largest principal moment of inertia and the
↪   torsional constant, while keeping the cross section area small.
↪   The goal can be quantified as $(I1**0.8 * I2**0.2)/A$ with I1
↪   being the largest principal moment of inertia, I2 being the
↪   smallest principal moment of inertia and A being the beam cross
↪   section area.

The beam cross-sectional dimension should not go beyond 0.15m, with
↪   the center staying close to the origin.

You are required to design the beam cross section. The shape doesn't
↪   have to be symmetric. After you create the geometry, you should
↪   select the regions of the beam. **The Name of the selection must
↪   be `beam`**.

The constraints are as follow:
1. The shape center should be close to the origin.
2. The shape should stay inside the circle boundary with radius 0.2m.
3. The area should not be smaller than 2e-3 m^2.

The reward is calculated as follow:
1. 0 if constraints are violated.
2. $(I1**0.8 * I2**0.2)/A$, weighted geometry average of the largest
↪   principal moment of inertia and the smallest principal moment of
↪   inertia, normalized by the cross section area, if constraints are
↪   satisfied.

## Example
An example solution is shown below. You should not copy the example
↪   solution, but you can refer to it to understand the task and
↪   create better ones.

```yaml
geometry:
  pol1:
    type: Polygon
```

```yaml
    table:
    - [0.05, 0.05]
    - [-0.05, 0.05]
    - [-0.05, 0.04]
    - [-0.005, 0.04]
    - [-0.005, -0.04]
    - [-0.05, -0.04]
    - [-0.05, -0.05]
    - [0.05, -0.05]
    - [0.05, -0.04]
    - [0.005, -0.04]
    - [0.005, 0.04]
    - [0.05, 0.04]
    fillet:
    - [4, 0.003]
    - [5, 0.003]
    - [10, 0.003]
    - [11, 0.003]
selection:
  beam:
    type: UnionSelection
    geometries:
    - pol1
```

---

**Prompt for task Magnetic Torque**

```
You are a helpful AI Assistant that provides well-reasoned and
↪  detailed responses.
You first think about the reasoning process as an internal monologue
↪  and then provide the user with the answer.
Respond in the following format: <think>
...
</think>
<answer>
...
</answer>.
## Task Description

You are a helpful AI Assistant and scientist with strong physical
↪  background and wonderful geometric designing ideas.
You are asked to generate the geometry design of a component using
↪  yaml files under certain constraints. You will first create
↪  geometries of your design, and then assign functions to the
↪  geometries according to the specific requirements.

Your final answer should contain a yaml file enclosed in
↪  ```yaml\n(your code)```. The yaml file should have at least two
↪  parts: geometry and selection. The specific requirements are as
↪  follow:

1. geometry: A list of objects with type and type-specific parameters.
↪  The types and parameters are as follows:
    Polygon: (2D) You can use it to create rectangles, triangles, etc.
        table:  Ordered list of n vertices as [x, y] points. The
        ↪  polygon is formed by **connecting consecutive points**
        ↪  (p_i->p_{i+1}) and **automatically closing** the shape
        ↪  (p_n->p_1).
        fillet: (Optional) A list of [i, r] tuples, where i is the
        ↪  index (starting from 1) of a polygon vertex defined in the
        ↪  above table, and r is the fillet radius for that
        ↪  corresponding vertex.
```

```
    Ellipse: (2D) You can use it to create circles.
        semiaxes: [horizontal, vertical] axis lengths
        pos: [center_x, center_y] center position
        rot: (Optional) Rotation angle (degree) counterclockwise
        angle: (Optional) Angular span (degree) counterclockwise. e.g.
        ↪  by setting angle=180 you can draw a upward semicircle.
    LineSegment: (1D)
        coord1: [start_x, start_y]
        coord2: [end_x, end_y]
    CircularArc: (1D)
        r: Radius
        angle1: Start angle (degree) counterclockwise, 0 degree
        ↪  represent positive direction of X-axis.
        angle2: End angle (degree) counterclockwise
    CubicBezier: (1D)
        p: Control points as [[x0,x1,x2,x3], [y0,y1,y2,y3]]
        w: Weight values as [w0,w1,w2,w3]
    InterpolationCurve: (1D)
        table: Ordered list of [x,y] points to interpolate through.
        ↪  The curve will pass every points smoothly (polynomial
        ↪  interpolation for x and y).
    ParametricCurve: (1D)
        parname: Name of parameter
        parmin: Minimum value of parameter
        parmax: Maximum value of parameter
        coord: Expressions about the parameter like ["expression_x",
        ↪  "expression_y"]. Trigonometric functions here use radians
    ConvertToSolid: (2D) Geometry formed by end-to-end connected 1D
    ↪  curves.
        geometries: A dictionary of 1D geometries (using the same
        ↪  structure as the top-level geometry section, recursive).
        ↪  **They Must connect end-to-end and form a simply connected
        ↪  space**.
    Union: (2D) Union of 2D geometries.
        geometries: A dictionary of geometries (recursive).
    Intersection: (2D) Intersection of 2D geometries.
        geometries: A dictionary of geometries (recursive).
    Difference: (2D) Difference of the 2D geometries.
        geometries_add: A dictionary of geometries to keep
        ↪  (recursive).
        geometries_subtract: A dictionary of geometries to subtract
        ↪  (recursive).

After **geometry** was created, the shapes will be splitted into
↪  **non-overlapping connected regions**.
    – Overlapping 2D shapes create new regions (e.g., two intersecting
    ↪  circles → 3 regions)
    – Enclosed 2D shapes split regions (e.g., circle inside polygon →
    ↪  2 regions: circle interior + polygon-ring)
    – 1D curves through 2D shapes create sub-regions (e.g., line
    ↪  segment through rectangle → alternating regions)
The **regions** can be represented by the following ways:
    – point: You can select an interior point of the region to
    ↪  represent it. The point should never on boundaries/corners.
    ↪  One point per region suffices.
    – geometry: The 2d shapes you created might be splitted into
    ↪  several regions. You can select the geometry to represent all
    ↪  the regions in it.

2. selection: After regions are created, you will assign different
↪  functions to regions using selections.
    UnionSelection: Union of all the regions selected below.
```

```
        points: (Optional) List of [x,y] points representing distinct
        ↪  regions.
        geometries: (Optional) List of 2d geometry names you created
        ↪  above. By listing geometries here, you can select all the
        ↪  region this geometry contains.
        selections: (Optional) List of other selection names you
        ↪  created.
    IntersectionSelection: Intersection of all the regions selected
    ↪  below.
        same parameters as UnionSelection
    DifferenceSelection: Select the regions in Add but not in
    ↪  Subtract.
        add: same parameters as UnionSelection.
        subtract: same parameters as UnionSelection.

Finally a yaml file will be like the following sample:

```yaml
geometry:
    uni1: # Name of this geometry
        type: Union
        geometries: # create geometries recursively below
            uni_el1: # Name of the first ellipse to union
                type: Ellipse # Type of this geometry
                semiaxes: [2.0, 1.0]  # Specific parameters
                pos: [1.0, 1.0]
            uni_pol1: # Name of the second polygon
                type: Polygon
                table:
                - [-1.0, -0.3]
                - [2.0, -1.0]
                - [1.0, 1.0]
    line1: # This line splits the ellipse into 2 regions.
        type: LineSegment
        coord1: [1.0, 2.0]
        coord2: [3.0, 1.0]

selection:
    sel1: # Name of this selection
        type: DifferenceSelection
        add:
            geometries:
            - uni1 # Select all the regions in uni1
        subtract:
            points:
            - [2.5, 1.5] # Remove the region where (2.5, 1.5) in. This
            ↪  region is part of ellipse but splitted by the line
            ↪  segment.
```
```

## Geometric Design of 2D Iron Core

You are asked to design a 2D iron core. The iron core has large
↪ permeability and is subject to a constant far field magnetic field
↪ intensity which applies a magnetic torque (pointing out of the 2D
↪ plane) on the iron core. The objective is to maximize the magnetic
↪ torque while keeping the iron core small. The goal can be
↪ quantified as $|Tz|/A$ where Tz is the torque in the out of plane
↪ direction and A is the area of the core.

```
The iron core should be designed inside a circle air domain of radius
↪  0.05m centered at the origin (0,0). The boundary of the circle
↪  domain is subject to a constant magnetic field intensity of
↪  [0,1e5] A/m.

You are required to give the geometry of the iron core. The shape
↪  doesn't have to be symmetric. After you create the geometry, you
↪  should select the regions of the core. **The Name of the selection
↪  must be `core`**.

The constraints are as follow:
1. The core center should be close to the origin and inside the circle
↪  air domain of radius 0.05m centered at (0,0).
2. The boundary of the core should stay at least 0.02m away from the
↪  circle air domain.
3. The Area of the core should between 2e-4 and 2e-3 m^2.

The reward is calculated as follow:
1. 0 if constraints are violated.
2. $|Tz|/A$, the absolute value of magnetic torque generated by the
↪  constant far field magnetic field intensity on the iron core
↪  normalized by the iron core area, if constraints are satisfied.

## Example
An example solution is shown below. You should not copy the example
↪  solution, but you can refer to it to understand the task and
↪  create better ones.

```yaml
geometry:
  pol1:
    type: Polygon
    table:
    - [0.01, -0.003]
    - [0.02, 0.001]
    - [0.01, 0.01]
    - [-0.01, 0.005]
    - [-0.02, -0.008]
    - [-0.02, -0.008]
    - [-0.003, -0.01]
selection:
    core:
        type: UnionSelection
        geometries:
        - pol1
```
```

**Prompt for task Periodic Heat**

```
You are a helpful AI Assistant that provides well-reasoned and
↪  detailed responses.
You first think about the reasoning process as an internal monologue
↪  and then provide the user with the answer.
Respond in the following format: <think>
...
</think>
<answer>
...
</answer>.
## Task Description
```

```
You are a helpful AI Assistant and scientist with strong physical
↪   background and wonderful geometric designing ideas.
You are asked to generate the geometry design of a component using
↪   yaml files under certain constraints. You will first create
↪   geometries of your design, and then assign functions to the
↪   geometries according to the specific requirements.

Your final answer should contain a yaml file enclosed in
↪   ```yaml\n(your code)```. The yaml file should have a part named
↪   geometry. The specific requirements are as follow:

1. geometry: A list of objects with type and type-specific parameters.
↪   The types and parameters are as follows:
    Polygon: (2D) You can use it to create rectangles, triangles, etc.
        table:  Ordered list of n vertices as [x, y] points. The
        ↪   polygon is formed by **connecting consecutive points**
        ↪   (p_i->p_{i+1}) and **automatically closing** the shape
        ↪   (p_n->p_1).
        fillet: (Optional) A list of [i, r] tuples, where i is the
        ↪   index (starting from 1) of a polygon vertex defined in the
        ↪   above table, and r is the fillet radius for that
        ↪   corresponding vertex.
    Ellipse: (2D) You can use it to create circles.
        semiaxes: [horizontal, vertical] axis lengths
        pos: [center_x, center_y] center position
        rot: (Optional) Rotation angle (degree) counterclockwise
        angle: (Optional) Angular span (degree) counterclockwise. e.g.
        ↪   by setting angle=180 you can draw a upward semicircle.
    LineSegment: (1D)
        coord1: [start_x, start_y]
        coord2: [end_x, end_y]
    CircularArc: (1D)
        r: Radius
        angle1: Start angle (degree) counterclockwise, 0 degree
        ↪   represent positive direction of X-axis.
        angle2: End angle (degree) counterclockwise
    CubicBezier: (1D)
        p: Control points as [[x0,x1,x2,x3], [y0,y1,y2,y3]]
        w: Weight values as [w0,w1,w2,w3]
    InterpolationCurve: (1D)
        table: Ordered list of [x,y] points to interpolate through.
        ↪   The curve will pass every points smoothly (polynomial
        ↪   interpolation for x and y).
    ParametricCurve: (1D)
        parname: Name of parameter
        parmin: Minimum value of parameter
        parmax: Maximum value of parameter
        coord: Expressions about the parameter like ["expression_x",
        ↪   "expression_y"]. Trigonometric functions here use radians
    ConvertToSolid: (2D) Geometry formed by end-to-end connected 1D
    ↪   curves.
        geometries: A dictionary of 1D geometries (using the same
        ↪   structure as the top-level geometry section, recursive).
        ↪   **They Must connect end-to-end and form a simply connected
        ↪   space**.
    Union: (2D) Union of 2D geometries.
        geometries: A dictionary of geometries (recursive).
    Intersection: (2D) Intersection of 2D geometries.
        geometries: A dictionary of geometries (recursive).
    Difference: (2D) Difference of the 2D geometries.
        geometries_add: A dictionary of geometries to keep
        ↪   (recursive).
```

```
        geometries_subtract: A dictionary of geometries to subtract
        ↪    (recursive).
```

## Structure Design of the 3D heat transfer unit cell

You are asked to design a unit cell structure in 3D. The objective is
↪  to maximize the effective thermal conductivity with limited
↪  material usage, which can be quantified as
↪  $trace(k_{eff})/\rho_{eff}$ where $k_{eff}$ is the effective
↪  thermal conductivity matrix of shape 3*3, and rho_eff is the
↪  effective density.

You should first define a 2D rectangular unit cell domain by giving
↪  the width and height of the domain.

You then design the hollow part of the 2D unit cell. You must create a
↪  geometry named `hollow`, represents the hollow part of the unit
↪  cell. Four copies of this geometry object will be created by
↪  translating the original object with the following vectors
↪  [-cell_width/2, -cell_height/2], [-cell_width/2, cell_height/2],
↪  [cell_width/2, -cell_height/2], [cell_width/2, cell_height/2].

The unit cell domain will be subtracted by the geometry object and its
↪  copies. The subtracted areas are filled with air (thermal
↪  conductivity ~0.026W/(m * K), density ~1.174kg/m^3) and the
↪  remaining areas are filled with aluminum (thermal conductivity
↪  238W/(m * K), density 2700 kg/m^3).
The final 3D unit cell structure is generated by extruding the 2D unit
↪  cell.

The unit cell will subject to periodic boundary condition in x, y, and
↪  z directions. Your design should provide higher effective thermal
↪  conductivity using a reasonable amount of aluminum and carefully
↪  designed hollow shape.

The constraints are as follow:
1. The original geometry object(`hollow`) should not overlap with the
↪  boundary of the domain. But its copies may overlap with the
↪  boundary.
2. The original and copied geometry objects should not overlap or
↪  adjacent with each other.
3. The effective density should not exceed 2000 kg/m^3. You should
↪  control the usage of aluminum (by create larger hollow parts).
4. The unit cell domain is centered strictly at the origin. The
↪  geometry you designed should be centered approximately at the
↪  origin, as it may not be symmetric.

The reward is calculated as follow:
1. 0 if constraints are violated.
2. $trace(k_eff)/rho_eff$, the effective thermal conductivity of the
↪  unit cell structure normalized by the effective density, if
↪  constraints are satisfied.

## Example
An example solution is shown below. You should not copy the example
↪  solution, but you can refer to it to understand the task and
↪  create better ones.

```yaml
cell:
    sizes: [5e-3, 3e-3]
geometry:
  hollow:
```

```
    type: Polygon
    table:
    - [1.61e-3, 0]
    - [8.04e-4, 1.45e-3]
    - [-8.04e-4, 1.45e-3]
    - [-1.61e-3, 0]
    - [-8.04e-4, -1.45e-3]
    - [8.04e-4, -1.45e-3]
```

**Prompt for task Demultiplexer**

```
You are a helpful AI Assistant that provides well-reasoned and
↪  detailed responses.
You first think about the reasoning process as an internal monologue
↪  and then provide the user with the answer.
Respond in the following format: <think>
...
</think>
<answer>
...
</answer>.
## Task Description

You are a helpful AI Assistant and scientist with strong physical
↪  background and wonderful geometric designing ideas.
You are asked to generate the geometry design of a component using
↪  yaml files under certain constraints. You will first create
↪  geometries of your design, and then assign functions to the
↪  geometries according to the specific requirements.

Your final answer should contain a yaml file enclosed in
↪  ```yaml\n(your code)```. The yaml file should have at least two
↪  parts: geometry and selection. The specific requirements are as
↪  follow:

1. geometry: A list of objects with type and type-specific parameters.
↪  The types and parameters are as follows:
    Polygon: (2D) You can use it to create rectangles, triangles, etc.
        table:  Ordered list of n vertices as [x, y] points. The
        ↪  polygon is formed by **connecting consecutive points**
        ↪  (p_i->p_{i+1}) and **automatically closing** the shape
        ↪  (p_n->p_1).
        fillet: (Optional) A list of [i, r] tuples, where i is the
        ↪  index (starting from 1) of a polygon vertex defined in the
        ↪  above table, and r is the fillet radius for that
        ↪  corresponding vertex.
    Ellipse: (2D) You can use it to create circles.
        semiaxes: [horizontal, vertical] axis lengths
        pos: [center_x, center_y] center position
        rot: (Optional) Rotation angle (degree) counterclockwise
        angle: (Optional) Angular span (degree) counterclockwise. e.g.
        ↪  by setting angle=180 you can draw a upward semicircle.
    LineSegment: (1D)
        coord1: [start_x, start_y]
        coord2: [end_x, end_y]
    CircularArc: (1D)
        r: Radius
        angle1: Start angle (degree) counterclockwise, 0 degree
        ↪  represent positive direction of X-axis.
        angle2: End angle (degree) counterclockwise
    CubicBezier: (1D)
```

```
            p: Control points as [[x0,x1,x2,x3], [y0,y1,y2,y3]]
            w: Weight values as [w0,w1,w2,w3]
        InterpolationCurve: (1D)
            table: Ordered list of [x,y] points to interpolate through.
            ↪  The curve will pass every points smoothly (polynomial
            ↪  interpolation for x and y).
        ParametricCurve: (1D)
            parname: Name of parameter
            parmin: Minimum value of parameter
            parmax: Maximum value of parameter
            coord: Expressions about the parameter like ["expression_x",
            ↪  "expression_y"]. Trigonometric functions here use radians
        ConvertToSolid: (2D) Geometry formed by end-to-end connected 1D
        ↪  curves.
            geometries: A dictionary of 1D geometries (using the same
            ↪  structure as the top-level geometry section, recursive).
            ↪  **They Must connect end-to-end and form a simply connected
            ↪  space**.
        Union: (2D) Union of 2D geometries.
            geometries: A dictionary of geometries (recursive).
        Intersection: (2D) Intersection of 2D geometries.
            geometries: A dictionary of geometries (recursive).
        Difference: (2D) Difference of the 2D geometries.
            geometries_add: A dictionary of geometries to keep
            ↪  (recursive).
            geometries_subtract: A dictionary of geometries to subtract
            ↪  (recursive).

After **geometry** was created, the shapes will be splitted into
↪  **non-overlapping connected regions**.
    - Overlapping 2D shapes create new regions (e.g., two intersecting
    ↪  circles → 3 regions)
    - Enclosed 2D shapes split regions (e.g., circle inside polygon →
    ↪  2 regions: circle interior + polygon-ring)
    - 1D curves through 2D shapes create sub-regions (e.g., line
    ↪  segment through rectangle → alternating regions)
The **regions** can be represented by the following ways:
    - point: You can select an interior point of the region to
    ↪  represent it. The point should never on boundaries/corners.
    ↪  One point per region suffices.
    - geometry: The 2d shapes you created might be splitted into
    ↪  several regions. You can select the geometry to represent all
    ↪  the regions in it.

2. selection: After regions are created, you will assign different
↪  functions to regions using selections.
    UnionSelection: Union of all the regions selected below.
        points: (Optional) List of [x,y] points representing distinct
        ↪  regions.
        geometries: (Optional) List of 2d geometry names you created
        ↪  above. By listing geometries here, you can select all the
        ↪  region this geometry contains.
        selections: (Optional) List of other selection names you
        ↪  created.
    IntersectionSelection: Intersection of all the regions selected
    ↪  below.
        same parameters as UnionSelection
    DifferenceSelection: Select the regions in Add but not in
    ↪  Subtract.
        add: same parameters as UnionSelection.
        subtract: same parameters as UnionSelection.

Finally a yaml file will be like the following sample:
```

```yaml
geometry:
    uni1: # Name of this geometry
        type: Union
        geometries: # create geometries recursively below
            uni_el1: # Name of the first ellipse to union
                type: Ellipse # Type of this geometry
                semiaxes: [2.0, 1.0]  # Specific parameters
                pos: [1.0, 1.0]
            uni_pol1: # Name of the second polygon
                type: Polygon
                table:
                - [-1.0, -0.3]
                - [2.0, -1.0]
                - [1.0, 1.0]
    line1: # This line splits the ellipse into 2 regions.
        type: LineSegment
        coord1: [1.0, 2.0]
        coord2: [3.0, 1.0]

selection:
    sel1: # Name of this selection
        type: DifferenceSelection
        add:
            geometries:
            - uni1 # Select all the regions in uni1
        subtract:
            points:
            - [2.5, 1.5] # Remove the region where (2.5, 1.5) in. This
            ↪   region is part of ellipse but splitted by the line
            ↪   segment.
```

## Geometric Design of a 2D sound wave demultiplexer

You are asked to design a 2D sound wave demultiplexer. The
↪   demultiplexer takes incident sound wave from port 1, and omits
↪   sound wave at port 2 and 3. The objective is to maximize the
↪   difference of sound pressure (on log scale) at two outlet ports,
↪   which is calculated as $log10(port2.P_{out})-log10(port3.P_{out})$
↪   with $P_{out}$ being the sound pressure at the outlet ports.

The entire pressure acoustic region will be a circle of radius 0.1m
↪   centered at (0,0). The incident wave comes from the negative
↪   x-direction (9 o'clock). The sound waves are then ommited at 1
↪   o'clock (port 2) and 5 o'clock (port 3) of the acoustic region.

You should design void geometry (material to be removed) so that the
↪   sound wave will propagate through the remaining geometry and
↪   maximize the objective function. You should create a list of basic
↪   geometries and then select from them to form the void regions.
↪   **The Name of the selection must be `void`**. Keep in mind that
↪   the void geometry should stay inside the acoustic region and at
↪   least 0.15m away from the boundary of the acoustic region.

The constraints are as follow:
1. After removing the void geometry, the remaining part should still
↪   be connected.
2. The void geometry should stay inside the acoustic region, and at
↪   least 0.02m away from the boundary of the acoustic region.

The reward is calculated as follow:

1. 0 if constraints are violated.
2. $log10(port2.P_{out})-log10(port3.P_{out})$, the log scale pressure
   ↪  difference between port 2 and port 3, if constraints are
   ↪  satisfied.

## Example
An example solution is shown below. You should not copy the example
↪  solution, but you can refer to it to understand the task and
↪  create better ones. Feel free to add more basic geometries.

```yaml
geometry:
  barrier:
    type: ConvertToSolid
    geometries:
      cir_inner:
        type: CircularArc
        r: 0.08
        angle1: -120
        angle2: 0
      cir_outer:
        type: CircularArc
        r: 0.06
        angle1: -120
        angle2: 0
      line1:
        type: LineSegment
        coord1: [0.06, 0.0]
        coord2: [0.08, 0.0]
      line2:
        type: LineSegment
        coord1: [0.06*cos(-120*pi/180), 0.06*sin(-120*pi/180)]
        coord2: [0.08*cos(-120*pi/180), 0.08*sin(-120*pi/180)]
selection:
  void:
    type: UnionSelection
    geometries: [barrier]
```

### D.3  CIRCLE PACKING

**Prompt for task Circle Packing**

```
You are an expert software developer tasked with iteratively improving
↪  a codebase.
Your job is to analyze the current program and suggest improvements
↪  based on feedback from previous attempts.
Focus on making targeted changes that will increase the program's
↪  performance metrics.
Respond in the following format: <think>
...
</think>
<answer>
...
</answer>.
# Problem Description
```

```
You are an expert mathematician specializing in circle packing
↪   problems and computational geometry. Your task is to improve a
↪   constructor function that directly produces a specific arrangement
↪   of 26 circles in a unit square, maximizing the sum of their radii.
↪   The AlphaEvolve paper achieved a sum of 2.635 for n=26.

Key geometric insights:
- In dense regions, circles often follow hexagonal packing patterns,
↪   with the theoretical maximum density for infinite packing being
↪   pi/(2sqrt(3))=0.9069.
- However, when confined to a finite square, **edge effects** disrupt
↪   perfect symmetry and make pure hexagonal packing suboptimal.
- Optimal arrangements often require **variable-sized circles**, as
↪   this can improve space utilization compared to equal radii. Larger
↪   circles can be placed toward the center, with smaller circles
↪   strategically fitted near edges and corners.
- Effective designs may use **layered or shell-like patterns** rather
↪   than strict hexagonal grids. Hybrid approaches|combining regular
↪   arrangements in dense regions with adaptive adjustments near
↪   boundaries|are common in the densest known packings.
- The **optimization method** plays a critical role: physics-inspired
↪   simulations or algorithms with well-tuned parameters can yield
↪   better configurations than purely geometric intuition.
- Mathematical research indicates that for certain specific values of
↪   n, special arrangements can achieve unusually high densities.

You may either designing an explicit constructor of the result or
↪   explore search-based, optimization, or even multi-stage
↪   optimization methods, as long as they can finish running within 1
↪   minutes.

## Current Program
Status: {current_status}
```python
{current_program}
```

## Task
Suggest improvements to the program that will lead to better
↪   performance on the specified metrics.

You MUST use the exact SEARCH/REPLACE diff format shown below to
↪   indicate changes:

<<<<<<< SEARCH
# Original code to find and replace (must match exactly)
=======
# New replacement code
>>>>>>> REPLACE

You can suggest multiple changes. Each SEARCH section must **exactly**
↪   match code in the current program.
Be thoughtful about your changes and explain your reasoning
↪   thoroughly.

Make sure your rewritten program still contains `construct_packing()`
↪   function and maintains the same outputs. **You can
↪   add/delete/modify other functions arbitrarily.**

If you want to use new packages, please import them. Usable packages:
↪   scipy, sympy, shapely, pulp, cvxpy, nlopt, deap
```

```
If your code's execution time exceeds 1 minute, you will receive 0
↪   reward. Pay attention to the runtime efficiency!

IMPORTANT: Do not rewrite the entire program – focus on targeted
↪   improvements.
```

## D.4 FUNCTION MINIMIZATION

**Prompt for task Minimize Function**

```
# Problem Description

You are an expert in optimization algorithms. Your task is to improve
↪   a function minimization algorithm that minimizes a complex
↪   non-convex function with multiple local minima. The function is
↪   defined in {dimension}-dimensional space with the following
↪   expression:
```python
{formula}
```

## Current Program
Status: {current_status}
```python
{current_program}
```

## Task

Suggest improvements to the program that will lead to better
↪   performance on the specified metrics.

Your code's execution time should not exceed 10 seconds. Pay attention
↪   to the runtime efficiency!

You MUST use the exact SEARCH/REPLACE diff format shown below to
↪   indicate changes:

<<<<<<< SEARCH
# Original code to find and replace (must match exactly)
=======
# New replacement code
>>>>>>> REPLACE

Performance is evaluated using:
1. value_score: Closeness to minimum function value: |global_min| /
↪   (|global_min| + |found_value – global_min|)
2. distance_score: Proximity to true solution point: 1/(1 +
↪   distance_to_global_min)
3. standard_deviation_score: Solution stability across runs:
↪   (1/(1+std_x1) + 1/(1+std_x2) + ...)/dim
4. speed_score: Execution efficiency: min(1/avg_runtime_in_seconds,
↪   10)/10
5. reliability_score: successful_runs/total_runs. Successful run has
↪   no tracebacks and timeouts.
6. combined_score: **This is the final reward you received.** 100%
↪   value_score.

If you want to use new packages, please import them.
```

```
Make sure your rewritten program still contains `run_search()`
↪   function and maintains the same outputs. You can add/delete/modify
↪   other functions arbitrarily.

IMPORTANT: Do not rewrite the entire program – focus on targeted
↪   improvements.
```

## D.5 SYMBOLIC REGRESSION

**Prompt for Chemistry tasks**

```
You are an expert software developer. Your job is to write a Python
↪   function based on feedback from previous attempts.
Write your code in exactly the following format:
```python
# your code
```
Your code's execution time is limited, so pay attention to runtime
↪   efficiency!
If you use new packages, please import them.
Ensure the program still contains the func() function and produces the
↪   same outputs; other functions can be added, deleted, or modified
↪   freely.
IMPORTANT: The current task is a symbolic regression problem. Write a
↪   Python expression in func() where parameter scales are as similar
↪   as possible (use linear scaling or translation if needed). This
↪   helps later optimization when all parameters are initialized
↪   randomly in [0,1].
Respond in the following format: <think>
...
</think>
<answer>
...
</answer>.
Your task is to evolve a Python function `func(x, params)` that models
↪   a scientific process, considering the physical meaning and
↪   relationships of inputs, by predicting output variables based on
↪   input variables.

The function signature is:

```python
def func(x: np.ndarray, params: np.ndarray) -> np.ndarray:
```

- `x` is a 2D NumPy array of shape `(n_samples, 2)`
- `params` is a 1D NumPy array of up to 10 parameters
- The function should return a 1D NumPy array of predictions with
↪   shape `(n_samples,)`

**Current Problem:**
Model the dA_dt (Rate of change of concentration in chemistry reaction
↪   kinetics) using the input features: t (Time), A (Concentration at
↪   time t)
Thus, `x` contains 2 columns: t (Time), A (Concentration at time t).

The initial version of `func` is a simple linear model. Parameters in
↪   `params` will be optimized externally using the BFGS algorithm
↪   based on unseen training data.

Your objective is to evolve `func` to improve predictive performance
↪   on unseen data. Aim for a balance between:
```

- **Accuracy**: Lower mean squared error (MSE) on training data
- **Simplicity**: Prefer concise, interpretable expressions

Model performance (score = -log_10(mse)) will be evaluated on a
↪  held-out dataset. Ensure the model is free of potential numerical
↪  errors (e.g., log0, division by 0).
## Current Program
Status: Initial Program
```python
def func(x, params):
    """
    Calculates the model output using a linear combination of input
    ↪  variables
    or a constant value if no input variables. Operates on a matrix of
    ↪  samples.

    Args:
        x (np.ndarray): A 2D numpy array of input variable values,
        ↪  shape (n_samples, n_features).
                        n_features is 2.
                        If n_features is 0, x should be shape
                        ↪  (n_samples, 0).
                        The order of columns in x must correspond to:
                        (t, A).
        params (np.ndarray): A 1D numpy array of parameters.
                             Expected length: 10.

    Returns:
        np.ndarray: A 1D numpy array of predicted output values, shape
        ↪  (n_samples,).
    """
    result = x[:, 0] * params[0] + x[:, 1] * params[1]
    return result
```

**Prompt for Biology tasks**

You are an expert software developer. Your job is to write a Python
↪  function based on feedback from previous attempts.
Write your code in exactly the following format:
```python
# your code
```
Your code's execution time is limited, so pay attention to runtime
↪  efficiency!
If you use new packages, please import them.
Ensure the program still contains the func() function and produces the
↪  same outputs; other functions can be added, deleted, or modified
↪  freely.
IMPORTANT: The current task is a symbolic regression problem. Write a
↪  Python expression in func() where parameter scales are as similar
↪  as possible (use linear scaling or translation if needed). This
↪  helps later optimization when all parameters are initialized
↪  randomly in [0,1].
Respond in the following format: <think>
...
</think>
<answer>
...
</answer>.

Your task is to evolve a Python function `func(x, params)` that models
↪  a scientific process, considering the physical meaning and
↪  relationships of inputs, by predicting output variables based on
↪  input variables.

The function signature is:

```python
def func(x: np.ndarray, params: np.ndarray) -> np.ndarray:
```

- `x` is a 2D NumPy array of shape `(n_samples, 2)`
- `params` is a 1D NumPy array of up to 10 parameters
- The function should return a 1D NumPy array of predictions with
↪  shape `(n_samples,)`

**Current Problem:**
Model the dP_dt (Population growth rate) using the input features: t
↪  (Time), P (Population at time t)
Thus, `x` contains 2 columns: t (Time), P (Population at time t).

The initial version of `func` is a simple linear model. Parameters in
↪  `params` will be optimized externally using the BFGS algorithm
↪  based on unseen training data.

Your objective is to evolve `func` to improve predictive performance
↪  on unseen data. Aim for a balance between:
- **Accuracy**: Lower mean squared error (MSE) on training data
- **Simplicity**: Prefer concise, interpretable expressions

Model performance (score = -log_10(mse)) will be evaluated on a
↪  held-out dataset. Ensure the model is free of potential numerical
↪  errors (e.g., log0, division by 0).
## Current Program
Status: Initial Program
```python
def func(x, params):
    """
    Calculates the model output using a linear combination of input
    ↪  variables
    or a constant value if no input variables. Operates on a matrix of
    ↪  samples.

    Args:
        x (np.ndarray): A 2D numpy array of input variable values,
        ↪  shape (n_samples, n_features).
                        n_features is 2.
                        If n_features is 0, x should be shape
                        ↪  (n_samples, 0).
                        The order of columns in x must correspond to:
                        (t, P).
        params (np.ndarray): A 1D numpy array of parameters.
                             Expected length: 10.

    Returns:
        np.ndarray: A 1D numpy array of predicted output values, shape
        ↪  (n_samples,).
    """
    result = x[:, 0] * params[0] + x[:, 1] * params[1]
    return result
```

**Prompt for Physics tasks**

```
You are an expert software developer. Your job is to write a Python
↪  function based on feedback from previous attempts.
Write your code in exactly the following format:
```python
# your code
```

Your code's execution time is limited, so pay attention to runtime
↪  efficiency!
If you use new packages, please import them.
Ensure the program still contains the func() function and produces the
↪  same outputs; other functions can be added, deleted, or modified
↪  freely.
IMPORTANT: The current task is a symbolic regression problem. Write a
↪  Python expression in func() where parameter scales are as similar
↪  as possible (use linear scaling or translation if needed). This
↪  helps later optimization when all parameters are initialized
↪  randomly in [0,1].
Respond in the following format: <think>
...
</think>
<answer>
...
</answer>.
Your task is to evolve a Python function `func(x, params)` that models
↪  a scientific process, considering the physical meaning and
↪  relationships of inputs, by predicting output variables based on
↪  input variables.

The function signature is:

```python
def func(x: np.ndarray, params: np.ndarray) -> np.ndarray:
```

- `x` is a 2D NumPy array of shape `(n_samples, 3)`
- `params` is a 1D NumPy array of up to 10 parameters
- The function should return a 1D NumPy array of predictions with
↪  shape `(n_samples,)`

**Current Problem:**
Model the dv_dt (Acceleration in Nonl-linear Harmonic Oscillator)
↪  using the input features: x (Position at time t), t (Time), v
↪  (Velocity at time t)
Thus, `x` contains 3 columns: x (Position at time t), t (Time), v
↪  (Velocity at time t).

The initial version of `func` is a simple linear model. Parameters in
↪  `params` will be optimized externally using the BFGS algorithm
↪  based on unseen training data.

Your objective is to evolve `func` to improve predictive performance
↪  on unseen data. Aim for a balance between:
- **Accuracy**: Lower mean squared error (MSE) on training data
- **Simplicity**: Prefer concise, interpretable expressions

Model performance (score = -log_10(mse)) will be evaluated on a
↪  held-out dataset. Ensure the model is free of potential numerical
↪  errors (e.g., log0, division by 0).
## Current Program
Status: Initial Program
```python
def func(x, params):
```

```
    """
    Calculates the model output using a linear combination of input
    ↪  variables
    or a constant value if no input variables. Operates on a matrix of
    ↪  samples.

    Args:
        x (np.ndarray): A 2D numpy array of input variable values,
        ↪  shape (n_samples, n_features).
                        n_features is 3.
                        If n_features is 0, x should be shape
                        ↪  (n_samples, 0).
                        The order of columns in x must correspond to:
                        (x, t, v).
        params (np.ndarray): A 1D numpy array of parameters.
                             Expected length: 10.

    Returns:
        np.ndarray: A 1D numpy array of predicted output values, shape
        ↪  (n_samples,).
    """
    result = x[:, 0] * params[0] + x[:, 1] * params[1] + x[:, 2] *
    ↪  params[2]
    return result
```

**Prompt for Material Science tasks**

```
You are an expert software developer. Your job is to write a Python
↪  function based on feedback from previous attempts.
Write your code in exactly the following format:
```python
# your code
```

Your code's execution time is limited, so pay attention to runtime
↪  efficiency!
If you use new packages, please import them.
Ensure the program still contains the func() function and produces the
↪  same outputs; other functions can be added, deleted, or modified
↪  freely.
IMPORTANT: The current task is a symbolic regression problem. Write a
↪  Python expression in func() where parameter scales are as similar
↪  as possible (use linear scaling or translation if needed). This
↪  helps later optimization when all parameters are initialized
↪  randomly in [0,1].
Respond in the following format: <think>
...
</think>
<answer>
...
</answer>.
Your task is to evolve a Python function `func(x, params)` that models
↪  a scientific process, considering the physical meaning and
↪  relationships of inputs, by predicting output variables based on
↪  input variables.

The function signature is:

```python
def func(x: np.ndarray, params: np.ndarray) -> np.ndarray:
```
```

```
- `x` is a 2D NumPy array of shape `(n_samples, 2)`
- `params` is a 1D NumPy array of up to 10 parameters
- The function should return a 1D NumPy array of predictions with
↪   shape `(n_samples,)`

**Current Problem:**
Model the sigma (Stress) using the input features: epsilon (Strain), T
↪   (Temperature)
Thus, `x` contains 2 columns: epsilon (Strain), T (Temperature).

The initial version of `func` is a simple linear model. Parameters in
↪   `params` will be optimized externally using the BFGS algorithm
↪   based on unseen training data.

Your objective is to evolve `func` to improve predictive performance
↪   on unseen data. Aim for a balance between:
- **Accuracy**: Lower mean squared error (MSE) on training data
- **Simplicity**: Prefer concise, interpretable expressions

Model performance (score = -log_10(mse)) will be evaluated on a
↪   held-out dataset. Ensure the model is free of potential numerical
↪   errors (e.g., log0, division by 0).
## Current Program
Status: Initial Program
```python
def func(x, params):
    """
    Calculates the model output using a linear combination of input
    ↪   variables
    or a constant value if no input variables. Operates on a matrix of
    ↪   samples.

    Args:
        x (np.ndarray): A 2D numpy array of input variable values,
        ↪   shape (n_samples, n_features).
                        n_features is 2.
                        If n_features is 0, x should be shape
                        ↪   (n_samples, 0).
                        The order of columns in x must correspond to:
                        (epsilon, T).
        params (np.ndarray): A 1D numpy array of parameters.
                             Expected length: 10.

    Returns:
        np.ndarray: A 1D numpy array of predicted output values, shape
        ↪   (n_samples,).
    """
    result = x[:, 0] * params[0] + x[:, 1] * params[1]
    return result
```
```

# E   METHODOLOGICAL CHALLENGES AND COMPARATIVE ANALYSIS OF RL-EA INTEGRATION

This appendix details the specific technical challenges associated with integrating Reinforcement Learning (RL) and Evolutionary Algorithms (EAs). We further analyze why a naive sequential combination (e.g., AlphaEvolve) fails to scale effectively compared to the proposed HELIX framework, supported by empirical evidence from the Circle Packing problem.

## E.1 TECHNICAL CHALLENGES AND SOLUTIONS

The integration of RL and EAs presents several non-trivial challenges, primarily stemming from the conflicting objectives and operational domains of the two paradigms. HELIX addresses these as follows:

**Goal Mismatch and Unification.** A fundamental disconnect exists between RL, which learns a policy mapping states to actions, and EAs, which act as population-based optimization methods relying on recombination and mutation. Integrating these requires a principled bridge rather than a naive combination.

- **In-Context Learning as a Bridge:** HELIX adopts an in-context learning paradigm where previously discovered high-quality solutions are injected into the prompt as explicit memory. This transforms the Large Language Model (LLM) into a *parameterized mutation operator*, conditioned on historical trajectories.
- **Unified Optimization:** We employ Group Relative Policy Optimization (GRPO) to train this mutation operator. GRPO naturally generates diverse rollouts that serve as the population for evolutionary selection. Consequently, policy optimization (RL) and evolutionary search (EA) are coupled within a closed loop: test-time scaling via evolution provides high-quality data for RL, while RL improves the mutation operator for subsequent evolutionary steps.

**Diversity Estimation in Giant Code Spaces.** Traditional evolutionary metrics are ill-suited for code optimization, where the action space is discrete, high-dimensional, and highly structured. Measuring individual diversity in this domain is challenging. HELIX resolves this by utilizing an embedding-based approach to quantify semantic distances between code individuals. We compute population diversity via k-nearest neighbors (kNN) in this embedding space, providing a scalable and semantically meaningful metric to guide selection.

## E.2 LIMITATIONS OF NAIVE INTEGRATION: A CASE STUDY

To demonstrate why HELIX offers a necessary advancement over "naive" integration, we compare it against the AlphaEvolve paradigm. AlphaEvolve represents a sequential approach: post-training an LLM on general domains followed by applying evolutionary algorithms to downstream tasks without further policy updates.

We conducted a comparative experiment on the *Circle Packing* problem (maximizing the sum of radii for 26 non-overlapping circles in a unit square). We evaluated Direct Prompting (BO64), OpenEvolve (an open-source reproduction of AlphaEvolve), and HELIX using Qwen 14B and 32B base models.

Table 2: Performance Comparison on Circle Packing Task

| Method | Score (Sum of Radii) |
| --- | --- |
| Direct Prompt (Qwen 14B) | 1.673 |
| OpenEvolve (Qwen 14B) | 1.586 |
| OpenEvolve (Qwen 32B) | 1.956 |
| **HELIX (Qwen 14B)** | **2.636** |

As shown in Table 2, OpenEvolve with Qwen 14B performed worse than the Direct Prompt baseline, despite utilizing significantly more computational resources. Our analysis identifies two critical failure modes in naive integration of OpenEvolve:

**1. Constraints of Initialization Bias.** Naive approaches are heavily constrained by their initial seed solutions. OpenEvolve generates a small set of seed trials (e.g., 5) and iterates upon them. If these initial trials lack diversity or occupy a low-performance region, the evolutionary process stagnates in local optima. In contrast, Direct Prompting (BO64) benefits from 64 i.i.d. evaluations, offering a broader initial coverage that the naive evolutionary process failed to surpass.

**2. Rejection of Novelty and Destructive Changes.** A more subtle failure mode is the rejection of potentially high-reward strategies that require initial "destructive" changes. In our Qwen 32B OpenEvolve experiment (4,147 trials), the model predominantly attempted to adjust coordinates directly. Only 12 trials (0.3%) attempted a radically different approach using `scipy.optimize.minimize`.

- **The Failure:** All 12 trials initially yielded a reward of $0.0$ due to minor compilation errors or timeouts. Traditional evolutionary selection, driven by immediate reward or superficial code features (e.g., length), discarded these candidates.

- **The Consequence:** The system failed to explore the `scipy` approach, which—once debugged—is capable of yielding scores $> 2.0$.

### E.3   THE HELIX ADVANTAGE

HELIX overcomes the aforementioned limitations through two specific mechanisms:

1. **Explicit Diversity Accounting:** By using an embedding model to distinguish methods semantically, HELIX assigns a high *Diversity Score* to the rare `scipy`-based solutions, even when their immediate reward is low. This ensures they are retained in the population for further mutation/debugging.

2. **Parameter Learning via RL:** Once a diversity-preserved rollout successfully fixes the implementation bug (generating a high-reward solution $s_{t+1}^\star$), HELIX utilizes this trajectory for RL updates. This update increases the probability of the policy generating similar sophisticated methods in future steps.

This establishes a positive feedback loop: diversity metrics preserve potential innovation, and RL consolidates successful realizations of that innovation into the model parameters, allowing HELIX to break out of local optima where naive methods stagnate.

## F   THEORETICAL ANALYSIS OF THE FRAMEWORK

In this section, we employ a simplified mathematical model to provide theoretical insights into the advantages of our algorithm in solving complex scientific problems. We demonstrate the efficiency of HELIX in discovering optimal solutions compared to baseline methods.

### F.1   MATHEMATICAL SETUP AND PRELIMINARIES

First, we establish the geometric and probabilistic foundations of the problem. For a given problem $q$, we assume the existence of the following structures:

- **Solution Space:** A set of solutions $\mathcal{S}$, which can be viewed as a simply connected open manifold in a complex space.

- **Reward Function:** A continuous function $R : \mathcal{S} \to \mathbb{R}^+$.

- **Embedding:** A mapping $\Phi : \mathcal{S} \to \mathbb{R}^n$ that maps the solution space to an Embedding Space $\mathbb{R}^n$, satisfying:

    1. **Continuity:** For any $s_1, s_2 \in \mathcal{S}$ derived via similar methods, their embeddings $v_1 = \Phi(s_1)$ and $v_2 = \Phi(s_2)$ are adjacent in $\mathbb{R}^n$.
    2. **Injectivity:** Distinct solutions have distinct embeddings.
    3. **Open Map:** $\Phi$ maps open sets in $\mathcal{S}$ to open sets in $\mathbb{R}^n$.

We define the reward function in the embedding space $\mathbb{R}^n$ as follows. For any $v \in \mathbb{R}^n$:

$$r(v) = \begin{cases} R(\Phi^{-1}(v)) & v \in \Phi(\mathcal{S}) \\ 0 & v \notin \Phi(\mathcal{S}) \end{cases}, \tag{29}$$

where $s = \Phi^{-1}(v)$ is the solution corresponding to $v$. Restricted to the image set, $\Phi : \mathcal{S} \to \Phi(\mathcal{S})$ is a bijection, making its inverse well-defined. Given the continuity of $\Phi$ and $R$, and the open mapping property of $\Phi$, $r(v)$ is continuous.

**Definition F.1** (LLM Transition Process). For a solution $s$, an LLM parameterized by $\theta$ transforms the solution by outputting an action $a \sim \pi_\theta(\cdot|s)$, resulting in $s' = T(s, a)$. Based on this, we define the LLM Transition Function on $\mathbb{R}^n$ as a stochastic process:

$$T_\theta^\star : \mathbb{R}^n \to (\Omega \to \mathbb{R}^n), \quad T_\theta^\star(v) = \Phi(T(\Phi^{-1}(v), a)), \quad \text{where } a \sim \pi_\theta(\cdot|\Phi^{-1}(v)). \tag{30}$$

For tractability, we approximate $T_\theta^\star$ as an independent Normal distribution:

$$T_\theta^\star(v) \sim \mathcal{N}(v + \delta_\theta(v), \sigma\mathbf{I}). \tag{31}$$

This implies each transition follows $v \to v + \delta_\theta(v) + \xi$, where $\xi \sim \mathcal{N}(0, \sigma\mathbf{I})$. This Gaussian approximation is justified as LLMs typically generate modest modifications to the current solution, making local approximations valid in the embedding space.

### F.2 THEORETICAL ANALYSIS OF GRPO

#### F.2.1 SETUP AND ASSUMPTIONS

In the GRPO method, since the prompt is fixed, the model evolves solely from an initial solution $v_0$. The transition samples from $\mathcal{N}(v_0 + \delta_\theta(v_0), \sigma\mathbf{I})$. GRPO estimates the gradient of the reward function near $v = v_0 + \delta_\theta(v_0)$ and updates the model parameters. The effective update dynamics in the embedding space can be described as:

$$\delta_\theta(v_0) \leftarrow \delta_\theta(v_0) + \eta\nabla_v r(v_0 + \delta_\theta(v_0)), \tag{32}$$

which simplifies to the gradient ascent process $v \leftarrow v + \eta\nabla_v r(v)$, where $\eta$ is the learning rate.

To analyze convergence, we introduce the following assumption regarding the reward landscape.

**Assumption F.2** (Reward Landscape). We assume the reward function $r(v)$ consists of two Gaussian peaks, representing a local optimum ($v_{loc}$) and a global optimum ($v_{opt}$):

$$r(v) = A_{loc} \exp\left(-\frac{\|v - v_{loc}\|^2}{2w^2}\right) + A_{opt} \exp\left(-\frac{\|v - v_{opt}\|^2}{2w^2}\right). \tag{33}$$

Let $L = \|v_{opt} - v_{loc}\|$ be the distance between the optima.

**Theorem F.3** (Convergence to Local Optimum of GRPO). *Let $v_0$ be the initial solution for GRPO. GRPO will converge to the local optimum near $v_{loc}$ if the following conditions are met:*

1. **Separation:** $L > 2w$. *There is sufficient separation between the global and local optima.*

2. **Amplitude:** $A_{loc} > A_{opt} \cdot \frac{L}{w} \cdot \exp\left(-\frac{L^2}{2w^2}\right)$. *The local optimum is not significantly weaker than the global optimum locally.*

3. **Initialization:** *Decomposing the initial solution as $v_0 = v_{loc} + v_\perp + \gamma_0(v_{opt} - v_{loc})$, where $v_\perp \perp (v_{opt} - v_{loc})$, we require $\gamma_0 < \gamma_{barrier} \approx \frac{1}{2} - \frac{w^2}{L^2}\ln\frac{A_{opt}}{A_{loc}}$. This implies the initial solution is geometrically closer to the local optimum's basin of attraction.*

*Comment.* These assumptions hold in many scientific problems where distinct methods (local vs. global) have a large semantic gap ($L > 2w$), and initial "naive" solutions naturally fall closer to simpler local optima. This illustrates that GRPO, lacking memory or population mechanisms, is prone to trapping in local optima.

#### F.2.2 PROOF OF THEOREM F.3

We decompose the gradient of $r(v)$. Let $v$ be parameterized as $v = v_{loc} + v_\perp + \gamma(v_{opt} - v_{loc})$. The gradient $\nabla r(v)$ satisfies:

$$\nabla r(v) = \frac{A_{loc}}{w^2}(v_{loc} - v)\exp\left(-\frac{\|v - v_{loc}\|^2}{2w^2}\right) + \frac{A_{opt}}{w^2}(v_{opt} - v)\exp\left(-\frac{\|v - v_{opt}\|^2}{2w^2}\right). \tag{34}$$

Projecting onto the line connecting the optima $(v_{opt} - v_{loc})$:

$$\langle \nabla r(v), v_{opt} - v_{loc} \rangle = -\frac{A_{loc}L^2}{w^2}\gamma \exp\left(-\frac{\gamma^2 L^2 + \|v_\perp\|^2}{2w^2}\right)$$
$$+ \frac{A_{opt}L^2}{w^2}(1-\gamma)\exp\left(-\frac{(1-\gamma)^2 L^2 + \|v_\perp\|^2}{2w^2}\right). \tag{35}$$

Projecting onto the perpendicular component $v_\perp$:

$$\langle \nabla r(v), v_\perp \rangle = -\frac{\|v_\perp\|^2}{w^2}\left(A_{loc}\left(-\frac{\|v - v_{loc}\|^2}{2w^2}\right) + A_{opt}\left(-\frac{\|v - v_{opt}\|^2}{2w^2}\right)\right). \tag{36}$$

First, analyzing the dynamics of $v_\perp$:

$$\frac{d}{dt}\|v_\perp\|^2 = 2\langle \dot{v}_\perp, v_\perp \rangle \propto -C(v)\|v_\perp\|^2 < 0. \tag{37}$$

Regardless of initialization, $v_\perp$ decays exponentially to 0. The system converges to the linear manifold connecting $v_{loc}$ and $v_{opt}$. Assuming $v_\perp = 0$, the dynamics of $\gamma$ are governed by:

$$\frac{d\gamma}{dt} \propto -\gamma A_{loc}\exp\left(-\frac{\gamma^2 L^2}{2w^2}\right) + (1-\gamma)A_{opt}\exp\left(-\frac{(1-\gamma)^2 L^2}{2w^2}\right). \tag{38}$$

Solving for equilibrium points ($\frac{d\gamma}{dt} = 0$) yields a stable local equilibrium near $\gamma \approx 0$, an unstable saddle point $\gamma_{barrier} \approx \frac{1}{2} - \frac{w^2}{L^2}\ln\frac{A_{opt}}{A_{loc}}$, and a stable global equilibrium near $\gamma \approx 1$. If $\gamma_0 < \gamma_{barrier}$, the system flows to the local optimum. $\qquad\square$

## F.3 THEORETICAL ANALYSIS OF EVOLVE AND HELIX

### F.3.1 SETUP: UNIFIED DRIFT-DIFFUSION AND SELECTION FRAMEWORK

We analyze the iterative processes of Evolve and HELIX by modeling them as continuous-time stochastic processes. Both algorithms maintain a population $\mathcal{P}$ and update it via $v'$.

- **Evolve (Selection-Diffusion):** In the Evolve algorithm, the model parameters cannot be adjusted. Thus, we assume the model has no inherent directional bias towards different methods of this specific problem ($\delta_\theta(v) \equiv 0$). The iteration simplifies to a random walk $v' = v + \sigma\xi$. At each step, a solution $v$ is drawn from $\mathcal{P}$, and $v' = v + \sigma\xi$ is generated. Critically, solutions with higher Reward are sampled with a higher probability. We can model this selection by a weight function $w(v) = \exp(\alpha r(v))$, where $\alpha$ represents the selection pressure. The new solution is added to the population: $\mathcal{P} \leftarrow \mathcal{P} \cup \{v'\}$.

- **HELIX (Drift-Diffusion):** HELIX maintains a population $\mathcal{P}$ and dynamically adjusts the directional bias $\delta_\theta(v)$. Through the GRPO mechanism, this direction will gradually approximate the gradient $\nabla r(v)$. Upon sufficient convergence, the HELIX iteration approximates a guided random walk: $v' = v + \eta\nabla r(v) + \sigma\xi$. In HELIX, we also sample high-Reward solutions with higher probability, but for mathematical tractability, we assume the selection weight parameter $\alpha = 0$, meaning the sampling weight is uniform ($w(v) \equiv 1$).

The comparison is summarized in Table 3.

Table 3: Comparison of Algorithm Dynamics

| Algorithm | Dynamics Equation | Drift $D(v)$ | Selection $w(v)$ |
|-----------|-------------------|--------------|------------------|
| Evolve | $v' = v + \sigma\xi$ | $\mathbf{0}$ | $\exp(\alpha r(v))$ |
| HELIX | $v' = v + \eta\nabla r(v) + \sigma\xi$ | $\eta\nabla r(v)$ | $1\ (\alpha = 0)$ |

**Theorem F.4** (Stationary Distribution). *Assuming the solution space is bounded, as $t \to \infty$, the population distribution $p^*(\mathbf{v})$ converges to:*

1. **Evolve:** *Converges to the principal eigenfunction of the associated Schrödinger operator. Under the WKB approximation ($\sigma \to 0$):*

$$p^*_{Evo}(\mathbf{v}) \asymp \exp\left(-\frac{\sqrt{2\alpha}}{\sigma} \int_{v_{opt}}^{v} \sqrt{r(v_{opt}) - r(u)}du\right). \tag{39}$$

2. **HELIX:** *Converges to a Boltzmann-Gibbs Measure:*

$$p^*_{Helix}(\mathbf{v}) \propto \exp\left(\frac{2\eta}{\sigma^2}r(\mathbf{v})\right). \tag{40}$$

**Comment on theorem F.4.**

**1. Concentration and $\sigma$ Scaling.** The concentration power of the stationary distributions—defined as the inverse of their variance—exhibits distinct scaling behaviors with respect to the noise parameter $\sigma$. Specifically, the concentration scales as $\mathcal{O}(1/\sigma)$ for Evolve and $\mathcal{O}(1/\sigma^2)$ for HELIX. Given that $\sigma \ll 1$ in high-precision search contexts, it follows that $1/\sigma^2 \gg 1/\sigma$. This inequality indicates that the sampling distribution of HELIX is exponentially more concentrated around the optimum than that of Evolve. Under identical environmental conditions, HELIX achieves a significantly more exhaustive exploration of the highly rewarded vicinity of the optimal solution.

**2. Intuitive Comparison (Quadratic Reward).** To provide a concrete comparison, we analyze the behavior under a local quadratic approximation of the reward function, $r(v) = r_{opt} - \frac{k}{2}\|v\|^2$ (centered at $v_{opt} = 0$). Deriving the exact Gaussian forms of the stationary distributions allows for a direct comparison of their variances, as summarized in Table 4.

Table 4: Comparison of Stationary Distributions under Quadratic Reward

| Algorithm | Gaussian Form $p^*(\mathbf{v})$ | Variance $\Sigma^2$ | Scaling vs. $\sigma$ |
|---|---|---|---|
| **HELIX** | $\propto \exp\left(-\frac{\eta k}{\sigma^2}\|v\|^2\right)$ | $\Sigma^2_{Helix} = \frac{\sigma^2}{2\eta k}$ | $\propto \sigma^2$ (Sharper) |
| **Evolve** | $\propto \exp\left(-\frac{\sqrt{\alpha k}}{2\sigma}\|v\|^2\right)$ | $\Sigma^2_{Evo} = \frac{\sigma}{\sqrt{\alpha k}}$ | $\propto \sigma$ (Broader) |

The ratio of their variances is given by:

$$\frac{\Sigma^2_{Helix}}{\Sigma^2_{Evo}} = \frac{\sigma^2/2\eta k}{\sigma/\sqrt{\alpha k}} = \frac{\sqrt{\alpha}}{2\eta\sqrt{k}} \cdot \sigma \propto \sigma. \tag{41}$$

As $\sigma \to 0$, this ratio tends to zero. This rigorously confirms that HELIX's mechanism—utilizing the gradient for directional movement—provides a superior capacity for stabilizing and concentrating the population compared to Evolve's reliance on scalar selection alone.

**3. Potential for Further Reinforcement.** It is worth noting that the current analysis assumes a uniform selection weight for HELIX ($\alpha = 0$). If we were to incorporate a non-trivial selection weight $w(v) = \exp(\alpha r(v))$ into the HELIX framework, the final stationary distribution would theoretically become even more concentrated. Although a quantitative closed-form solution for this combined Drift-Diffusion-Selection process is mathematically intractable, qualitative analysis suggests that this would further reinforce HELIX's focus and exploitation capabilities within high-reward regions.

#### F.3.2 Proof of Theorem F.4

**Part I: HELIX (Drift-Diffusion).** The dynamics follow the Langevin Equation $d\mathbf{v}_t = \eta\nabla r(\mathbf{v}_t)dt + \sigma d\mathbf{W}_t$. The probability density $p(\mathbf{v}, t)$ evolves via the Fokker-Planck equation:

$$\frac{\partial p}{\partial t} = -\nabla \cdot (p \cdot \eta\nabla r(\mathbf{v})) + \frac{\sigma^2}{2}\nabla^2 p. \tag{42}$$

At steady state ($\partial p / \partial t = 0$), the probability flux $\mathbf{J}$ vanishes:

$$\mathbf{J} = \eta p^* \nabla r(\mathbf{v}) - \frac{\sigma^2}{2} \nabla p^* = \mathbf{0} \implies \frac{\nabla p^*}{p^*} = \frac{2\eta}{\sigma^2} \nabla r(\mathbf{v}). \tag{43}$$

Integrating both sides yields $\ln p^*(\mathbf{v}) = \frac{2\eta}{\sigma^2} r(\mathbf{v}) + C$, confirming the Boltzmann distribution.

**Part II: Evolve (Selection-Diffusion).** The discrete selection-mutation process converges to the Replicator-Mutator Equation in continuous time:

$$\frac{\partial p}{\partial t} = \frac{\sigma^2}{2} \nabla^2 p + \alpha \left( r(v) - \bar{r} \right) p. \tag{44}$$

The stationary distribution $p^*$ satisfies the Schrödinger-like equation (where $E = \alpha\bar{r}$):

$$\frac{\sigma^2}{2} \nabla^2 p^* + \alpha r(v) p^* = E p^*. \tag{45}$$

Using the WKB Ansatz $p^*(v) = C(v) \exp(-S(v)/\sigma)$, and substituting into the equation, the leading order terms ($\sigma \to 0$) yield the Hamilton-Jacobi equation:

$$\frac{1}{2} \|\nabla S\|^2 + \alpha r(v) \approx E. \tag{46}$$

Setting the ground state condition at $v_{opt}$ gives $E = \alpha r(v_{opt})$. Solving for $\nabla S$:

$$\|\nabla S(v)\| = \sqrt{2\alpha(r(v_{opt}) - r(v))}. \tag{47}$$

Integrating along the path from $v_{opt}$ gives the action $S(v)$, yielding the final asymptotic form for $p^*_{Evo}$. $\qquad\square$

## G  FORMALIZED ALGORITHM

In this appendix, we provide the detailed procedural description of the HELIX framework. Algorithm 1 summarizes the full workflow, including sampling, prompt construction, model rollout, reinforcement learning updates, diversity estimation, and evolutionary population selection. These details complement the main text and offer a complete specification of the method.

## H  EXAMPLE OF MODEL OUTPUT

We present examples of the best solutions found by our framework across different task categories. These visualizations highlight how HELIX generates high-quality and interpretable outputs in diverse scientific domains.

### H.1  PHYSICS SIMULATION TASKS

**Acoustic demultiplexer.** Figure 14 displays the acoustic pressure field of our best-performing demultiplexer, which achieves a reward of 14.260.

**Iron core torque optimization.** The best iron core design is shown in Figure 15, where the magnetic flux density norm reaches a reward of 11.045.

**Beam design.** Figure 16 illustrates the von Mises stress pattern of the best beam structure discovered, which achieves a reward of 17.298.

**Meta-material optimization.** The temperature distributions of the optimized meta-material under two loading conditions are presented in Figure 17, yielding a reward of 1.278.

**Inductor design.** The optimized inductor is visualized in Figure 18, with a magnetic flux density norm field corresponding to a reward of 9.609.

---

**Algorithm 1** HELIX Framework

---

**Require:** Problem description $p$; initial solution(s) $s_0$; batch size $B$; GRPO group size $G$; number of
samples in prompt $n$; transition function $T$; reward function $R$; feedback function $F$; embedding
model $E$.

1: Initialize dataset $\mathcal{D}_0 = \{s_0\}$.
2: Initialize population $\mathcal{P}_0 = \mathcal{D}_0$.
3: Initialize policy model parameters $\theta$.

4: **for** iteration $t = 0, 1, 2, \ldots$ **do**

5:     Sample $B$ solutions from $\mathcal{P}_t$, obtaining $\{s_{t,i}\}_{i=1}^{B}$.         ▷ Prompt Construction
6:     **for** $i = 1$ to $B$ **do**
7:         Retrieve $n$ ancestral states of $s_{t,i}$: $\{f^{(k)}(s_{t,i})\}_{k=1}^{n-1}$.
8:         Construct prompt:

$$q_i = \text{ConstructPrompt}(\{p\} \cup \{s_{t,i}, R(s_{t,i}), F(s_{t,i})\} \cup$$
$$\{f^{(k)}(s_{t,i}), R(f^{(k)}(s_{t,i})), F(f^{(k)}(s_{t,i}))\}_{k=1}^{n-1}).$$

9:     **end for**

10:     **for** $i = 1$ to $B$ **do**         ▷ Model Rollout and Evaluation
11:         **for** $j = 1$ to $G$ **do**
12:             Generate action $a_{i,j} \sim \pi_\theta(\cdot \mid q_i)$.
13:             Obtain new solution $s_{t+1,i,j} = T(s_{t,i}, a_{i,j})$.
14:             Evaluate reward $r_{t+1,i,j} = R(s_{t+1,i,j})$.
15:             Record feedback $f_{t+1,i,j} = F(s_{t+1,i,j})$.
16:         **end for**
17:     **end for**

18:     **for** $i = 1$ to $B$ **do**         ▷ Reinforcement Learning Update
19:         **for** $j = 1$ to $G$ **do**
20:             Compute normalized advantage:

$$\tilde{r}_{t+1,i,j} = \frac{r_{t+1,i,j} - \text{mean}_j\{r_{t+1,i,j}\}}{\text{std}_j\{r_{t+1,i,j}\}}.$$

21:         **end for**
22:     **end for**
23:     Update policy: $\theta \leftarrow \theta - \gamma \cdot \nabla_\theta \mathcal{L}_{\text{GRPO}}$.

24:     **for** $i = 1$ to $B$ **do**         ▷ Diversity Estimation
25:         **for** $j = 1$ to $G$ **do**
26:             Compute embedding $h_{t+1,i,j} = E(s_{t+1,i,j})$.
27:             Compute diversity score $\text{Div}(s_{t+1,i,j})$ (as in Eq. (6)).
28:         **end for**
29:     **end for**

30:     Update dataset: $\mathcal{D}_{t+1} \leftarrow \mathcal{D}_t \cup \{s_{t+1,i,j}\}$.         ▷ Population Update
31:     Use NSGA-II to select next population:

$$\mathcal{P}_{t+1} = \text{SelectTop}_{\text{NSGA-II}}\left( \bigcup_{0 \leq s \leq t+1} \mathcal{D}_s \right).$$

32: **end for**

---

## H.2   Circle packing tasks

**Packing in square.** As shown in Figure 19, our framework successfully packs 26 circles inside a
square, achieving a sum of radii of 2.6359830849 and surpassing the previous world record.

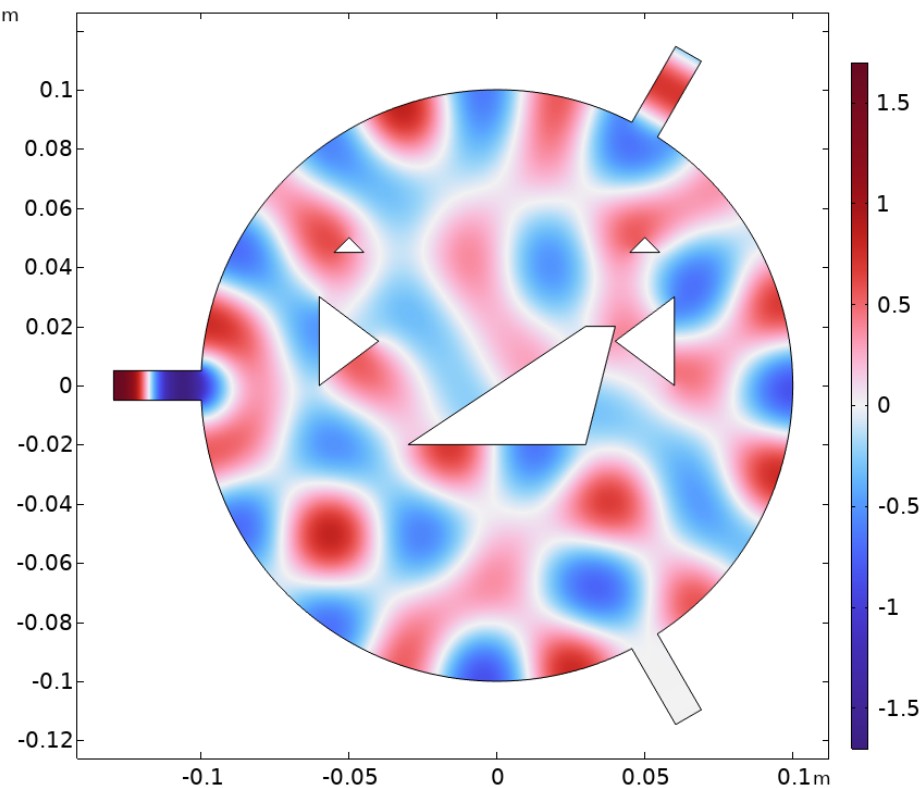

Figure 14: Acoustic pressure distribution in the optimal acoustic demultiplexer design obtained by our framework, with reward 14.260.

**Packing in disk.** Figure 20 demonstrates the packing of 26 circles inside a disk, reaching a total radius sum of 4.664465.

### H.3 MACHINE LEARNING TASKS

We further demonstrate how our framework can be applied to classical machine learning problems, using both classification and regression benchmarks. The first example focuses on the Adult dataset, where we design a rich set of engineered features that combine polynomial transformations, ratios, interaction terms, and domain-specific indicators. This structured feature space, coupled with a LightGBM classifier and hyperparameter tuning, enables our model to achieve a strong performance of 82.07 in macro F1 score (Figure 21).

For regression, we turn to the Transparent Conductor Task. Here, we integrate multi-scale geometric descriptors with spatial informatics derived from 3D Kernel Density Estimation and Principal Component Analysis. This framework transforms raw atomic coordinates into a comprehensive feature set capturing both local coordination environments and global structural topology. With these enhancements, our model coupled with an XGBoost regressor attains the second-highest score on the participants' leaderboard, corresponding to an RMSLE of 0.04915 (Figure 22).

## I LLM USAGE STATEMENT

Large Language Models (LLMs) were used solely to aid writing and polishing the manuscript. All research ideas, experiments, and analyses were conceived and conducted by the authors, who take full responsibility for the content.

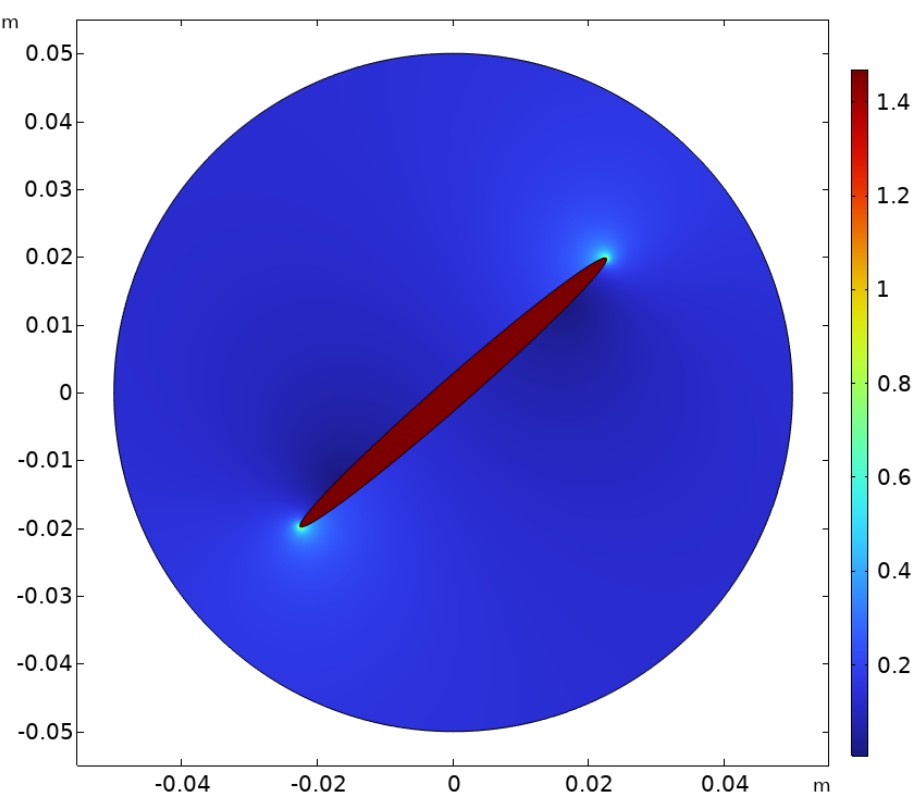

Figure 15: Magnetic flux density norm field for the optimized iron core configuration identified by our framework, achieving reward 11.045.

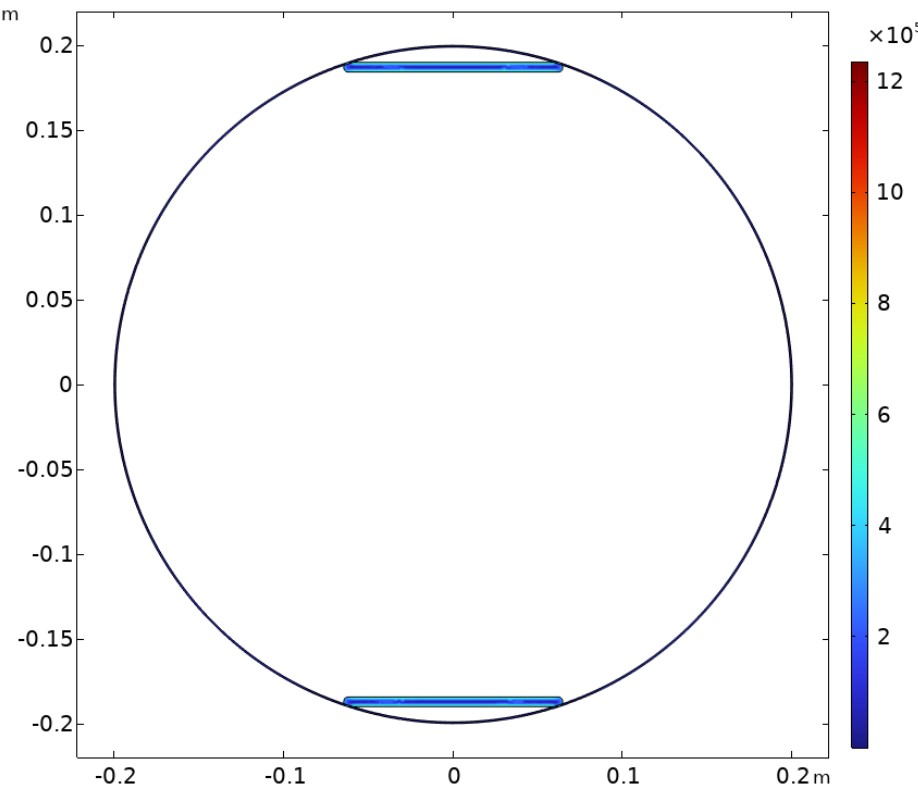

Figure 16: Von Mises stress distribution of the optimized beam design obtained by our framework, with reward 17.298.

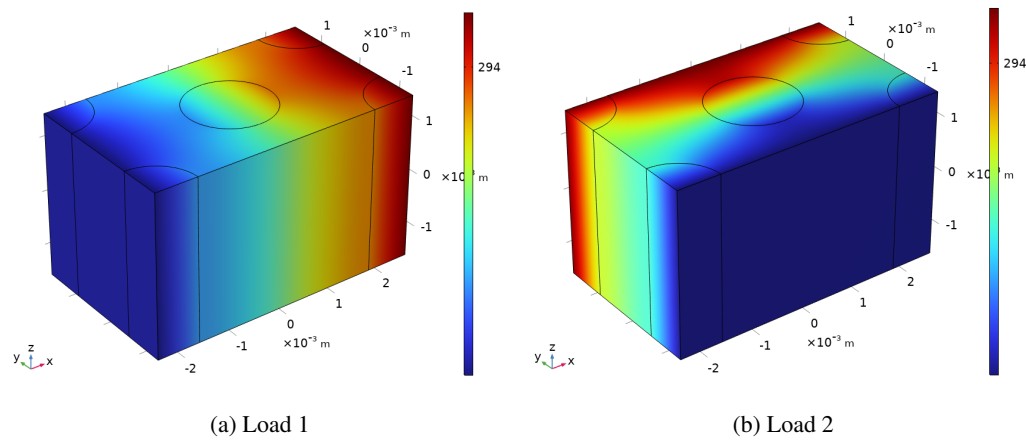

(a) Load 1          (b) Load 2

Figure 17: Optimized temperature fields of the meta-material designed by our framework under two different load conditions, achieving reward 1.278.

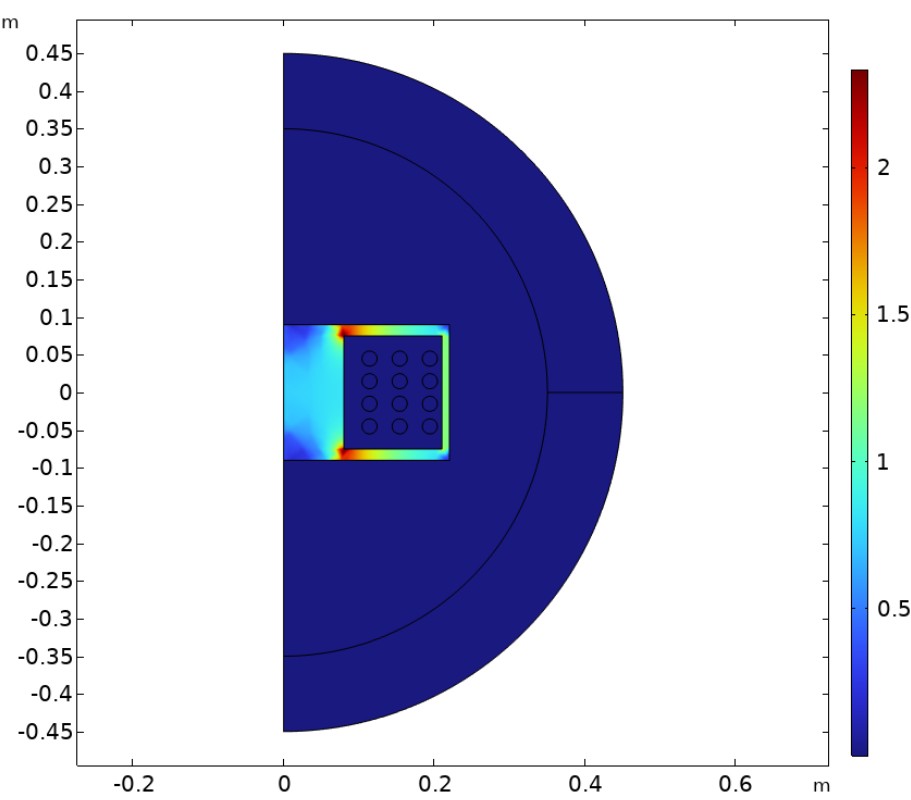

Figure 18: Magnetic flux density norm field of the best inductor configuration identified by our framework, achieving reward 9.609.

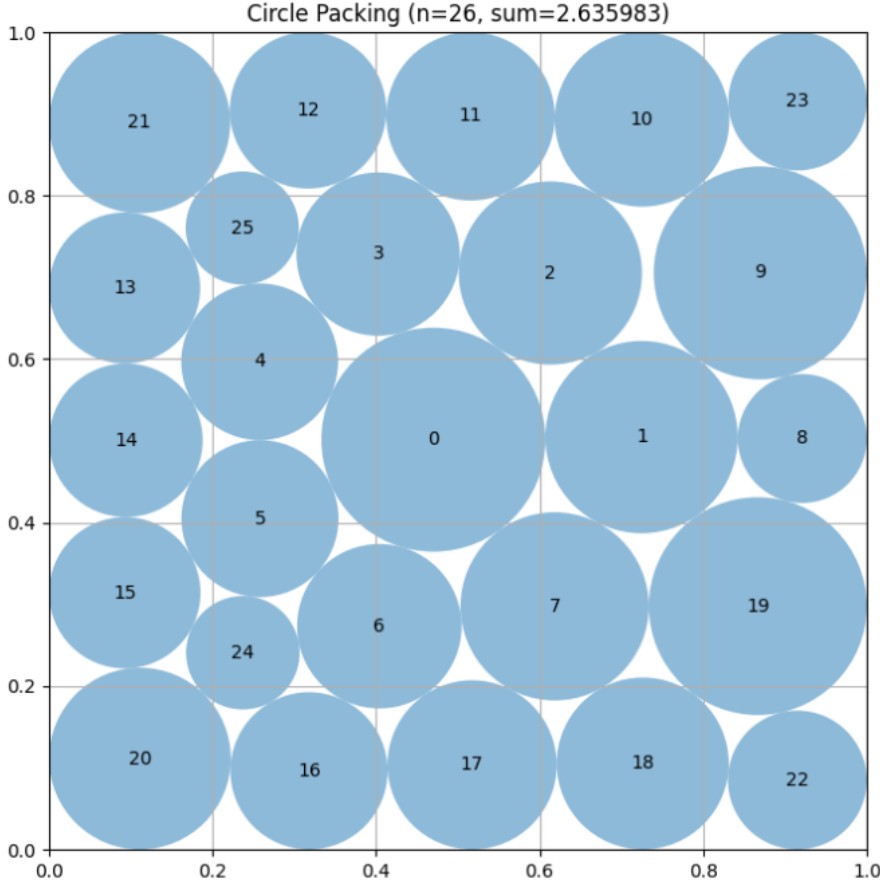

Figure 19: Arrangement of 26 circles within a square obtained by our framework, achieving a record-breaking sum of radii of 2.63598308.

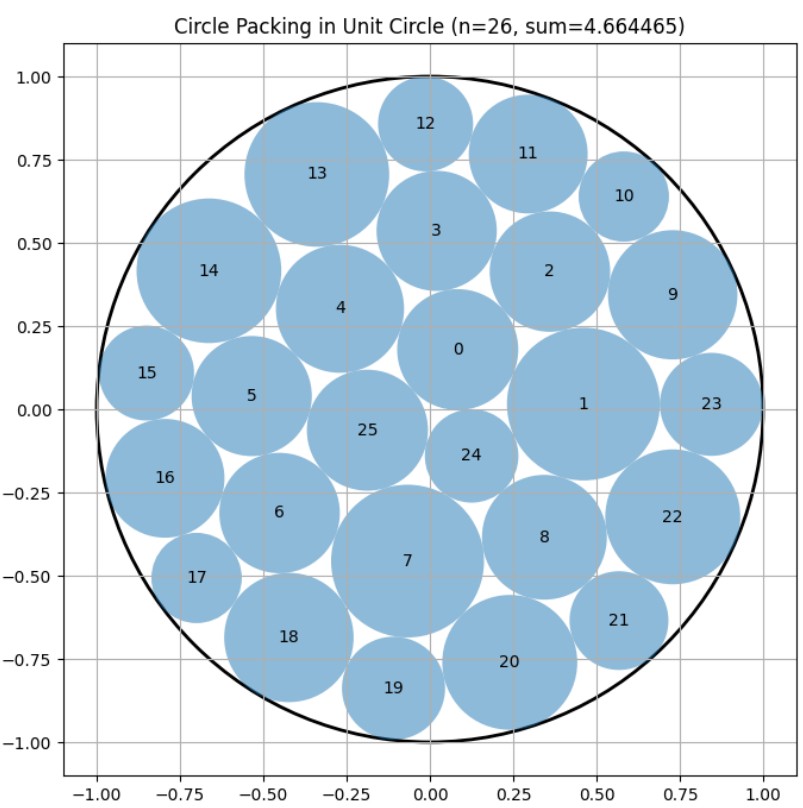

Figure 20: Optimized circle packing of 26 disks within a unit disk by our model, yielding a sum of radii of 4.664465.

```python
def engineer_features(X):
    features = X.copy()
    num_cols = [col for col in X.columns if X[col].dtype != 'object' and
    col not in ['fnlwgt', 'education-num']]

    # Core interaction features
    features['age_hpw_product'] = features['age'] * features['hours-per-
    week']
    features['capital_total'] = features['capital-gain'] + features['
    capital-loss']
    features['log_fnlwgt'] = np.log(features['fnlwgt'] + 1)
    # Enhanced polynomial features
    for col in ['age', 'hours-per-week', 'capital-gain', 'capital-loss']:
        features[f'{col}_sq'] = features[col] ** 2
        features[f'{col}_cb'] = features[col] ** 3
    # Age-based features with log transformation
    features['log_age'] = np.log(features['age'] + 1)
    # Capital features with log transformations
    features['capital_gain'] = np.log(features['capital-gain'] + 1)
    features['capital_loss'] = np.log(features['capital-loss'] + 1)
    # Binned features for age and hours per week
    features['age_group'] = pd.cut(features['age'], bins=5, labels=False)
    features['hour_group'] = pd.cut(features['hours-per-week'], bins=5,
    labels=False)
    # Economic status features combining multiple variables
    features['economic_status'] = (features['age'] / features['hours-per-
    week']) * (features['capital_total'])
    # Additional indicators for capital gains and losses
    features['has_capgain'] = (features['capital-gain'] > 0).astype(int)
    features['has_caploss'] = (features['capital-loss'] > 0).astype(int)
    # Professional education indicator
    features['isProfessional'] = ((features['education'] == 'Prof-
    specialty') | (features['education'] == 'Exec-managerial') | (
    features['education'] == 'Assoc-acdm')).astype(int)
    # Managerial education indicator
    features['isManagerial'] = ((features['education'] == 'Exec-
    managerial') | (features['education'] == 'Assoc-voc')).astype(int)
    # Interaction between numerical features
    interaction_cols = ['age', 'hours-per-week', 'capital_gain', '
    capital_loss']
    for i in range(len(interaction_cols)):
        for j in range(i+1, len(interaction_cols)):
            col1 = interaction_cols[i]
            col2 = interaction_cols[j]
            features[f'{col1}_x_{col2}'] = features[col1] * features[col2
    ]
    # Ratio and difference features
    features['capital_gain_ratio'] = features['capital_gain'] / features[
    'capital_loss'].replace(0, 1)
    features['capital_diff'] = features['capital_gain'] - features['
    capital_loss']

    return features
```

Figure 21: Python code of feature engineering for solving classification task on Adult dataset. Together with a LGBMClassifier and parameter search, our model achieved 82.07 marco f1 score.

```python
1  def extract_features(rawdata):
2      """Returns a pd.DataFrame of enhanced materials features."""
3      """
4      Returns a pd.DataFrame of enhanced materials features.
5      """
6      features = []
7      for _, row in rawdata.iterrows():
8          coords = row["coords"]
9          elements = row["elements"]
10         lattice_vec = row['lattice_vectors']
11         # Efficient atomic composition count using list comprehensions
12         atom_counts = {"Al": sum(1 for el in elements if el == 'Al'),
13                        "Ga": sum(1 for el in elements if el == 'Ga'),
14                        "In": sum(1 for el in elements if el == 'In'),
15                        "O": len(elements) - sum(1 for el in elements if
    el in {'Al', 'Ga', 'In'})}
16         # Structural features
17         struct = _calculate_coordination_number(coords, elements)
18         mean_coords = coords.mean(axis=0)
19         centroid = [mean_coords[0], mean_coords[1], mean_coords[2]]
20         # Lattice vector features
21         a, b, c = lattice_vec
22         vector_lengths = [np.linalg.norm(a), np.linalg.norm(b), np.linalg
    .norm(c)]
23         # Volume calculation
24         volume = np.abs(np.dot(np.cross(b, c), a))
25         # Angle calculations in radians
26         dot_ab = np.dot(a, b)
27         dot_ac = np.dot(a, c)
28         dot_bc = np.dot(b, c)
29         len_a = np.linalg.norm(a)
30         len_b = np.linalg.norm(b)
31         len_c = np.linalg.norm(c)
32         angle_ab = np.arccos(dot_ab / (len_a * len_b)) if (len_a * len_b)
    != 0 else 0
33         angle_ac = np.arccos(dot_ac / (len_a * len_c)) if (len_a * len_c)
    != 0 else 0
34         angle_bc = np.arccos(dot_bc / (len_b * len_c)) if (len_b * len_c)
    != 0 else 0
35         # Advanced structural features
36         bond_feats = _calculate_bond_lengths(coords, elements)
37         # Add 2D and 3D kernel density estimates for atomic distribution
38         from sklearn.neighbors import KernelDensity
39         from sklearn.decomposition import PCA
40         # Perform PCA to reduce dimensionality for density estimation
41         pca = PCA(n_components=2)
42         coords_pca = pca.fit_transform(coords)
43         # Calculate 2D kernel density
44         kde_2d = KernelDensity(bandwidth=0.5).fit(coords_pca)
45         densities_2d = np.exp(kde_2d.score_samples(coords_pca))
46         # Calculate 3D kernel density
47         kde_3d = KernelDensity(bandwidth=0.5).fit(coords)
48         densities_3d = np.exp(kde_3d.score_samples(coords))
49
50         features.append(...) # All features listed above
51     feature_df = pd.DataFrame(features)
52     return feature_df
```

Figure 22: Key pre-processing steps and geometric feature engineering implemented on the Transparent Conductor dataset. Together with an XGBRegressor, the model achieved a score of 0.04915 in RMSLE, securing the second-highest position on the participants' leaderboard.

