# OpenReview forum: "Helix: Evolutionary Reinforcement Learning for Open-Ended Scientific Problem Solving"
_ICLR.cc/2026/Conference — ICLR 2026 Poster_

### Official Review · Reviewer_TewV · 2025-10-25

**Soundness:** 3
**Presentation:** 3
**Contribution:** 2
**Rating:** 4
**Confidence:** 3

**Summary:**

This paper proposes a unified framework that integrates reinforcement learning (via GRPO), evolutionary search (via NSGA-II), and in-context learning to enable large language models to autonomously tackle complex, unbounded scientific problems. HELIX maintains a diverse population of candidate solutions, embedding and filtering them for both reward and semantic diversity, while reinforcement learning updates the policy to progressively improve reasoning and exploration efficiency. Through iterative feedback loops, the model learns to build upon previous high-quality solutions, balancing exploration and exploitation. Evaluated on 19 tasks across five domains—including machine learning, physics simulation, circle packing, function minimization, and symbolic regression—HELIX consistently outperforms strong baselines, including GPT-4o and task-specific algorithms, and achieves state-of-the-art results such as a new world record in the circle-packing benchmark.

**Strengths:**

- **\[S1] Important and timely problem.** The paper tackles the open-ended scientific problem-solving capability of LLMs, an increasingly important direction with potential impact across AI-driven discovery, optimization, and reasoning tasks.

- **\[S2] Thoughtful integration of multiple paradigms.** The proposed HELIX framework effectively combines reinforcement learning (GRPO-style policy optimization), evolutionary search (NSGA-II–based diversity–reward balancing), and in-context learning into a single hierarchical system. While each component is known, their coordination into an iterative, feedback-driven process is well-motivated and technically coherent.

- **\[S3] Comprehensive and diverse experimental evaluation.** The authors evaluate HELIX on 19 tasks spanning five domains—including machine learning, physics simulation, symbolic regression, and geometric optimization—demonstrating broad applicability and consistent improvements over strong baselines such as GPT-4o and AlphaEvolve-style systems.

**Weaknesses:**

See “Questions” below.

**Questions:**

- **\[Q1]** The proposed framework mainly combines existing components—GRPO-style reinforcement learning, NSGA-II–based evolutionary selection, and in-context prompting. Could the authors clarify what specific technical or methodological challenges arise when integrating these elements, and how HELIX overcomes them? In particular, what makes this combination more than an engineering effort or straightforward composition of known techniques?

- **\[Q2]** The paper motivates HELIX as balancing exploration and exploitation through the interplay of RL, evolution, and in-context learning. However, no formal theoretical analysis or guarantees are provided. Could the authors clarify whether there is any theoretical grounding (e.g., convergence, stability, or optimality analysis) supporting this hybrid framework, or is it intended purely as an empirical system-level contribution?

- **\[Q3]** While HELIX shows consistent empirical improvements over baselines across multiple benchmarks, many of these tasks (e.g., UCI datasets, classical optimization functions, small-scale physics simulations) are relatively well-studied and limited in scale. Could the authors clarify how to interpret the significance of these improvements—do they demonstrate genuine scientific discovery capabilities, or primarily reflect engineering and optimization gains within benchmark settings? In other words, how do these results translate to more challenging, real-world, or open-ended scientific problems?

---

> ### Author Response · Authors · 2025-11-20
> **Reply to Question 1**
>
> ## [Q1.1] what specific technical or methodological challenges arise when integrating these elements, and how HELIX overcomes them?
>
> **Answer.** When integrating reinforcement learning (RL) with evolutionary algorithms (EAs), several non-trivial challenges arise. HELIX addresses these challenges as follows:
>
> 1. **Mismatch of the goals of RL and EAs.**
>    RL aims to learn a policy that maps states to actions, whereas EAs are population-based optimization methods that generate new solutions by recombining and mutating existing ones. Therefore, their integration is not straightforward. HELIX maximizes the complementarity between the two via two key ideas.
>    **First, we adopt an in-context learning paradigm** in which previously discovered solutions are incorporated into the prompt as explicit memory. This enables the LLM to act as a *parameterized mutation operator*, conditioned on historical high-quality solutions. We then train this mutation operator using GRPO to improve the quality of mutations over time.
>    **Second, GRPO naturally produces diverse rollouts, which serve as an effective population for evolutionary selection.** In this way, policy optimization and evolutionary search are coupled within a unified framework, providing a principled bridge between RL and EA rather than a naive combination.
> 2. **Diversity estimation and Population selection in giant code action spaces.**
>    For optimization in code spaces—where the action space is extremely large and highly structured—measuring individual diversity using traditional hand-designed metrics is difficult. We propose an embedding-based approach to quantify the distance between two code individuals. Based on these distances, we compute uniqueness/diversity via k-nearest neighbors (kNN), which provides a scalable and semantically meaningful estimate of population diversity in the code domain.
>
> ## [Q1.2] what makes this combination more than an engineering effort or straightforward composition of known techniques?
>
> **Answer.** The **AlphaEvolve** is the easiest combination of RL and EAs. It first post-trains the LLM and then applies an evolutionary algorithm to solve each downstream task. However, as noted in the paper, this pipeline has a fundamental limitation: the model cannot effectively learn from the high-quality solutions discovered during evolutionary search, and thus its mutation capability does not improve with continued exploration.
>
> In contrast, HELIX leverages the explored solutions in two complementary ways. First, these solutions provide feedback signals for reinforcement learning, enabling us to directly update and improve the policy (i.e., the learned mutation operator). Second, they are stored as explicit memory and injected into prompts, allowing the model to condition on prior successful trajectories and generate better candidates at test time. This yields a unified, closed-loop framework in which **test-time scaling via evolutionary search and policy learning via RL mutually reinforce each other**.
>
> Overall, HELIX builds a fundamentally new paradigm that integrates test-time scaling with reinforcement learning to tackle complex, open-ended scientific problems. Empirically, this joint framework delivers substantial and consistent gains across multiple tasks compared to methods that combine RL and evolution only in a sequential or loosely coupled manner.

---

> > ### Author Response · Authors · 2025-11-24
> > **Reply to Additional Question 2 [Comment 2]**
> >
> > ## 2. Theoretical Analysis of GRPO
> >
> > ### Setup and Assumptions:
> >
> > In the GRPO method, since the prompt is sampled once and remains fixed, the model evolves **solely from an initial solution** $v_0$. The transition samples from $\mathcal N(v_0 + \delta_\theta(v_0), \sigma \mathbf I)$.
> >
> > Using GRPO, the model estimates the gradient of the reward function near $v = v_0 + \delta_\theta(v_0)$ using sampling results and updates the model parameters. We define this gradient update process as:
> > $$
> > \delta_\theta(v_0) \leftarrow \delta_\theta(v_0) + \eta \nabla_v r(v_0 + \delta_\theta(v_0))
> > $$
> > Effectively, the update is $v \leftarrow v + \eta \nabla_v r(v)$, where $\eta$ is the learning rate.
> >
> > **To demonstrate that GRPO tends to converge to local optima in this setting**, we introduce an assumption regarding the Reward Landscape.
> >
> > **Assumption 1 (Reward Landscape)**:
> > Assume the reward function $r(v)$ consists of two Gaussian peaks (representing a local optimum and a global optimum):
> > $$
> > r(v) = A_{loc}\exp \left(-\frac{\|v-v_{loc}\|^2}{2w^2}\right) + A_{opt}\exp\left(-\frac{\|v-v_{opt}\|^2}{2w^2}\right)
> > $$
> > Let $\|v_{opt} - v_{loc}\| = L$.
> >
> > ### Theorem 1 (Convergence to Local Optimum of GRPO):
> >
> > Let $v_0$ be the initial solution for GRPO. GRPO **will converge to the local optimum** near $v_{loc}$ if the following conditions are met:
> >
> > 1.  $L > 2w$: There is sufficient separation between the global and local optima.
> > 2.  $A_{loc} > A_{opt} \cdot \frac{L}{w} \cdot \exp \left(-\frac{L^2}{2w^2}\right)$: The local optimum is not significantly weaker than the global optimum.
> > 3.  Decomposing the initial solution as $v_0 = v_{loc} + v_\perp + \gamma_0 (v_{opt} - v_{loc})$, where $v_\perp \perp (v_{opt} - v_{loc})$, we require $\gamma_0 < \gamma_{barrier} \approx \frac{1}{2} - \frac{w^2}{L^2}\ln \frac{A_{opt}}{A_{loc}}$. This implies the initial solution is closer to the local optimum.
> >
> > ### Comment for Theorem 1:
> >
> > These assumptions are not overly strong. In real-world scientific problems:
> >
> > * There is often a significant semantic gap between local and global optima (e.g., using entirely different algorithms), satisfying Condition 1 ($L>2w$).
> > * Since $\exp (-\frac{L^2}{2w^2}) \ll 1$, Condition 2 is easily satisfied even if $A_{opt} > A_{loc}$.
> > * Initial solutions are often "naive" or rudimentary, placing them further from the complex global optimum and closer to simpler local optima, satisfying Condition 3.
> >
> > This illustrates that GRPO, relying solely on parameter updates without memory or population mechanisms, **is prone to converging to fixed points (local optima) and failing to explore the full solution space**.
> >
> > ### Proof of Theorem 1:
> >
> > We first decompose the gradient of $r(v)$. Let $v = v_{loc} + v_\perp + \gamma (v_{opt} - v_{loc})$. Then:
> > \begin{align}
> > \begin{aligned}
> > \nabla r(v) &= \frac{A_{loc}}{w^2}(v_{loc}-v)\exp \left(-\frac{\|v-v_{loc}\|^2}{2w^2}\right) + \frac{A_{opt}}{w^2}(v_{opt}-v)\exp \left(-\frac{\|v-v_{opt}\|^2}{2w^2}\right) \\\\
> > \langle \nabla r(v), v_{opt}-v_{loc}\rangle &= -\frac{A_{loc}L^2}{w^2}\gamma\exp \left(-\frac{\|v-v_{loc}\|^2}{2w^2}\right) + \frac{A_{opt}L^2}{w^2}(1-\gamma)\exp \left(-\frac{\|v-v_{opt}\|^2}{2w^2}\right)\\\\
> > &= -\frac{A_{loc}L^2}{w^2}\gamma\exp \left(-\frac{\gamma^2L^2+\|v_\perp\|^2}{2w^2}\right) + \frac{A_{opt}L^2}{w^2}(1-\gamma)\exp \left(-\frac{(1-\gamma)^2L^2+\|v_\perp\|^2}{2w^2}\right)\\\\
> > \langle \nabla r(v), v_\perp\rangle &= -\frac{\|v_\perp\|^2}{w^2}\left(A_{loc}\exp \left(-\frac{\|v-v_{loc}\|^2}{2w^2}\right) + A_{opt} \exp \left(-\frac{\|v-v_{opt}\|^2}{2w^2}\right)\right)
> > \end{aligned}
> > \end{align}
> >
> > First, analyzing the evolution of the perpendicular component $v_\perp$:
> > $$
> > \frac{d v_\perp}{dt} = -\frac{\eta}{w^2}v_\perp \left( A_{loc}\exp(\cdots) + A_{opt}\exp(\cdots) \right)
> > $$
> > It follows that:
> > $$
> > \frac{d}{dt}\|v_\perp\|^2 = -C(v)\|v_\perp\|^2 < 0
> > $$
> > Regardless of the initial state, $v_\perp$ decays exponentially to 0. Thus, the system eventually converges to the linear manifold connecting $v_{loc}$ and $v_{opt}$. We assume $v_\perp = 0$ henceforth.
> >
> > Focusing on the dynamics along this line:
> > $$
> > \frac{d\gamma}{dt} \propto \langle \nabla r(v), v_{opt}-v_{loc}\rangle \propto -\gamma A_{loc}\exp\left(-\frac{\gamma^2L^2}{2w^2}\right) + (1-\gamma) A_{opt}\exp\left(-\frac{(1-\gamma)^2L^2}{2w^2}\right)
> > $$
> > Analyzing the equilibrium points ($\frac{d\gamma}{dt}=0$) under the given conditions yields three solutions:
> >
> > 1.  **Stable Local Equilibrium:** $\gamma_{stable}^{(1)}$ (very close to 0, the local optimum).
> > 2.  **Unstable Saddle Point:** $\gamma_{barrier}$ (between 0 and 1), estimated as $\gamma_{barrier} \approx \frac{1}{2} - \frac{w^2}{L^2}\ln \frac{A_{opt}}{A_{loc}}$.
> > 3.  **Stable Global Equilibrium:** $\gamma_{stable}^{(2)}$ (very close to 1, the global optimum).
> >
> > From the differential equation analysis, if the initial $\gamma_0 < \gamma_{barrier}$, the system converges to $\gamma_{stable}^{(1)}$. Conversely, it converges to $\gamma_{stable}^{(2)}$. This completes the proof.

---

> > ### Author Response · Authors · 2025-11-24
> > **Reply to Additional Question 2 [Comment 3]**
> >
> > ## 3. Theoretical Analysis of Evolve and HELIX
> >
> > ### Setup: Unified Drift-Diffusion and Selection Framework
> >
> > The analysis is based on the iterative process used by the two algorithms, both maintaining a population $\mathcal{P}$ and updating it by generating new solutions $v'$.
> >
> > **Evolve Algorithm:**
> > In the Evolve algorithm, the model parameters cannot be adjusted. Thus, we assume the model has no inherent directional bias towards different methods of this specific problem ($\delta_\theta(v) \equiv 0$). The iteration simplifies to a random walk: $v \to v + \sigma \xi$, with $\xi \sim \mathcal{N}(0, \mathbf{I})$. At each step, a solution $v$ is drawn from $\mathcal{P}$, and $v' = v + \sigma \xi$ is generated. Critically, **solutions with higher Reward are sampled with a higher probability**. We can model this selection by a weight function $w(v) = \exp(\alpha r(v))$, where $\alpha$ represents the **selection pressure** ($\alpha$ increases the focus on high-Reward solutions). The new solution is added to the population: $\mathcal{P} \leftarrow \mathcal{P} \cup \{v'\}$.
> >
> > **HELIX Algorithm:**
> > HELIX maintains a population $\mathcal{P}$ and dynamically adjusts the directional bias $\delta_\theta(v)$. Through the GRPO mechanism, this direction will gradually approximate the gradient $\nabla r(v)$. Upon sufficient convergence, the HELIX iteration approximates a guided random walk: $v' = v + \eta \nabla r(v) + \sigma \xi$. In HELIX, we also sample high-Reward solutions with higher probability, but for mathematical tractability, we assume the selection weight parameter $\alpha = 0$, meaning the sampling weight is uniform ($w(v) \equiv 1$).
> >
> > | Algorithm  | Dynamics                                 | Drift $D(v)$                    | Selection Weight $w(v)$      | Continuous-Time Model                        |
> > | :--------- | :--------------------------------------- | :------------------------------ | :--------------------------- | :------------------------------------------- |
> > | **Evolve** | $v' = v + \sigma \xi$                    | $D(v) \equiv \mathbf{0}$        | $w(v) = \exp(\alpha r(v))$   | **Selection-Diffusion** (Replicator-Mutator) |
> > | **HELIX**  | $v' = v + \eta \nabla r(v) + \sigma \xi$ | $D(v) \approx \eta \nabla r(v)$ | $w(v) \equiv 1$ ($\alpha=0$) | **Drift-Diffusion** (Langevin Dynamics)      |
> >
> > ### Theorem 2: Stationary Distribution of Evolve and HELIX
> >
> > **Theorem 2.** Assuming the projected solution space $\Phi(\mathcal{S}) \subset \mathbb{R}^n$ is bounded, as time $t \to \infty$, the population distribution $p^*(\mathbf{v})$ for both algorithms converges to a stationary distribution:
> >
> > 1. **Evolve (Selection-Diffusion):** Converges to the **Principal Eigenfunction** of the associated Schrödinger operator, which, under the WKB approximation, has the asymptotic form:
> >    $$
> >    p^*_{Evo}(\mathbf{v}) \asymp \exp\left(-\frac{\sqrt{2\alpha}}{\sigma}\int_{v_{opt}}^{v} \sqrt{r(v_{opt})-r(u)}du\right)
> >    $$
> >
> > 2. **HELIX (Drift-Diffusion):** Converges to a **Boltzmann-Gibbs Measure**:
> >    $$
> >    p^*_{Helix}(\mathbf{v}) \propto \exp\left(\frac{2\eta}{\sigma^2}r(\mathbf{v})\right)
> >    $$
> >    *(Note: $\asymp$ denotes asymptotic equivalence, specifically for the leading order term in the exponent as $\sigma \to 0$.)*

---

> ### Author Response · Authors · 2025-11-20
> **Reply to Question 2-3**
>
> ## [Q2]. Theoretical grounding of the framework.
>
> **Answer.** Due to the intrinsic complexity of large language models, the dominant line of research on RL for LLMs remains largely empirical: algorithmic effectiveness is typically demonstrated through consistent performance improvements rather than through fully developed theoretical guarantees. Our work follows this established paradigm. Importantly, we validate the benefits of HELIX on a sufficiently rich set of tasks and benchmarks that are both challenging and representative, thereby providing strong empirical evidence for its effectiveness.
>
>
>
> ## [Q3]. how do these results translate to more challenging, real-world, or open-ended scientific problems?
>
> **Answer.**  The results and ideas/designs generated by our method are transferrable to large-scale problems due to the following reasons.
>
> 1.  **First, the problems we study are not merely toy settings.**  In particular, the physical simulation tasks and tabular dataset we consider have close analogues in real industrial workflows, where problem scale and complexity are comparable to those in our experiments. Thus, our evaluation targets scenarios that are meaningfully representative of real-world scientific and engineering applications.
>
> 2.  **Second, our method exhibits a multi-stage, iterative pattern of discovery surpassing even human experts.**  For example, in the ML tasks, the model autonomously explores richer solution spaces by performing hyperparameter search and constructing new hand-crafted features. In the physical simulation tasks, it proposes non-trivial design variations such as employing larger circular beams, or combining circular beams with flanges. These behaviors indicate that the model is not only optimizing within a fixed template, but is actively acquiring and refining novel design strategies during exploration—strategies that in several cases go beyond those produced by strong competition participants or even domain experts.
>
> 3.  **Third, some newly discovered design heuristics and strategies are irrelavant to problem scales and are transferrable to larger scale problems**—e.g., feature engineering principles, optimization recipe, and reusable physical design patterns. Such transferable insights can be applied to larger-scale problems or datasets; the only difference is that validating and learning on them require larger computation resources, not fundamental changes to the framework.

---

> ### Comment · Reviewer_TewV · 2025-11-21
>
> Thanks for the rebuttal. Regarding Q1, I have several further questions:
> - Are those things in the paper? If yes, where are they? If not, where do you want to add them?
> - Can you elaborate a bit more on whether and how one would fail if they just naively integrate these elements, instead of the sophisticated way as in HELIX?
>
> Regarding Q2, I am totally fine with it to be fully empirical, but I think it is better to be explicit.
>
> Regarding Q3, I think your arguments are definitely reasonable, but they are largely philosophical without clear empirical *evidence*, do you have any results or at least ideas about additional experiments?

---

> ### Author Response · Authors · 2025-11-24
> **Reply to Additional Question 1**
>
> Thank you for your timely response! Our Reply is as follows:
>
> ## Q1.1 Including the discussion of [Q1] to the paper
>
> The analysis presented has not yet been included in the previous version of manuscript. We have included the detailed analysis in **Appendix E** in the updated version. Furthermore, we will carefully synthesize the key findings from this analysis and integrate them into the **Experiments** section of the main text in the near future.
>
> ## Q1.2 How one would fail if they just naively integrate elements instead of way as in HELIX
>
> As we discussed in our previous response, **AlphaEvolve** represents the simplest naive combination of RL and EAs. It initially post-trains an LLM on general domains and subsequently applies evolutionary algorithms on the downstream task.
>
> However, the model's capabilities are not effectively enhanced through this evolutionary process, frequently causing the exploration to become **stuck in local optima** due to the fixed limitations of the base model. Furthermore, the inherent randomness of the initial seed solutions and insufficient population diversity significantly restrict the overall potential of such naive approaches. We demonstrate this through a detailed experiment using the **Circle Packing** problem (to arange 26 non-overlapping circles in unit square to maximize sum of radii):
>
> We conducted 3 experiments on both the Qwen 14B and Qwen 32B models using **OpenEvolve** (an open-source reproduction of AlphaEvolve). The results were unsatisfactory; notably, OpenEvolve with Qwen 14B **failed to surpass the Direct Prompt (BO64) baseline.**
>
> | DirectPrompt Qwen 14B | OpenEvolve Qwen 14B | OpenEvolve Qwen 32B | HELIX Qwen 14B |
> | --------------------- | ------------------- | ------------------- | -------------- |
> | 1.673                 | 1.586               | 1.956               | 2.636          |
>
> Our in-depth analysis suggests two critical limitations of OpenEvolve: (Details will follow in the next response due to length limits.)

---

> ### Author Response · Authors · 2025-11-24
> **Reply to Additional Question 1.2 [Continued]**
>
> Our in-depth analysis suggests two critical limitations of OpenEvolve:
>
> 1. **Constrained by Initial Solutions and Limited Diversity**:
>    * **Observation:** The direct prompt baseline performs **64 independent and identically distributed (i.i.d.) evaluations**. In contrast, the OpenEvolve algorithm starts by generating only **5 seed trials** and then iteratively evolves them. Although OpenEvolve significantly utilizes more LLM calls than the Direct Prompt (BO64) baseline, it performs even worse.
>    * **The Failure:** The core issue is that OpenEvolve is largely constrained by initial solutions, which introduces a **high degree of randomness and uncertainty**. If the initial trials lack diversity or have a low performance ceiling, OpenEvolve tends to iterate only on this limited basis. This behavior ultimately constrains the performance of OpenEvolve to the quality and diversity of the **initial solutions**.
>
> 2. **Reluctance towards Destructive or Novel Changes**:
>    * **Observation:** In the Circle Packing experiment with Qwen 32B, OpenEvolve generated 4147 trials. The vast majority were attempts at arranging the positions and radii of circles directly. Only **12 trials (0.3%)** attempted to use the `scipy.optimize.minimize` function for result refinement, and all these attempts **failed with 0.0 reward**. However, these attempts failed due to compilation error or the time limit, which **can be easily fixed and result in high rewards (e.g. >2.0) if the bugs are fixed**.
>    * **The Failure:**
>      1. This indicates that due to the **extreme sparsity of novel methods in the population** (only 0.3%) and the **low initial reward** of these new trials, **OpenEvolve was driven away from exploring this potentially novel approach**, missing the chance to discover feasible higher reward.
>      2. This neglect is rooted in the limitations of traditional evolutionary algorithms (like OpenEvolve): **Diversity** is typically maintained through simple mechanisms such as building Islands, random crossover, or constructing code features (e.g., length or reward). However, such approaches **cannot guarantee that diverse solution methods receive sufficient attention** (e.g., using `scipy.optimize` in Circle Packing might not significantly affect code length, thus the code features remains merely the same) . Consequently, new, potentially valuable ideas are overlooked.
>
> However, HELIX, given the **exact same initial solutions and prompt**, explores much more effectively by overcoming the above two issues through the following sophisticated mechanisms:
>
> 1. **Explicit Diversity Accounting via Embedding Calculation**: We use an embedding model (codet5p-110m-embedding) to **explicitly distinguish different methods at a semantic level**. This assigns a high **Diversity Score** to new attempts (even if they initially have a low Reward), ensuring they are explored more fully, which in turn provides a chance to fix existing errors and achieve better results.
> 2. **Parameter Learning via Reinforcement Learning**: Concurrently, we incorporate **Reinforcement Learning** to dynamically integrate successful experiences into the model's parameters.
>    * Assume the Embedding Model successfully samples a solution $s_t$ using the `scipy.optimize` method (a high-diversity, but possibly low-reward state due to initial errors). The algorithm generates $G$ distinct rollouts: $s_{t+1, j}$ from $s_t$. If, in one trial, the model correctly fixes the code issue and creates a significantly superior solution $s_{t+1}^\star$, the model uses this Reward feedback to **learn the successful experience**. This leads to a higher probability of **applying a similar method to other solution iterations**, thereby generating a large number of similar, superior solutions.
>    * This growth of new, superior solutions within the population initiates a **positive feedback loop**, inherently steering the system towards **concentrating further exploration** on these newly discovered and validated methodologies.
>
> In summary, HELIX's design enables the model to **break through from the constraints of existing methods**, **discover new methodologies**, **correct initial errors** in these new approaches, and then **update the model's strategy** through parameters using the successful experience, ultimately yielding a far better and more effective exploration process.

---

> > ### Author Response · Authors · 2025-11-26
> > **Response to Your Comments**
> >
> > Thank you very much for your feedback. We are delighted to hear that our previous responses and new results were convincing and have successfully addressed your primary technical concerns.
> >
> > We fully understand your concern regarding the scope of the revisions and the feasibility of integrating them without disrupting the integrity of the original manuscript. We have carefully planned the integration of these updates as follow:
> >
> > - **Regarding Q1:**
> >   - We have rewritten the method portion in the **Introduction** to more coherently and naturally highlight the motivation and mechanics of combining Evolutionary Algorithms with Reinforcement Learning in HELIX. This also clarifies the distinction between our approach and a simplistic combination without requiring a structural overhaul.
> >   - Furthermore, we have added the performance analysis of OpenEvolve versus HELIX on the Circle Packing problem to the **Ablation Study** in the **Experiment** section, while moving the extensive detailed analysis to **Appendix E** to keep the main text concise.
> > - **Regarding Q2:** To preserve the readability of the main methodology sections, we have placed the new comprehensive theoretical analysis entirely in **Appendix F**.
> > - **Regarding Q3:** We will integrate the new **Nomad2018 Competition results** directly into the **main experiment table** (**Table 1**). We are currently finalizing the baseline testing for this task, which will be completed within a few days. Once finished, we will update the table and add a corresponding concise analysis in the **Experiments** section to ensure the comparison is complete.

---

> ### Author Response · Authors · 2025-11-24
> **Reply to Additional Question 2 [Comment 1]**
>
> ## Q2 Explicit theoretical analysis of the framework
> We wish to reiterate: Due to the intrinsic complexity of large language models, **the dominant line of research on RL for LLMs remains largely empirical**: algorithmic effectiveness is typically demonstrated through consistent performance improvements rather than through fully developed theoretical guarantees.
>
> Furthermore, **it requires incorporating the scientific problem itself into the mathematical modeling** to analyzing the advantages of a proposed architecture that seeks diverse solutions for a single problem. This significantly increases complexity, rendering formal and rigorous convergence, stability, or optimality analysis infeasible.
>
> However, we employ a simplified mathematical model to provide theoretical insights into our algorithm's advantages in solving complex scientific problems. Below, we demonstrate the efficiency of HELIX in discovering optimal solutions using a simplified framework.
>
> ## 1. Mathematical Setup and Preliminaries
>
> First, assume that for a problem $q$, there exists:
>
> * **Solution Space:** A set of solutions $\mathcal S$, which can be viewed as a simply connected open manifold in a complex space.
> * **Reward Function:** A continuous function $R: \mathcal S \to \mathbb R^+$.
> * **Embedding:** A mapping $\Phi: \mathcal S \to \mathbb R^n$ that maps the solution space to **Embedding Space** $\mathbb R^n$, satisfying:
>   1.  **Continuity:** For any $s_1, s_2 \in \mathcal S$ representing solutions derived via similar methods, their embeddings $v_1 = \Phi(s_1)$ and $v_2 = \Phi(s_2)$ are adjacent in $\mathbb R^n$.
>   2.  **Injectivity:** Any two distinct solutions $s_1, s_2 \in \mathcal S$ have distinct embeddings.
>   3.  **Open Map:** $\Phi$ maps open sets to open sets.
>
> We define the **Reward Function in $\mathbb R^n$** as follows: For any $v \in \mathbb R^n$:
> $$
> r(v) = \begin{cases}
> R(\Phi^{-1}(v)) & v \in \Phi(\mathcal S) \\\\
> 0 & v \notin \Phi(\mathcal S)
> \end{cases},
> $$
> where $s = \Phi^{-1}(v)$ is the solution corresponding to $v$. Restricted to the image set, $\Phi: \mathcal S \to \Phi(\mathcal S)$ is a bijection, making its inverse well-defined. Due to the continuity of $\Phi$ and $R$, and the fact that $\Phi$ is an open map, $r(v)$ is also continuous.
>
> **LLM Transition Process:**
> For a solution $s$, an LLM parameterized by $\theta$ transforms the solution by outputting $a \sim \pi_\theta(\cdot | s)$, resulting in $s' = T(s, a)$. Based on this, we define the **LLM Transition Function on $\mathbb R^n$** (a stochastic process with probability space $\Omega$):
> $$
> T^\star_\theta: \mathbb R^n \to (\Omega \to \mathbb R^n), \quad T^\star_\theta(v) = \Phi(T(\Phi^{-1}(v), a)), \quad \text{where } a \sim \pi_\theta(\cdot | \Phi^{-1}(v))
> $$
> For simplicity, we assume $T^\star_\theta$ is independent and follows a Normal distribution:
> $$
> T^\star_\theta(v) \sim \mathcal N(v + \delta_\theta(v), \sigma \mathbf I)
> $$
> This implies each transition satisfies $v \to v + \delta_\theta(v) + \xi$, where $\xi \sim \mathcal N(0, \sigma \mathbf I)$. This assumption is reasonable because LLMs typically generate modest modifications to the current solution, with a significantly lower probability of producing drastic changes. Consequently, this behavior can be effectively approximated as a Gaussian distribution within the embedding space.

---

> ### Author Response · Authors · 2025-11-24
> **Reply to Additional Question 2 [Comment 4]**
>
> ### Comment on Theorem 2:
>
> 1. **Concentration and $\sigma$ Scaling:** The concentration power of the distributions (the inverse of their variance) scales as $\frac{1}{\sigma}$ for Evolve and $\frac{1}{\sigma^2}$ for HELIX. Since $\sigma \ll 1$ in high-precision search, $\frac{1}{\sigma^2} \gg \frac{1}{\sigma}$, indicating that **HELIX's sampling is exponentially more concentrated around the optimum**. Under the same conditions, **HELIX achieves much more exhaustive exploration of the highly rewarded vicinity of the optimal solution.**
>
> 2. **Intuitive Comparison (Quadratic Reward):** We use a simple quadratic function $r(v) = r_{opt} - \frac{k}{2}\|v\|^2$ (centered at $v_{opt}=0$) to derive the exact Gaussian form of the distributions for direct comparison:
>
>    | Distribution | Exact Gaussian Form $p^* \propto \exp(-C\|v\|^2)$            | Variance $\Sigma^2$ (Concentration)               | Scaling vs. $\sigma$         |
>    | :----------- | :----------------------------------------------------------- | :------------------------------------------------ | :--------------------------- |
>    | **HELIX**    | $\exp\left(-\frac{\eta k}{\sigma^2} \|v\|^2 \right)$         | $\Sigma^2_{Helix} = \frac{\sigma^2}{2\eta k}$     | $\propto \sigma^2$ (Sharper) |
>    | **Evolve**   | $\exp\left(-\frac{\sqrt{\alpha k}}{2\sigma} \|v\|^2 \right)$ | $\Sigma^2_{Evo} = \frac{\sigma}{\sqrt{\alpha k}}$ | $\propto \sigma$ (Broader)   |
>
>    The resulting variance ratio $\Sigma^2_{Helix} / \Sigma^2_{Evo} \propto \frac{\sigma}{\sqrt{\eta}}$, which tends to zero as $\sigma \to 0$. This rigorously confirms that HELIX's ability to use the gradient for directional movement provides a superior mechanism for stabilizing and concentrating the population compared to Evolve's reliance on scalar selection alone.
>
> 3. **Potential for Further Reinforcement:** If we were to consider a non-trivial selection weight $w(v) = \exp(\alpha r(v))$ for HELIX as well, the final stationary distribution would be intuitively more concentrated, though it is mathematically infeasible for a quantitative analysis. This would indeed further reinforce HELIX's focus and exploration within high-reward regions.
>
> ### Proof of Theorem 2:
>
> #### **Part I: HELIX (Drift-Diffusion - Exact Solution)**:
>
> The HELIX algorithm, when converged, follows a dynamics characterized by a gradient-driven drift and additive Gaussian noise. This process is modeled by the continuous **Langevin Equation**:
> $$
> d\mathbf{v}_t = \eta \nabla r(\mathbf{v}_t) dt + \sigma d\mathbf{W}_t
> $$
> where $d\mathbf{W}_t$ is an $n$-dimensional Wiener process (Brownian motion).
>
> The time evolution of the probability density $p(\mathbf{v}, t)$ for this system is governed by the **Fokker-Planck (FP) Equation**:
> $$
> \frac{\partial p}{\partial t} = -\nabla \cdot \mathbf{J} = -\nabla \cdot (p D(\mathbf{v})) + \frac{1}{2} \nabla^2 (\Sigma p)
> $$
> In our case, the covariance matrix is $\Sigma = \sigma^2 \mathbf{I}$, simplifying the FP equation.
>
> The total probability flux $\mathbf{J}$ is defined as:
> $$
> \mathbf{J} = p D(\mathbf{v}) - \frac{\sigma^2}{2} \nabla p
> $$
>
> **1. Stationary Condition (Zero Flux)**
>
> The stationary distribution $p_{Helix}^\*(\mathbf v)$ is reached when $\frac{\partial p}{\partial t} = 0$, which implies the total probability flux $\mathbf{J}$ must be zero everywhere:
> $$
> \mathbf{J} = p^* D(\mathbf{v}) - \frac{\sigma^2}{2} \nabla p^* = \mathbf{0}
> $$
>
> **2. Solving the Zero-Flux Condition**
>
> Substituting the drift term $D(\mathbf{v}) = \eta \nabla r(\mathbf{v})$ into the zero-flux condition:
> $$
> p_{Helix}^\* (\eta \nabla r(\mathbf v)) = \frac{\sigma^2}{2} \nabla p_{Helix}^\*
> $$
> We rearrange the equation to isolate the gradient of the distribution:
> $$
> \nabla p_{Helix}^\* = p_{Helix}^\* \left(\frac{2\eta}{\sigma^2}\right) \nabla r(\mathbf{v})
> $$
>
> **3. Integration and Final Form**
>
> We divide by $p_{Helix}^\*$ and recognize the left side as the gradient of the natural logarithm of $p_{Helix}^\*$:
> $$
> \nabla \ln p_{Helix}^\* = \frac{\nabla p_{Helix}^\*}{p_{Helix}^\*} = \left(\frac{2\eta}{\sigma^2}\right) \nabla r(\mathbf{v})
> $$
> Integrating both sides with respect to $\mathbf v$:
> $$
> \ln p_{Helix}^\*(\mathbf v) = \left(\frac{2\eta}{\sigma^2}\right) r(\mathbf v) + C'
> $$
> where $C'$ is the integration constant. Exponentiating yields the stationary distribution, which is the exact **Boltzmann-Gibbs Measure**:
> $$
> p^*_{Helix}(\mathbf{v}) = \exp(C') \exp\left(\frac{2\eta}{\sigma^2} r(\mathbf{v})\right) \propto \exp\left(\frac{2\eta}{\sigma^2} r(\mathbf{v})\right)
> $$
> This demonstrates that HELIX concentrates the population exponentially in regions proportional to the reward function $r(\mathbf{v})$, with an effective inverse temperature $\beta_{eff} = \frac{2\eta}{\sigma^2}$.

---

> ### Author Response · Authors · 2025-11-24
> **Reply to Additional Question 2 [Comment 5]**
>
> #### **Part II: Evolve (Selection-Diffusion - WKB Approximation)**
>
> The Evolve algorithm operates as a selection-mutation process. Its stationary distribution $p_{Evo}^\*$ is determined by the balance between the diffusion (mutation $\sigma$) and the growth/decay (selection $\alpha$).
>
> **1. Transition from Discrete Iteration to Continuous Dynamics (Replicator-Mutator Equation)**
>
> The discrete process is:
> $$
> p_{t+1}(v') = \int K(v'|v) \frac{w(v)p_t(v)}{\bar{w}_t} dv
> $$
> where $K(v'|v) = \mathcal{N}(v'|v, \sigma^2 \mathbf{I})$ is the mutation kernel, $w(v) = e^{\alpha r(v)}$ is the selection weight, and $\bar{w}_t = \int w(u)p_t(u)du$ is the normalization factor.
>
> Taking the continuous-time limit ($\Delta t \to 0$) of this selection-mutation process, we arrive at the **Replicator-Mutator Equation** (a well-established result in evolutionary dynamics, also known as the generalized Fisher-KPP equation in some contexts):
> $$
> \frac{\partial p}{\partial t} = \frac{\sigma^2}{2} \nabla^2 p + \alpha \left( r(v) - \bar{r} \right) p
> $$
> where $\bar{r} = \int r(v)p(v)dv$ is the average reward, and $\alpha (r(v) - \bar{r})p$ models the net growth rate (selection pressure).
>
> **2. The Steady-State Equation (Schrödinger Form)**
>
> The stationary distribution $p^\*$ satisfies $\frac{\partial p}{\partial t} = 0$. Letting $E = \alpha \bar{r}$ (which is the principal eigenvalue, as $p^\*$ is the corresponding eigenfunction), we obtain the time-independent Schrödinger-like equation:
> $$
> \frac{\sigma^2}{2} \nabla^2 p^\* + \alpha r(v) p^\* = E p^\*
> $$
>
> **3. Applying the WKB Ansatz (Semi-Classical Limit)**
>
> Since we are interested in the asymptotic behavior for small noise $\sigma \to 0$ (the semi-classical limit), we use the WKB Ansatz for the density $p^\*$:
> $$
> p^\*(v) = C(v) \exp\left( - \frac{S(v)}{\sigma} \right)
> $$
> where $S(v)$ is the action or quasi-potential. Substituting $p^*(v)$ into the Schrödinger equation:
> $$
> \frac{\sigma^2}{2} \nabla^2 \left( e^{-S/\sigma} \right) + \alpha r(v) e^{-S/\sigma} = E e^{-S/\sigma}
> $$
> Dividing by $e^{-S/\sigma}$ and using the identity $\nabla^2 (e^{-S/\sigma}) = e^{-S/\sigma} \left( \frac{1}{\sigma^2} \|\nabla S\|^2 - \frac{1}{\sigma} \nabla^2 S \right)$, we get:
> $$
> \frac{\sigma^2}{2} \left( \frac{1}{\sigma^2} \|\nabla S\|^2 - \frac{1}{\sigma} \nabla^2 S \right) + \alpha r(v) = E
> $$
> Expanding the term $\frac{\sigma^2}{2}$:
> $$
> \frac{1}{2} \|\nabla S\|^2 - \frac{\sigma}{2} \nabla^2 S + \alpha r(v) = E
> $$
>
> **4. Solving the Hamilton-Jacobi Equation**
>
> As $\sigma \to 0$, the high-order term $\frac{\sigma}{2} \nabla^2 S$ vanishes. The equation simplifies to the **Hamilton-Jacobi Equation** (Eikonal Equation):
> $$
> \frac{1}{2} \|\nabla S\|^2 + \alpha r(v) \approx E
> $$
> The ground state distribution $p^*$ is peaked at the optimal solution $v_{opt}$, where $\nabla S(v_{opt}) = \mathbf{0}$. Setting this condition determines the eigenvalue: $E = \alpha r(v_{opt})$.
> Substituting $E$ back:
> $$
> \|\nabla S(v)\| = \sqrt{2\alpha (r(v_{opt}) - r(v))}
> $$
>
> **5. Final Integral Form**
>
> $S(v)$ is obtained by integrating the norm of its gradient along the path from $v_{opt}$ to $v$:
> $$
> S(v) = \int_{v_{opt}}^v \|\nabla S(u)\| \, du = \sqrt{2\alpha} \int_{v_{opt}}^v \sqrt{r(v_{opt}) - r(u)} \, du
> $$
> Finally, substituting $S(v)$ back into the Ansatz $p^* \asymp e^{-S/\sigma}$ yields the asymptotic stationary distribution:
> $$
> p^*_{Evo}(\mathbf{v}) \asymp \exp\left( - \frac{\sqrt{2\alpha}}{\sigma} \int_{v_{opt}}^v \sqrt{r(v_{opt}) - r(u)} \, du \right)
> $$

---

> ### Author Response · Authors · 2025-11-24
> **Reply to Additional Question 3**
>
> ## Q3 Results and ideas about further experiments
>
> We appreciate the reviewer's feedback regarding the need for more concrete empirical evidence to support the transferability of our results.
>
> To address this, we provide a **new empirical result** on a challenging, open-ended scientific problem, and we clarify the **practical constraints** that limit evaluation on even larger, real-world tasks.
>
> ### New Empirical Evidence: The Nomad2018 Competition
>
> To demonstrate HELIX’s ability to handle complex, open-ended scientific problems that require non-trivial feature engineering and complex analysis, we evaluated it on the **Nomad2018 Predicting Transparent Conductors** task from the **MLE Bench** (Machine Learning Engineering Benchmark).
>
> This Kaggle competition requires participants to write code for analyzing the molecular structure of materials and predicting their key performance indicators. This problem is **highly representative of real-world scientific discovery** as it demands domain knowledge and advanced ML engineering.
>
> HELIX achieved the second-best result among human submissions for this competition:
>
> | Method                                                  | Direct Prompt (BO64) | Gold Medal Threshold | Human second place | HELIX  | Human best |
> | ------------------------------------------------------- | -------------------- | -------------------- | ------------------ | ------ | ---------- |
> | root mean square logarithmic error (lower means better) | 0.0603               | 0.0559               | 0.0511             | 0.0492 | 0.0490     |
>
> This result underscores two critical points:
>
> 1. **Genuine Discovery Capability:** HELIX, utilizing a smaller base model, was able to **significantly outperform strong human submissions**, demonstrating a capability that goes far beyond simple optimization within a fixed template.
> 2. **Performance on Complex Scientific Engineering:** Success in the Nomad2018 Kaggle competition confirms HELIX’s ability to effectively generate surprising and effective solutions for challenging, **open-ended scientific engineering problems**. This task is highly specialized, requiring contestants to **analyze the atomic structure (based on elemental coordinates) of materials to predict specific key performance indicators**—a capability directly relevant to real-world scientific discovery and material design.
>
> ### Constraints on Real-World Problem Evaluation
>
> While the Nomad task provides strong evidence, directly evaluating our framework on massive, full-scale industrial problems is limited by practical constraints:
>
> - **Speed and Cost of Evaluation:** For tasks like circuit design or materials science, a single high-fidelity simulation or deep learning model training run can take **hours or days**. This makes the task computationally infeasible for a research project.
> - **Resource Requirements:** Validating solutions for real-world deep learning tasks requires access to vast **GPU clusters**, and high-performance simulations demand specialized supercomputing environments, which are typically beyond the scope of a standard benchmark.
>
> ### Ideas for Future Experiments
>
> Given these constraints, our short-term strategy is to select **problem prototypes** that distill the core capabilities required for real-world tasks, and our long-term vision focuses on integrating better external models:
>
> * **Short-Term**: Continue focusing on **high-fidelity prototypes** that isolate core scientific capabilities, just like what we selected for our evaluation: **Physics Simulation** (understanding the physical world), **Machine Learning Benchmarks** like **MLE-Bench(solving practical problems)** , **Circle Packing** (mastery of combinatorics/geometry), and **Function Minimization** (novel algorithmic design). The proven capability on these prototypes is directly transferable.
> * **Long-Term**:
>   1. **Integration of Effective World Models:** Rely on more efficient external models, such as **PDE-solving proxy models** or refined **LLM-based Reward Models**, to speed up and stabilize evaluation.
>   2. **Building and Using better Computing Infra:**  use better clusters and build better infra to allow for effective, large-scale, and time-intensive validation of truly industrial-scale problems.

---

> > ### Comment · Reviewer_TewV · 2025-11-25
> >
> > Thank you for the detailed response. Your additional discussions and new results are largely convincing and address my technical concerns.
> >
> > However, my primary concern now shifts to the scope of the necessary revisions. Given that the changes required are quite significant, I am concerned about the feasibility of incorporating them without a subsequent round of peer review to verify the final manuscript. Could you please comment on how you intend to integrate these major updates while ensuring the integrity of the original submission?

---

> ### Comment · Reviewer_TewV · 2025-11-27
>
> Thanks for the reply. It sounds concrete and doable to me.
> I look forward to the revised version and I will raise my evaluation to 6.
>
> I will express that opinion during the discussion among reviewers too.
> That said, if other (meta)reviewers think another round of peer review is better, I would also say that's fair.
>
> All the best to the authors.

---

### Official Review · Reviewer_RxUg · 2025-10-31

**Soundness:** 2
**Presentation:** 2
**Contribution:** 2
**Rating:** 4
**Confidence:** 2

**Summary:**

The paper proposes HELIX, a framework to learn an LLM policy that iteratively produces diverse yet high quality solutions for open-ended problems.

**Strengths:**

- The paper studies a setting of current interest to the community: open-ended problem solving.
- The approach proposed in the paper, HELIX, is novel and interesting (though some details are unclear).
- The experiments show the efficacy of the proposed method on interesting scientific reasoning tasks.

**Weaknesses:**

- My primary concern with the paper is with the description of the algorithm. It's unclear how NSGA-II is integrated into the pipeline. For instance, in Figure 2, the solutions selected via NSGA-II are passed to the RL algorithm as feedback. How does that work?
- While the experiments are interesting, on many tasks, the Direct Prompt baseline works pretty well already. Additionally, on some tasks, Open Evolve performs worse than Direct Prompt. The authors should use more challenging domains to evaluate their method.

**Questions:**

See weaknesses.

---

> ### Author Response · Authors · 2025-11-20
> **Reply to Weakness 1**
>
> ## W1: Detailed description of the algorithm
>
> We appreciate the reviewer’s thoughtful comments regarding the algorithmic framework. To provide a clearer understanding of the underlying mechanism of the HELIX framework, we present below a detailed explanation.
>
> Formally, the algorithm begins with the problem description $p$ and one (or a set of) initial solution(s) $s_0$, which form the initial dataset $\mathcal{D}_0 = \lbrace s_0\rbrace$. The evolutionary population is also initialized as $\mathcal{P}_0 = \mathcal{D}_0$.
>
> For the $t$-th iteration, the framework proceeds as follows:
>
> 1. **Sample from population**:
>    Given the batch size $B$, we randomly select $B$ samples from the current population $\mathcal P_t$, denoted as $\lbrace s_{t, i}\rbrace _{1 \le i \le B}$.
>
> 2. **Prompt construction**:
>    To leverage the **in-context learning** ability of LLMs, we construct the prompt $q_i$ for each sampled state $s_{t,i}$ by integrating itself and its $n$ ancestral states in the lineage tree (a historical trace of the solution $s_{t,i}$'s iterative refinement), i.e., $\lbrace f^{(k)}(s_{t, i})\rbrace_{1\le  k < n}$, along with their corresponding **rewards** $\lbrace R(f^{(k)}(s_{t,i}))\rbrace_{k < n}$ and **feedbacks** $\lbrace F(f^{(k)}(s_{t,i}))\rbrace_{k < n}$.
>    Formally,
>
>    $$
>    q_i = \text{ConstructPrompt}(\lbrace p\rbrace  \cup \lbrace s_{t,i}, R(s_{t,i}), F(s_{t,i})\rbrace  \cup \lbrace f^{(k)}(s_{t,i}), R(f^{(k)}(s_{t,i})), F(f^{(k)}(s_{t,i}))\rbrace _{1\le k<n}).
>    $$
>    Here, $R(\cdot)$ represents the reward function that measures the quality of a generated solution, and $F(\cdot)$ denotes the auxiliary feedback (e.g., textual or structured evaluations) provided by the evaluator to guide future refinements.
>
>    * **Explanation of lineage tree and ancestral states**:
>      In evolutionary algorithms, each new state $s_{t+1}$ is derived from a previous state $s_t$ through a mutation $a_t$, i.e., $s_{t+1} = T(s_t, a_t)$, where $T$ is the transition function. This recursive relationship naturally forms a tree structure — the **lineage tree** — and the **ancestral states** of $s_t$ are the preceding nodes along its evolutionary path.
>
> 3. **Model rollout and evaluation**:
>    The policy model $\pi_\theta$ generates $G$ rollouts for each input prompt $q_i$, resulting in
>
>    $$
>    \lbrace a_{i,j}\rbrace_{1 \le i \le B,\, 1 \le j \le G},
>    $$
>    and we can apply such mutation to current state $s_{t,i}$ and get $s_{t+1,i,j} = T(s_{t,i}, a_{i,j})$. After evaluation, we have the reward and textual feedback for each state:
>    $$
>    r_{t+1, i,j} = R(s_{t+1, i, j}), \quad f_{t+1, i,j} = F(s_{t+1, i, j}).
>    $$
>    The feedback $f_{t+1,i,j}$ is retained as contextual information for future iterations.
>
> 4. **Reinforcement learning update**:
>    The rewards are used to compute the normalized advantage:
>
>
>    $$
>    \tilde r_{t+1,i,j} = \frac{r_{t+1,i,j} - mean_j \lbrace r_{t+1,i,j} \rbrace }{std_j \lbrace r_{t+1,i,j}\rbrace },
>    $$
>
>    and the policy parameters are updated according to the GRPO objective:
>    $$
>    \theta \leftarrow \theta - \gamma \cdot \nabla_\theta \mathcal{L}_{\text{GRPO}}.
>    $$
>
> 5. **Diversity estimation**:
>    An embedding model $E$ encodes each new solution into a latent representation:
>    $$
>    h_{t+1,i,j} = E(s_{t+1, i, j}).
>    $$
>    The diversity score $\text{Div}(s)$ is then computed according to Eq. (6) in the updated manuscript. (Eq. (5) for the old version of manuscript).
>
> 6. **Evolutionary population construction**:
>    To balance **diversity** and **quality**, we employ the **NSGA-II** algorithm to construct the next population. NSGA-II is a multi-objective selection mechanism that identifies solutions with both high reward $R(s)$ and high diversity $\text{Div}(s)$.
>    The updated population is:
>
>    $$
>    \mathcal P_{t+1} = SelectTop(\bigcup_{0\le s\le t+1}\mathcal{D}_s).
>    $$
>
> We have included these details in the revised manuscript to further enhance the completeness and clarity of the method description.

---

> ### Author Response · Authors · 2025-11-20
> **Reply to Weakness 2**
>
> ## W2.1: Open Evolve performs worse than Direct Prompt
>
> We acknowledge the observation that OpenEvolve performs worse than the Direct Prompt baseline in some cases. We attribute this discrepancy primarily to the following two reasons:
>
> 1. **Sensitivity to Initialization and Search Strategy**: In our implementation, the Direct Prompt baseline performs **64 random evaluations**, sampling the search space broadly. In contrast, the OpenEvolve algorithm begins by generating several seed prompts and then iteratively evolves them. This evolutionary design introduces a high degree of **sensitivity to initialization**: if the selected seeds are suboptimal, subsequent evolution—even with hundreds of iterations—may become stuck in a local optimum and fail to converge to a desirable result.
> 2. **High Variance in Direct Prompt Rollouts**: We have observed that in some Direct Prompt trials, the reward deviation between different rollouts is notably high. This suggests the Direct Prompt's average score is often inflated by rare, successful runs. For example, in our direct prompt experiment using GPT-4o for the Adult Income task, the model produced only **1 successful rollout** with a reward of 76.91, while the remaining **63 rollouts** received a reward of 0.00 due to compilation errors or illegal outputs.
>
> ## W2.2: Selecting more challenging domains to evaluate the method
>
> Firstly, we argue that the difficulty of improving solution for open-ended scientific problem is not always linearly reflected by the size of the gain in metric. Even **minute improvements can signify immense underlying challenges**.
>
> For instance, in the **Circle Packing problem**, Alpha Evolve improved upon the best-known human result of 2.634 to 2.635863. Following this, many attempts were made by the community, with the best-known subsequent result reaching 2.635927. We have further advanced this frontier, achieving an upper bound of 2.635983. Similarly, for other complex problems, particularly in domains like Machine Learning and Function Minimization, what appears to be a small metric improvement often represents a significant and non-trivial advancement. Our current results are thus sufficiently demonstrative of our method's capability in tackling challenging problems.
>
> Secondly, **the problems we study are not merely toy settings.**  In particular, the physical simulation tasks and tabular dataset we consider have close analogues in real industrial workflows, where problem scale and complexity are comparable to those in our experiments. Thus, our evaluation targets scenarios that are meaningfully representative of real-world scientific and engineering applications.
>
> Furthermore, we selected a scientific task from **MLE Bench** (a benchmark proposed by OpenAI to evaluate LLM agents on machine learning engineering) to further demonstrate our method's performance on challenging open-ended scientific problems: **Nomad2018 Predicting Transparent Conductors**. It is a competition on Kaggle which requires participants to write code for analyzing the molecular structure of the material and predicting their key performance indicators as Transparent Conductors. Our proposed algorithm, HELIX, achieved the **second-best result among human submissions** for this problem:
>
> | Method                                                  | Direct Prompt (BO64) | Gold Medal Threshold | Human 2rd place | HELIX  | Human best |
> | ------------------------------------------------------- | -------------------- | -------------------- | ------------------ | ------ | ---------- |
> | root mean square logarithmic error (lower means better) | 0.0603               | 0.0559               | 0.0511             | 0.0492 | 0.0490     |
>
> This result underscores HELIX's ability to utilize a smaller model to generate surprising **effectiveness on challenging scientific problems**, significantly **outperforming the base model.**

---

> ### Author Response · Authors · 2025-11-26
> **Checking for Any Additional Comments**
>
> Dear Reviewer,
>
> I hope this message finds you well. As the discussion period is nearing its end with **only one week remaining**, I wanted to ensure we have addressed all your concerns satisfactorily. If there are any additional points or feedback you'd like us to consider, please let us know. Your insights are invaluable to us, and we're eager to address any remaining issues to improve our work.
>
> Thank you for your time and effort in reviewing our paper.

---

### Official Review · Reviewer_Zrt5 · 2025-11-01

**Soundness:** 4
**Presentation:** 4
**Contribution:** 3
**Rating:** 8
**Confidence:** 3

**Summary:**

This paper introduces HELIX, a hierarchical evolutionary reinforcement learning framework that combines GRPO-based policy optimization, multi-objective evolutionary search (via NSGA-II), and in-context learning for solving open-ended scientific problems. The authors argue that HELIX enables LLMs to balance exploration and exploitation while iteratively improving solution quality. The method is tested on 19 diverse tasks (physics simulations, machine learning, symbolic regression, circle packing, and mathematical optimization) and achieves superior results on 16 tasks compared to strong baselines, including GPT-4o.

**Strengths:**

- Novel combination of reinforcement learning, evolutionary algorithms, and in-context prompting, addressing key limitations of existing approaches (e.g., entropy collapse, lack of diversity).

- Strong empirical results across a wide range of domains, including physical design and scientific optimization, demonstrating the generality of the method.

- Clear framework design and well-articulated motivation connecting HELIX’s components to the nature of open-ended scientific discovery.

- Ablation and scaling analyses provide convincing evidence that both diversity-aware evolution and reinforcement learning contribute meaningfully to performance.

**Weaknesses:**

- While HELIX shows impressive performance across various scientific tasks, the paper provides limited discussion on the sensitivity of the framework to its key hyperparameters. In particular, the reward normalization constants (e.g., denominators used in Eq. 6–10 for physics tasks) and NSGA-II parameters (e.g., population size, crowding distance, KNN-based diversity metric) appear to play a crucial role in balancing exploration and exploitation, yet no analysis is offered regarding their influence on stability or convergence. This raises concerns about how robust HELIX remains under different hyperparameter settings.

**Questions:**

- How sensitive is HELIX to the reward normalization constants and NSGA-II hyperparameters?

---

> ### Author Response · Authors · 2025-11-20
>
> ## Q1: The sensitivity of our method to the reward normalization constants
>
> We thank the reviewer for highlighting this detail. The reward normalization constants in Eq. (6)–(10) are indeed domain-specific calibration metrics, typically set by experts based on the score of the provided initial solution to maintain numerical stability.
>
> However, we would like to clarify that **our method is theoretically invariant to the reward scale**, meaning the absolute magnitude of the reward normalization constants does not affect the algorithm's performance. This robustness stems from two key reasons:
>
> 1. **For GRPO Algorithms**: In GRPO algorithm, the advantage of rollouts is calculated via group-based standardization: $\hat A_{i,t} = \frac{r_i - \text{mean}(r_i)}{\text{std}(r_i)}$, where $r_i$ is the reward for the $i$-th rollout. If the reward is scaled by $s$, the numerator and denominator will also be scaled by $s$. These factors cancel out, leaving the advantage $\hat A_{i,t}$ and the resulting gradients identical.
> 2. **For Evolutionary Algorithms**: The evolutionary component relies exclusively on the **relative partial order** (Pareto dominance) of solutions to perform non-dominated sorting. Since linear scaling preserves the relative order of values, the selection of solutions for the population remains unaffected by the reward scale.
>
> ## Q2: The sensitivity of our method to the parameters of evolutionary algorithms
>
> We appreciate your thoughtful comments regarding the influence of our method's hyperparameters.
>
> 1. **Population Size**: we currently fixed the population size $|\mathcal P_t|$ to batchsize $B$. This is primarily due to we only take $B$ samples from the population $\mathcal P_t$ at iteration $t$, and we will construct a new population $\mathcal P_{t+1}$ by incorporating the newly generated solutions into the dataset. Thus, if the size exceeds $B$, the samples that are not selected would be discarded in the next iteration. Such a setting would introduce unnecessary stochasticity without a clear theoretical justification that it would enhance the algorithm's performance.
>
> 2. **Crowding Distance**: The crowding distance is not a hyperparameter that we can adjust. It is a metric calculate by NSGA-II algorithm itself, which is formally defined as:
>    $$
>    Dist = \frac{R(s_{r+})-R(s_{r-})} {R_{max} - R_{min}} + \frac{\text{Div}(s_{d+})-\text{Div}(s_{d-})}{D_{max} - D_{min}},
>    $$
>    where $s_{r+}, s_{r-}, s_{d+}, s_{d-}$ represents the solutions immediately above and below $s$ in the reward and diversity rankings, respectively.
>
> 3. **KNN-based diversity metric**: Regarding the sensitivity to the parameter k in our KNN-based diversity metric, we conducted an experiment on the Adult Income task using different values for the number of nearest neighbors (k). The results are shown below:
>
> | k  | Performance of task Adult Income |
> | ---- | -------------------------------- |
> | 1    | 81.88                            |
> | 5    | 82.07                            |
> | 10   | 81.86                            |
> | 100  | 81.63                            |
>
> As you can see, the algorithm is relatively robust regards to the change of $k$. This stability is expected because k in this context primarily serves as a computational technique to accelerate the diversity calculation by estimating local density. It is not intended to introduce dramatic changes to the underlying estimation of diversity for each solution.

---

> ### Author Response · Authors · 2025-11-26
> **Checking for Any Additional Comments**
>
> Dear Reviewer,
>
> I hope this message finds you well. As the discussion period is nearing its end with **only one week remaining**, I wanted to ensure we have addressed all your concerns satisfactorily. If there are any additional points or feedback you'd like us to consider, please let us know. Your insights are invaluable to us, and we're eager to address any remaining issues to improve our work.
>
> Thank you for your time and effort in reviewing our paper.

---

### Official Review · Reviewer_c9MA · 2025-11-02

**Soundness:** 3
**Presentation:** 1
**Contribution:** 2
**Rating:** 4
**Confidence:** 3

**Summary:**

This paper introduces HELIX, a hybrid method that enhances the reasoning capabilities of LLMs by combining reinforcement learning (RL) with verifiable rewards, evolutionary algorithms, and in-context learning. The authors argue that the RL fine-tuning (GRPO) phase increases domain-agnostic adaptation, the evolutionary nature promotes the solution diversity, and the in-context learning improves the solution quality using previous diverse high-quality solutions. This method is validated on several scientific and engineering tasks, showing improvement over baselines.

**Strengths:**

1. This work effectively combines the known strategies for reasoning with LLMs, the RL fine-tuning, evolutionary algorithms, and in-context learning. To my knowledge, this combination is new.
2. The proposed method shows strong empirical results compared to baselines across many scientific problems, showing its practicality.
3. The source code is provided.

**Weaknesses:**

1. (Clarity) One of my main concerns is the clarity of the paper. While this work is understandable and easy-to-read at a high level, there is some ambiguity at low levels and many details seem to be missing. I left many questions to clarify some points that I couldn't understand; see the questions below. Additionally, I believe providing a detailed formalised algorithm and LLM prompts used for each experiment would be valuable.
2. (Method) I believe GRPO is an on-policy RL algorithm, but the algorithm seems to train it with off-policy samples from the previous steps (since $(q,a) \sim \mathcal{D}$ where $\mathcal{D}$ is previously discovered solutions). Please provide any rationale for this choice, or correct me if I'm wrong.
3. (Evaluation) Some necessary evaluation settings seem to be missing, especially the standard deviation or confidence interval for each experiment. I assume that the proposed method consumes significantly more LLM calls than the current baselines (direct prompt, open evolve, task-specific methods), as it requires fine-tuning. Therefore, I believe that the fine-tuning method, e.g., vanilla GRPO, should also be included as a main baseline for Table 1.

(Minor)
4. Line 90: Please add the reference for GRPO
5. Line 92: Using "NSGA-II" without any explanation makes readers confused; a short explanation of it (at least that it is an evolutionary algorithm) would be helpful.
6. Line 115-117: The related works are only roughly introduced. Providing more details regarding them would be valuable, e.g., how KL-Conv addressed the limitation, what the memory-less RL is, and why it cannot leverage previous solutions.

**Questions:**

1. Line 158, Eq. (2): does this objective guarantee Eq. (1)?
2. Line 158, Eq. (2): why you take average over $s_t\sim \mathcal{P}$, not $\sim \mathcal{D}$?
3. Figure 2: What is the lineage tree? An explanation seems to be needed for it.
4. Line 211, Eq. (4): a) What is the formal definition of $q$? b) What is the formal definition of $\hat{A}$? c) Can you define the MDP for your GRPO fine-tuning?
5. Line 245, Eq. (5): a) Is this diversity measure new to this work, or borrowed from other works? b) It seems to require the pairwise distance calculation. Isn't it computationally heavy?
6. Is the LLM finetuned to be used for different kinds of tasks? For example, for the Machine Learning benchmark, is a single LLM used for all the subtasks (Adult Income, Bank Marketing, and Boston Housing), or do we need to finetune an LLM for each subtask?
7. How to generate the initial solutions?
8. Doesn't HELIX suffer from a similar limitation as the evolutionary-algorithm-based approaches, designing problem-specific prompts for each task? If I understand correctly, HELIX needs prompts to combine previous solutions to generate a new one, just like the EAs need prompts to mix (crossover/mutation) solutions. How did you compare the complexity of designing the prompt between your algorithm and EAs?
9. Line 253, what does "nondominated sorting procedure" mean?
10. Line 256, what do "a crowding distance measure" and the "objective space" mean?
11. Is "Evolutionary Search" in the title of section 3.2 indicating the in-context learning with previous solutions ("ancestral states" in Line 220)?

---

### LLM usage disclosure
I did not use an LLM for my review, but I used grammar check software.

---

> ### Author Response · Authors · 2025-11-20
> **Reply to Weakness 1**
>
> ## W1.1: Detailed description of the algorithm
> We appreciate the reviewer’s thoughtful comments regarding the algorithmic framework. To provide a clearer understanding of the underlying mechanism of the HELIX framework, we present below a detailed explanation.
>
> Formally, the algorithm begins with the problem description $p$ and one (or a set of) initial solution(s) $s_0$, which form the initial dataset $\mathcal{D}_0 = \lbrace s_0\rbrace$. The evolutionary population is also initialized as $\mathcal{P}_0 = \mathcal{D}_0$.
>
> For the $t$-th iteration, the framework proceeds as follows:
>
> 1. **Sample from population**:
>    Given the batch size $B$, we randomly select $B$ samples from the current population $\mathcal P_t$, denoted as $\lbrace s_{t, i}\rbrace _{1 \le i \le B}$.
>
> 2. **Prompt construction**:
>    To leverage the **in-context learning** ability of LLMs, we construct the prompt $q_i$ for each sampled state $s_{t,i}$ by integrating itself and its $n$ ancestral states in the lineage tree (a historical trace of the solution $s_{t,i}$'s iterative refinement), i.e., $\lbrace f^{(k)}(s_{t, i})\rbrace_{1\le  k < n}$, along with their corresponding **rewards** $\lbrace R(f^{(k)}(s_{t,i}))\rbrace_{k < n}$ and **feedbacks** $\lbrace F(f^{(k)}(s_{t,i}))\rbrace_{k < n}$.
>    Formally,
>
>    $$
>    q_i = \text{ConstructPrompt}(\lbrace p\rbrace  \cup \lbrace s_{t,i}, R(s_{t,i}), F(s_{t,i})\rbrace  \cup \lbrace f^{(k)}(s_{t,i}), R(f^{(k)}(s_{t,i})), F(f^{(k)}(s_{t,i}))\rbrace _{1\le k<n}).
>    $$
>    Here, $R(\cdot)$ represents the reward function that measures the quality of a generated solution, and $F(\cdot)$ denotes the auxiliary feedback (e.g., textual or structured evaluations) provided by the evaluator to guide future refinements.
>
>    * **Explanation of lineage tree and ancestral states**:
>      In evolutionary algorithms, each new state $s_{t+1}$ is derived from a previous state $s_t$ through a mutation $a_t$, i.e., $s_{t+1} = T(s_t, a_t)$, where $T$ is the transition function. This recursive relationship naturally forms a tree structure — the **lineage tree** — and the **ancestral states** of $s_t$ are the preceding nodes along its evolutionary path.
>
> 3. **Model rollout and evaluation**:
>    The policy model $\pi_\theta$ generates $G$ rollouts for each input prompt $q_i$, resulting in
>
>    $$
>    \lbrace a_{i,j}\rbrace_{1 \le i \le B,\, 1 \le j \le G},
>    $$
>    and we can apply such mutation to current state $s_{t,i}$ and get $s_{t+1,i,j} = T(s_{t,i}, a_{i,j})$. After evaluation, we have the reward and textual feedback for each state:
>    $$
>    r_{t+1, i,j} = R(s_{t+1, i, j}), \quad f_{t+1, i,j} = F(s_{t+1, i, j}).
>    $$
>    The feedback $f_{t+1,i,j}$ is retained as contextual information for future iterations.
>
> 4. **Reinforcement learning update**:
>    The rewards are used to compute the normalized advantage:
>
>
>    $$
>    \tilde r_{t+1,i,j} = \frac{r_{t+1,i,j} - mean_j \lbrace r_{t+1,i,j} \rbrace }{std_j \lbrace r_{t+1,i,j}\rbrace },
>    $$
>
>    and the policy parameters are updated according to the GRPO objective:
>    $$
>    \theta \leftarrow \theta - \gamma \cdot \nabla_\theta \mathcal{L}_{\text{GRPO}}.
>    $$
>
> 5. **Diversity estimation**:
>    An embedding model $E$ encodes each new solution into a latent representation:
>    $$
>    h_{t+1,i,j} = E(s_{t+1, i, j}).
>    $$
>    The diversity score $\text{Div}(s)$ is then computed according to Eq. (6) in the updated manuscript. (Eq. (5) for the old version of manuscript).
>
> 6. **Evolutionary population construction**:
>    To balance **diversity** and **quality**, we employ the **NSGA-II** algorithm to construct the next population. NSGA-II is a multi-objective selection mechanism that identifies solutions with both high reward $R(s)$ and high diversity $\text{Div}(s)$.
>    The updated population is:
>
>    $$
>    \mathcal P_{t+1} = SelectTop(\bigcup_{0\le s\le t+1}\mathcal{D}_s).
>    $$
>
> We have included these details in the revised manuscript to further enhance the completeness and clarity of the method description.
> ## W1.2: LLM Prompts used for each experiments
>
> Thank you for your valuable suggestion. We have added the LLM prompts used for each experiment in the **Appendix D** .
>
> As shown in the **Appendix D**, our prompts mainly contain the **problem description**, the **required output format**, and the **initial solution**. We did not heavily incorporate task-specific priors or expert hints into the prompts.

---

> ### Author Response · Authors · 2025-11-20
> **Reply to Weakness 2-3**
>
> ## W2: Does HELIX train GRPO with off-policy samples?
>
> No, we train the model in a fully **on-policy** manner. In fact, the confusion here comes from a **notation issue**: in most GRPO-related literature, the symbol $a$ usually denotes the *standard ground-truth answer*. However, in our setting, it denotes open-ended scientific problems without a single correct answer.
>
> In the paper, we first introduce the original GRPO algorithm, and then explain how our method adopts it. For this reason, Eq. (4) directly follows the **original GRPO formulation**, including the notation $a$. We will refine the presentation in future revisions to avoid such misunderstandings.
>
> For more details of our GRPO training method, please refer to our official comment titled Method Detail, as well as our responses to **Q2** and **Q4**.
>
> ## W3.1: Standard deviation or confidence interval for each experiment.
>
> We appreciate the reviewer’s suggestion. We fully agree that reporting standard deviations or confidence intervals is standard practice in typical machine learning experiments, especially for small-scale model training or fine-tuning.
>
> However, for large-scale LLM training or post-training, computing standard deviations or confidence intervals is prohibitively resource-consuming, as each run requires substantial computation time and cost. Within our available budget and timeline, it is not feasible to repeat all experiments multiple times. Importantly, reporting standard deviations is not a universally adopted practice in recent large-model training or post-training works. For example, several outstanding NeurIPS 2025 Oral or Spotlight papers do not include standard deviation or confidence interval results:
>
> - **Large Language Diffusion Models** (NeurIPS 2025 Oral):
>   - Paper: https://arxiv.org/abs/2502.09992
>   - Conference link: https://neurips.cc/virtual/2025/loc/san-diego/poster/118608
> - **DAPO: Improving Multi-Step Reasoning Abilities of Large Language Models with Direct Advantage-Based Policy Optimization** (NeurIPS 2025 Spotlight):
>   - Paper: https://arxiv.org/abs/2412.18279
>   - Conference link: https://neurips.cc/virtual/2025/loc/san-diego/poster/119738
>
> Nevertheless, we acknowledge the value of reporting standard deviations to reflect the stability and robustness of our method. Therefore, we repeated a subset of task in our experiment three times and summarize the results below:
>
> | Task          | Run 1 | Run 2 | Run 3  | Avg    | Std        |
> | ------------- | ----- | ----- | ------ | ------ | ---------- |
> | Adult         | 82.07 | 82.06 | 81.87  | 82.00  | 9.201×10⁻² |
> | BankMarketing | 80.65 | 80.50 | 80.32  | 80.49  | 1.349×10⁻¹ |
> | Inductor      | 9.609 | 9.011 | 12.373 | 10.331 | 1.464×10⁰  |
> | Keanes 30d    | 0.994 | 0.997 | 1.000  | 0.997  | 2.456×10⁻³ |
>
>  Regarding the standard deviation, the results indicate that the performance across repeated trials is highly stable, with consistently small variances. This suggests that our method is robust and does not exhibit significant fluctuations across runs.
>
> ## W3.2: GRPO as a main baseline for Table 1
>
> Thank you for the valuable suggestion.
>
> In Section 4.3 Ablation Study, we have already reported the results of vanilla GRPO (denoted as *TrainOnly*) on the Boston Housing and Circle Packing tasks, together with a detailed analysis of its limitations. We also included the *EvoOnly* setting, which uses only our proposed Evolutionary Pipeline without any model training. Importantly, all these methods used the same number of LLM calls, demonstrating that the performance improvements of our approach come from the innovative combination of Evolution and Reinforcement Learning, rather than from additional sampling.
>
> However, we agree that vanilla GRPO should also be included as a baseline in the main experiments. We have now collected GRPO results for more tasks. The comparison between GRPO and our method is shown below:
>
> | Task                       | GRPO  | Ours  |
> | -------------------------- | ----- | ----- |
> | Adult $\uparrow$           | 81.78 | 82.07 |
> | BankMarketing $\uparrow$   | 79.59 | 80.65 |
> | BostonHousing $\downarrow$ | 2.844 | 1.747 |
> | Circle Packing $\uparrow$  | 2.612 | 2.636 |
> | Inductor $\uparrow$        | 6.888 | 9.609 |
> | Keanes 30d $\uparrow$      | 0.986 | 0.994 |
>
> As we can see, our method outperforms vanilla GRPO on every task.
>
> Furthermore, during training, we frequently observe instability of GRPO on scientific problems: it often falls into local optima or even produces zero reward during long training periods due to compilation errors or solutions violates constraints. These issues highlight the limitations of memory-less RL, which cannot leverage previously discovered solutions and the corresponding textual feedback without contextual support. In contrast, HELIX achieves improvements in both performance and robustness through its integration of Evolutionary Algorithms and In Context Learning.

---

> ### Author Response · Authors · 2025-11-20
> **Reply to Weakness 4-6, Question 1-4(b)**
>
> ## W4,5,6: Detailed Improvements for Clarity
>
> We appreciate your helpful comment. The discussions on GRPO, NSGA-II, and Related Work have been revised and improved in the updated version of the manuscript.
>
> ## Q1: Does objective in Eq. (2) guarantee Eq. (1)?
>
> First of all, for open-ended scientific problems, it is beyond human capability to find the globally optimal solution $s$ that maximize the reward $R(s)$ in Eq. (1). Therefore, we propose to iteratively refine the current solution $s_t$ in the population $\mathcal P$ through an action $a$ performed by LLM.
>
> Given the population distribution $\mathcal P$, we optimize the LLM parameters using the objective in Eq. (2). Assuming the optimization works effectively and improves policy parameters from $\theta_1$ to $\theta_2$, which satisfies:
> $$
> \mathbb E_{s_t\sim \mathcal P, a_t^{(1)}\sim \pi_{\theta_1}}[R(s_t,a_t^{(1)})]\le \mathbb E_{s_t\sim \mathcal P, a_t^{(2)}\sim \pi_{\theta_2}}[R(s_t,a_t^{(2)})]
> $$
> where $R(s_t, a_t^{(k)}) = R(s_{t+1}^{(k)})$ by definition. Consequently, we obtain:
> $$
> \mathbb E_{s_t\sim \mathcal P} [R(s_{t+1}^{(1)})]\le \mathbb E_{s_t\sim \mathcal P} [R(s_{t+1}^{(2)})]
> $$
> which suggests the newly generated solution $s_{t+1}^{(2)}$ by the optimized model is expected to outperform $s_{t+1}^{(1)}$.
>
> To conclude, for a given $\mathcal P$, if we can found a global optimum $\theta^*$ in Eq. (2), the **Expectation** of Eq. (1) will also be maximized. This demonstrates Eq. (2) serves as a good approximation of Eq. (1) and improvements on the proxy objective in Eq. (2) will finally lead to improvements on Eq. (1).
>
> ## Q2: why take average over $s_t\sim \mathcal P$ but not $\mathcal D$
>
> This is the key distinction between standard GRPO optimization and our proposed **HELIX** framework.
>
> In vanilla GRPO and most reinforcement learning paradigms, the objective is to maximize the reward over a **fixed dataset** $\mathcal D$:
> $$
> \max_\theta \mathbb E_{q\sim \mathcal D, a\sim \pi_\theta(\cdot |q)} [ R(a)]
> $$
> However, to truly “stand on the shoulders of giants,” it is necessary to iteratively incorporate feedback from past trials and refine solutions over time, which means the dataset itself should **dynamically evolve** by integrating past results, leading to the following objective:
> $$
> \max_\theta \mathbb E_{(q,s_t)\sim \mathcal D, a_t\sim \pi_\theta(\cdot|q, s_t)} [R(s_t, a_t)]
> $$
> Nevertheless, we observed that the states $s_t\sim \mathcal D$ often contain **failure trials** (with low reward) and **duplicate solutions** (with low diversity), both of which can degrade the quality and diversity of future generated results. Thus we employ an evolutionary algorithm to **dynamically select valuable samples** from $\mathcal D$, which forms the **population** $\mathcal P$ that actually used for training and rollout, leading to the final version of our objective:
> $$
> \max_\theta \mathbb E_{(q,s_t)\sim \mathcal P, a_t\sim \pi_\theta(\cdot|q, s_t)} [R(s_t, a_t)]
> $$
> which is the same as Eq. (2).
>
> ## Q3: What is the lineage tree?
>
> The concept of the lineage tree has been explained in the detailed description of the method. Please refer to our official comment titled Method Detail.
>
> ## Q4(a): What is the formal definition of $q$?
>
> In short, $q$ refers to the **prompt** provided to the LLM, which can be formally defined as:
> $$
> q_i = ConstructPrompt(\lbrace p\rbrace  \cup \lbrace s_{t,i}, R(s_{t,i}), F(s_{t,i})\rbrace  \cup \lbrace f^{(k)}(s_{t,i}), R(f^{(k)}(s_{t,i})), F(f^{(k)}(s_{t,i}))\rbrace_{1\le k<n}).
> $$
> Here, $R(\cdot)$ represents the reward function that measures the quality of a generated solution, $F(\cdot)$ denotes the auxiliary feedback (e.g., textual or structured evaluations) provided by the evaluator to guide future refinements, and $\lbrace f^{(k)}(s_{t, i})\rbrace_{1\le  k < n}$ is its $n$ ancestral states in the lineage tree (a historical trace of the solution $s_{t,i}$'s iterative refinement)
>
> For further clarification, please refer to our official comment titled Method Detail.
>
> ## Q4(b): What is the formal definition of $\hat A$?
>
> In short, $\hat A$ denotes the **token-level advantage** computed in the vanilla GRPO. It is formally defined as:
>
> $$
> \tilde A_{i,j,k} = \tilde r_{i,j} =\frac{r_{i,j} - mean_j\lbrace r_{i,j}\rbrace }{std_j\lbrace r_{i,j}\rbrace },
> $$
> where $i$ denotes the $i$-th batch, $j$ represents the $j$-th rollout within the same GRPO group, and $k$ refers to the $k$-th token position.

---

> ### Author Response · Authors · 2025-11-20
> **Reply to Question 4(c)-6**
>
> ## Q4(c): Can you define the MDP for your GRPO fine-tuning?
>
> We formally define the MDP process of our GRPO fine-tuning as follow:
> $$
> \mathcal M = (S, A, P, R)
> $$
> where:
>
> * $S$ is the state space. For every state $s_t\in S$ we have $s_k = q \cup o_{s\le t}$, where $q$ is the prompt and $o_{s\le t}$ is the first $t$ tokens of model output. Specifically, $s_0 = q$ is the prompt.
> * $A$ is the action space. Here, $A$ is the full token vocabulary.
> * $P$ is the transition function that satisfies $P(s_{t+1}|s_t,a) = \mathbf 1\lbrace o_{t+1} = a\rbrace $. In other words, this is a deterministic MDP that appends every output token $a$ to the current state.
> * $R$ is the reward function. We can first define a stopping time $T = \inf \lbrace t|o_t = \text{EOS}\rbrace $, then the reward function is $R(s_t, a_t) = \begin{cases}0&t<T\\\text{Eval}(s_t)&t=T\end{cases}$, where $\text{Eval}$ represents the evaluation score of the output.
>
> For the detail of prompt construction and evaluation, please refer to our official comment titled Method Detail.
>
> ## Q5: Is the diversity measure in Eq. (5) new to this work? Is it computationally heavy?
>
> This diversity measure is not entirely new; it follows a widely adopted design principle based on **cosine similarity**. Specifically, it measures the average cosine similarity between a sample and its **$k$-nearest neighbors**, and converts it into a diversity score using $(1 - \text{similarity})$.
>
> Given the parameter $k$, the overall computational complexity is $O(kn + n\log n)$, where $O(n\log n)$ accounts for the nearest-neighbor search and $O(kn)$ corresponds to the pairwise cosine similarity computations. In practice, we set $k = 5$ for our main experiments, making the method computationally efficient.
>
> ## Q6: Is the LLM finetuned to be used for different kinds of tasks?
>
> Our fine-tuning strategy varied based on the task type:
>
> 1. **For Symbolic Regression tasks:** Each discipline (e.g., Physics, Chemistry) contains approximately 20-40 distinct subtasks. For these, we fine-tuned **one model per discipline**, which handles all subtasks within that category.
> 2. **For other tasks:** We fine-tuned **a separate model for each individual subtask**.
>
> Importantly, our algorithm does **not inherently restrict** the model from being simultaneously fine-tuned on multiple different tasks.
>
> To specifically address your query, we conducted an additional experiment where a single model was simultaneously trained on four different Machine Learning tasks. The results are summarized below: (Predicting Transparent Conductors is a newly added machine learning task adopt from kaggle competition, where HELIX achieves 2nd place among human participants. See our reply to reviewer RxUg.)
>
> |                                                | Train Seperately | Train Simultaneously |
> | ---------------------------------------------- | ---------------- | -------------------- |
> | Adult Income $\uparrow$                        | 82.07            | 81.85                |
> | Bank Marketing $\uparrow$                      | 80.65            | 80.01                |
> | Boston Housing $\downarrow$                    | 1.747            | 2.810                |
> | Predicting Transparent Conductors $\downarrow$ | 0.0492           | 0.0532               |
>
> As shown, simultaneous training results in a minor performance drop across all tasks. However, the performance on three out of the four tasks (Adult Income, Bank Marketing, and Predicting Transparent Conductors) remains reasonably comparable to the models trained separately.
>
> This slight decrease in performance is likely attributable to two factors:
>
> 1. Under the same total training epochs, each individual task receives **fewer effective LLM calls** compared to the separate training setup.
> 2. Although the tasks share similar underlying solution methodologies, the **diverse application scenarios** are highly distinct in nature, leading to a degree of interference and performance degradation.
>
> However, we hypothesize that if the different tasks share mutually beneficial knowledge or involve similar solution methodologies, training them together would yield greater performance benefits (i.e., positive knowledge transfer). We plan to explore this hypothesis in future work.

---

> ### Author Response · Authors · 2025-11-20
> **Reply to Question 7**
>
> ## Q7: How to generate the initial solutions?
>
> Currently, the initial solution is a simple, handcrafted one to help the model understand the **function interface** and **interaction format**, with almost no embedded expert knowledge.
>
> The initial methods used across all tasks are summarized in the table below. These initial methods are kept consistent and visible to all LLM baselines.
>
> | Tasks                 | Method                                                       |
> | --------------------- | ------------------------------------------------------------ |
> | Adult Income          | Logistic Regression                                          |
> | Bank Marketing        | Logistic Regression                                          |
> | Boston Housing        | Linear Regression                                            |
> | Inductor              | See Figure 10                                                |
> | Beam Bending          | See Figure 8                                                 |
> | Magnetic Torque       | See Figure 7                                                 |
> | Periodic Heat         | See Figure 9                                                 |
> | Demultiplexer         | See Figure 6                                                 |
> | Circle packing tasks  | Arrange circles with fixed centers and radii. (Optimal solutions often includes search and optimization algorithms) |
> | Function Minimization | 10000 iterations of random sample in domain.                 |
> | Symbolic Regression   | Linear Regression                                            |
>
> In the future, more sophisticated initial solutions proposed by human experts could be incorporated, which has the potential to further enhance the overall performance of the algorithm.

---

> ### Author Response · Authors · 2025-11-20
> **Reply to Question 8-9**
>
> ## Q8: HELIX’s Requirement for Task-Specific Prompt Design and comparation to EAs.
>
> Firstly, as you can see in Appendix D, the prompts we use consist primarily of the problem description, formatting specifications, and the initial solution.  We do not include any task-specific instructions, hints or expert priors tailored to certain tasks (with the only exception of the circle-packing task, which is aligned with the OpenEvolve prompt because the base model lacks basic geometric priors). However,  OpenEvolve usually requires multi-stage prompts to reproduce AlphaEvolve-level results. Importantly, none of our tasks require prompts designed to explicitly mix solutions (e.g., for crossover or mutation), which is often needed in EA-based prompting strategies.
>
> This difference arises because HELIX fully leverages the model’s **in-context learning** capabilities and integrates NSGA-II to achieve a more effective **balance between quality and diversity**. The comparison between HELIX and EAs is summarized below:
>
> 1. **Context construction of existing evolutionary algorithms**:
>
>    Current evolutionary algorithms typically construct context using samples from the same island or population, even though these samples may be **weakly related** (e.g., produced by different parents or using entirely distinct strategies). For instance, in OpenEvolve—the open-source implementation of AlphaEvolve—the experience set is formed by mixing most-recent programs, top-performing programs, and random programs.
>
>    **Such heterogeneous and discontinuous context often confuses LLMs, making it difficult for them to meaningfully combine strengths across methods** unless extensive hand-crafted prompts and hints are provided.
>
> 2. **The context construction strategy of HELIX**:
>
>    In contrast, HELIX builds its prompt from the **evolutionary history** of the current solution, enriched with feedback from the evaluator. Specifically, we retrieve the ancestors of the current sample from the lineage tree, forming a coherent trajectory of incremental improvements. This mimics how humans solve problems: we reflect on why previous attempts succeeded or failed and derive actionable insights for the next iteration.
>
>    It is worth emphasize that **each constructed context represents a continuous evolutionary trajectory**, produced by the same model architecture and following similar solution styles. As a result, the model can readily assimilate experiences from prior trial-and-error steps without being overwhelmed by abrupt structural differences.
>
> 3. **How HELIX introduce diversity to evolution**:
>
>    The above strategy ensures coherent evolution but could bias the model toward a narrow solution space. To resolve this, HELIX incorporates NSGA-II to explicitly account for **diversity**. By selecting starting points with high diversity, HELIX can **explore multiple promising evolutionary directions in parallel**.
>
>    Additionally, a key practical trick is that the prompt must explicitly instruct the model to generate solutions that **differ from previous attempts**, rather than merely copying or imitating them. This simple instruction substantially increases diversity and leads to significantly better overall performance.
>
> ## Q9: What does "nondominated sorting procedure" mean?
>
> This is a key component of the NSGA-II algorithm. For a detailed implementation, please refer to **Algorithm 1 in Reference [1]**.
>
> Here, we provide a brief overview of its functionality.
>
> Given two different solutions $s_a$ and $s_b$, we can calculate their respective reward values $R(s_a)$, $R(s_b)$ and diversity scores $\text{Div}(s_a)$, $\text{Div}(s_b)$. We say that $s_a$ **dominates** $s_b$ (denoted as $s_a \succ s_b$) if $s_a$ is no worse than $s_b$ in all objectives and strictly better in at least one objective.
>
> This partial-order relation $\succ$ naturally induces a **topological structure** over the set of solutions. Each solution can be assigned a **front number**, indicating its level in the Pareto fronts. Formally, a solution $s_k$ belongs to the $k$-th front if there exists a shortest chain $s_1 \prec s_2 \prec \cdots \prec s_k$, and no solution $s \in \mathcal{D}$ satisfies $s_k \prec s$.
>
> The NSGA-II algorithm performs a **topological sort** on this graph and orders the solutions according to their front numbers. This procedure is referred to as **nondominated sorting**. Intuitively, it identifies solutions that lie on the Pareto fronts—those that are relatively strong in both reward and diversity—ensuring that higher-quality, diverse solutions are prioritized.

---

> ### Author Response · Authors · 2025-11-20
> **Reply to Question 10-11, Reference**
>
> ## Q10(a): What does "a crowding distance measure" mean?
>
> Crowding distance measure is also a key component of the NSGA-II algorithm. For the detailed definition and assignment procedure, please refer to **Algorithm 2 in Reference [1]**. We also provide a brief explanation here.
>
> In Q9, we described how NSGA-II sorts solutions by their **front numbers**. Let $\mathcal{F}_k$ denote the set of solutions in the $k$-th front. When constructing a population of size $M$, the algorithm first selects all solutions from fronts with smaller numbers and fills any remaining slots from the next front, and so on.
>
> However, if the number of required samples is smaller than $|\mathcal{F}_k|$, a subset of more **valuable solutions** must be chosen from $\mathcal{F}_k$. Intuitively, solutions with **large differences** in **Reward** and **Diversity** are considered more distinct and are more likely to generate diverse and high-quality offspring. Therefore, NSGA-II computes a **crowding distance measure** for each solution in $\mathcal{F}_k$ to quantify how different it is from other solutions in terms of reward and diversity.
>
> Formally, for each solution, the algorithm sorts all solutions in $\mathcal F_k$ by $R(s)$ and $\text{Div}(s)$, recording the neighboring solutions in each ordering as $s_{r+}, s_{r-}, s_{d+}, s_{d-}$, representing the solutions immediately above and below $s$ in the reward and diversity rankings, respectively.
>
> The **crowding distance** is then computed as:
> $$
> Dist = \frac{R(s_{r+})-R(s_{r-})} {R_{max} - R_{min}} + \frac{\text{Div}(s_{d+})-\text{Div}(s_{d-})}{D_{max} - D_{min}}
> $$
> In summary, within the same front, solutions that are **more isolated in reward and diversity** are preferred.
>
> ## Q10(b):  What does "objective space" mean?
>
> The objective space refers to the two-dimensional space formed by Reward and Diversity. Each solution can be represented as a point in this space according to its reward and diversity values.
>
> A set of solutions is **well spread in objective space** if they are distributed evenly in this space, and crowding distance is used to measure this spread and prioritize more isolated solutions within the same front.
>
> ## Q11: Is "Evolutionary Search" indicating the in-context learning with previous solutions?
>
> Partially yes—our evolutionary process indeed relies on in-context learning, which enables the model to iteratively refine algorithms, absorb experience, and autonomously innovate over successive generations. However, **this is only one component of HELIX’s evolutionary search**.
>
> The more essential aspect lies in **maintaining a high-quality population**, rather than naively feeding all past solutions into the context. Most historical solutions are **redundant or uninformative**, which would greatly reduce exploration efficiency. Some early trials are **overly naive**, making them poor candidates for further refinement and potentially hindering progress.
> Therefore, HELIX integrates **NSGA-II** as a core part of its evolutionary search: it systematically filters and maintains a diverse, high-quality population, ensuring that the model receives informative evolutionary trajectories rather than noisy or low-value samples. This combination of in-context learning with principled population management is what truly enables HELIX to achieve effective evolutionary improvement.
>
> ## Reference:
>
> **[1]** Deb, Kalyanmoy, et al. "A fast and elitist multiobjective genetic algorithm: NSGA-II." *IEEE transactions on evolutionary computation* 6.2 (2002): 182-197.

---

> ### Author Response · Authors · 2025-11-26
> **Checking for Any Additional Comments**
>
> Dear Reviewer,
>
> I hope this message finds you well. As the discussion period is nearing its end with **only one week remaining**, I wanted to ensure we have addressed all your concerns satisfactorily. If there are any additional points or feedback you'd like us to consider, please let us know. Your insights are invaluable to us, and we're eager to address any remaining issues to improve our work.
>
> Thank you for your time and effort in reviewing our paper.

---

### Author Response · Authors · 2025-11-20
**Method Detail**

We appreciate the reviewer’s thoughtful comments regarding the algorithmic framework. To provide a clearer understanding of the underlying mechanism of the HELIX framework, we present below a detailed explanation.

Formally, the algorithm begins with the problem description $p$ and one (or a set of) initial solution(s) $s_0$, which form the initial dataset $\mathcal{D}_0 = \lbrace s_0\rbrace$. The evolutionary population is also initialized as $\mathcal{P}_0 = \mathcal{D}_0$.

For the $t$-th iteration, the framework proceeds as follows:

1. **Sample from population**:
   Given the batch size $B$, we randomly select $B$ samples from the current population $\mathcal P_t$, denoted as $\lbrace s_{t, i}\rbrace _{1 \le i \le B}$.

2. **Prompt construction**:
   To leverage the **in-context learning** ability of LLMs, we construct the prompt $q_i$ for each sampled state $s_{t,i}$ by integrating itself and its $n$ ancestral states in the lineage tree (a historical trace of the solution $s_{t,i}$'s iterative refinement), i.e., $\lbrace f^{(k)}(s_{t, i})\rbrace_{1\le  k < n}$, along with their corresponding **rewards** $\lbrace R(f^{(k)}(s_{t,i}))\rbrace_{k < n}$ and **feedbacks** $\lbrace F(f^{(k)}(s_{t,i}))\rbrace_{k < n}$.
   Formally,

   $$
   q_i = \text{ConstructPrompt}(\lbrace p\rbrace  \cup \lbrace s_{t,i}, R(s_{t,i}), F(s_{t,i})\rbrace  \cup \lbrace f^{(k)}(s_{t,i}), R(f^{(k)}(s_{t,i})), F(f^{(k)}(s_{t,i}))\rbrace _{1\le k<n}).
   $$
   Here, $R(\cdot)$ represents the reward function that measures the quality of a generated solution, and $F(\cdot)$ denotes the auxiliary feedback (e.g., textual or structured evaluations) provided by the evaluator to guide future refinements.

   * **Explanation of lineage tree and ancestral states**:
     In evolutionary algorithms, each new state $s_{t+1}$ is derived from a previous state $s_t$ through a mutation $a_t$, i.e., $s_{t+1} = T(s_t, a_t)$, where $T$ is the transition function. This recursive relationship naturally forms a tree structure — the **lineage tree** — and the **ancestral states** of $s_t$ are the preceding nodes along its evolutionary path.

3. **Model rollout and evaluation**:
   The policy model $\pi_\theta$ generates $G$ rollouts for each input prompt $q_i$, resulting in

   $$
   \lbrace a_{i,j}\rbrace_{1 \le i \le B,\, 1 \le j \le G},
   $$
   and we can apply such mutation to current state $s_{t,i}$ and get $s_{t+1,i,j} = T(s_{t,i}, a_{i,j})$. After evaluation, we have the reward and textual feedback for each state:
   $$
   r_{t+1, i,j} = R(s_{t+1, i, j}), \quad f_{t+1, i,j} = F(s_{t+1, i, j}).
   $$
   The feedback $f_{t+1,i,j}$ is retained as contextual information for future iterations.

4. **Reinforcement learning update**:
   The rewards are used to compute the normalized advantage:


   $$
   \tilde r_{t+1,i,j} = \frac{r_{t+1,i,j} - mean_j \lbrace r_{t+1,i,j} \rbrace }{std_j \lbrace r_{t+1,i,j}\rbrace },
   $$

   and the policy parameters are updated according to the GRPO objective:
   $$
   \theta \leftarrow \theta - \gamma \cdot \nabla_\theta \mathcal{L}_{\text{GRPO}}.
   $$

5. **Diversity estimation**:
   An embedding model $E$ encodes each new solution into a latent representation:
   $$
   h_{t+1,i,j} = E(s_{t+1, i, j}).
   $$
   The diversity score $\text{Div}(s)$ is then computed according to Eq. (6) in the updated manuscript. (Eq. (5) for the old version of manuscript).

6. **Evolutionary population construction**:
   To balance **diversity** and **quality**, we employ the **NSGA-II** algorithm to construct the next population. NSGA-II is a multi-objective selection mechanism that identifies solutions with both high reward $R(s)$ and high diversity $\text{Div}(s)$.
   The updated population is:

   $$
   \mathcal P_{t+1} = SelectTop(\bigcup_{0\le s\le t+1}\mathcal{D}_s).
   $$

We have included these details in the revised manuscript to further enhance the completeness and clarity of the method description.

---

### Meta-Review · Area_Chair_MpdK · 2026-01-06

**Summary:**

This paper introduces a framework for enhancing the reasoning capabilities of large language models by integrating reinforcement learning with verifiable rewards, evolutionary algorithms, and in-context learning.

The reviewers agree that the combination of these techniques is novel and well motivated, even though each component is individually well established. They also appreciate the strong empirical performance demonstrated across a variety of settings, as well as the availability of an open-source implementation.

Aside from requests for clarification, the main concerns raised by the reviewers relate to presentation clarity, the lack of a more systematic analysis of hyperparameter sensitivity, and the need for a more thorough ablation study.

These concerns were addressed during the review process, and the paper is now considered strong enough for acceptance

**Reviewer Concerns:**

As far as the AC can see, all concerns were largely addressed.

**Reviewer Scores:**

At least slight positive score adjustments are expected, and one reviewer explicitly stated this.

---

### Decision · Program_Chairs · 2026-01-26

Accept (Poster)